# Sustainable Management of Thickened Tailings in Chile and Peru: A Review of Practical Experience and Socio-Environmental Acceptance

**Carlos Cacciuttolo Vargas [1],* and Alex Marinovic Pulido [2]**

1   Civil Works and Geology Department, Catholic University of Temuco, Temuco 4780000, Chile
2   Facultad de Ingeniería, Universidad Privada del Norte, Cajamarca 06001, Peru
*   Correspondence: ccacciuttolo@uct.cl or carlos.cacciuttolo@gmail.com

**Abstract:** The "Thickened Tailings Disposal" (TTD) technology produces a high density mine waste that allows for the storage of this material without the need to manage large slurry tailings storage facilities (TSFs) and large dams. TTD has been applied considering site specific conditions of Chile and Peru, such as extreme climatic conditions, seismic risks, water scarcity, community demands, and environmental constraints. This review highlights the contribution of several experiences in Chile and Peru, which have chosen TTD technology for reduction of negative environmental impacts, mainly focusing on the following issues: (i) increase of tailings water recovery, (ii) reduction of TSFs footprint (impacted areas), (iii) decrease the risk of physical instability, avoiding the construction of high dams, and (iv) decrease of TSFs seepages. Finally, the article describes the advantages (benefits) and disadvantages (aspects to improve) of TTD, where nowadays a high degree of dewatering of tailings is seen as a safe option, considering the occurrence of some TSF dam failures recently worldwide, which has resulted in severe environmental pollution. A better environmental perception about TTD of authorities and communities, considering that this technology allows to satisfy the needs of stable and safe TSFs, make the TTD be more acceptable, popular and one of the best available technologies (BATs) for operations with mine tailings.

**Keywords:** thickened tailings; paste tailings; thickened tailings disposal (TTD); tailings thickening; positive displacement pumps; tailings beach slope; down valley discharge; cell dyke disposal; tailings storage facility

## 1. Introduction

Dry climate, water scarcity, community issues, and environmental constraints in Chile and Peru, make the efficient use of water and care for the environment important aspects in mining. For these reasons, and for a better perception of authority and community for thickened tailings management, considering that it is satisfying the need of stable TSFs, make it more acceptable, popular and one of the best available technologies (BATs) [1]. In recent years, the improvements in thickened tailings technologies (thickeners and high-pressure pumps), have allowed increased water recovery and transport of high viscosity tailings, respectively. These technologies have been successfully applied in Chile and Peru for production rates up to 100,000 mtpd; showing good performance improvements on large-scale projects with high ore production rates. In this scenario, there is still a need for more reliable equipment for paste tailings thickening plants on large scale, focusing on the tailings water recovery enhancing for its reuse in mining processing [2].

Considering the BATs for tailings management incorporating tailings dewatering techniques is possible to mention four main categories: (i) Conventional Tailings, (ii) Thickened Tailings, (iii) Paste Tailings and (iv) Filtered Tailings.

Conventional copper tailings typically range 25–40% solids weight concentrations (Cw), thickened copper tailings 40–65% solids weight concentrations (Cw), paste copper

tailings 65–80% solids weight concentrations (Cw), and filtered copper tailings over 80% solids weight concentrations (Cw) (solid concentrations may vary with particle size and shape, clay content, mineralogy, electrostatic forces and flocculant dosing).

Conventional, thickened, paste, and filtered tailings refers to a continuum of tailings with high solid concentrations and higher yield stress, due to the greater level of fluid removal from tailings before disposal.

In current Chilean and Peruvian large-scale mining in dry climate areas, most typical tailings disposal schemes consist of conventional or slightly thickened at modest levels of tailings solids weight concentration (Cw 25–40%). Conventional TSFs have dams built of the coarse fraction of tailings (cycloned tailings sand) obtained by hydrocyclones or have slightly thickened tailings deposits with dams built of borrowed material. Conventional tailing dams may have water recoveries as high as 65–75% in very well-operated TSFs, which means they have appropriate tailings distribution, good control of the pond (volume and location), and adequate seepage recovery. In conventional dams, water at the settling pond is decanted by floating pumps, or decant towers, and dam seepages are collected by a drainage system and cutoff trench systems. However, a high seasonal evaporation rate can substantially reduce water recovery from the pond area, and infiltration from the pond in contact with natural soil can produce water losses. Some mining operations with this technology are: Cerro Verde (Peru), Cuajone and Toquepala (Peru), Los Pelambres (Chile) and Los Bronces (Chile) [3,4].

Thickened Tailings Disposal (TTD) technology requires more background data than conventional tailings disposal. In the conventional approach, the properties of tailings are fixed by the concentrator plant, whereas in a TTD impoundment, the properties of the tailings and their placement are "engineered" to suit the topography of the disposal area [5]. The behavior of tailings in the two approaches is entirely different. In conventional disposal, tailings segregate as they flow and settle out to an essentially flat deposit, whereas in TTD technology, a sloping surface is obtained. The principal difference is that, in TTD technology, tailings are thickened before discharge to a homogeneous heavy consistency that results in laminar non-segregating flow. In this way, TTD produces high water recovery (70% of tailings water recovery) and a self-supporting deposit with sloping sides, requiring small dams. Some mining operations with this technology are: Toromocho (Peru), Constancia (Peru), Centinela (Chile), and Sierra Gorda (Chile) [3,4].

Paste Tailings Technology has been applied on a small production scale because a limitation of equipment manufacturing ability exists. This method permits obtaining a medium make-up water requirement (80% of tailings water recovery). However, in some cases, there are difficulties in tailings transportation requiring the use of positive displacement pumping (PD Pumps), resulting in the highest capital/operating costs. The main advantage of this method is that large dams are not required; only small dams are needed. Some mining operations with this technology are: Chungar (Peru), Cobriza (Peru), Las Cenizas (Chile), and Alhue (Chile) [3,4].

In the last 20 years, many mining projects around the world have applied a tailings disposal technology called dry stacking of filtered tailings. This technique produces an unsaturated cake that allows storage of this material without the need to manage large slurry tailings ponds. The application of this technology has accomplished: (i) an increase in water recovery from tailings (90%), (ii) a reduction of TSF footprint (impacted areas), and (iii) a decrease in the risk of physical instability because TSFs are self-supporting structures under compaction (such as dry stacks), and (iv) a better community perception. Some mining operations with this technology are: (i) Cerro Lindo (Peru), Catalina Huanca (Peru), El Peñon (Chile), and Mantos Blancos (Chile) [3,4].

The importance of this article is to present the advances that have been achieved in the last 20 years in the implementation of thickening of mine tailings in large-scale mining projects, considering, for example, mine tailings production of the order of 100,000 mtpd. Both in Chile and Peru, the socio-environmental conditions and restrictions due to the

demands of the community, as well as the care of freshwater resources in the basins, have favored the implementation of TTD technology.

Today the communities demand that the mining companies carry out tailings management that is more responsible, controlled and respectful of the environment. The tailings dam failure events recorded in Mount Polley Canada (2014), Fundao Samarco Brasil (2015) and Corrego de Feijao Brumandinho Brazil (2019) have severely impacted communities and the environment, causing mining companies to reassess their management and governance of mine tailings [6,7]. Improvements in standards, management systems, engineering designs and quality assurance control in construction processes have been some of the measures considered to carry out safer and more controlled tailings management [8].

Considering the lessons learned from recent tailings dam failure events, it is possible to mention that tailings impoundments using conventional tailings technology represent a safety risk because they store large amounts of water. This is how mining tailings thickening technologies allow a considerably smaller amount of water to be stored in tailings storage facilities, reducing the risks of liquefaction, piping (internal erosion by seepage) and overtopping [7,9,10].

Environmental aspects that consider the reduction of fresh water in mining processes have favored the implementation of thickened tailings technologies, which efficiently recover water from thickening equipment. Advances in the development of thickening equipment and the use of flocculants have allowed TTD technology to position itself as an attractive alternative for sustainable tailings management. On the contrary, conventional tailings deposits have water losses that are difficult to control due to seepage, evaporation in the supernatant pond and tailings beaches [11].

Advances in both centrifugal and positive displacement pumping systems have allowed a better insertion of thickened tailings technology, managing to transport mining tailings hydraulically in pipes over long distances, steep topographies and at high pressures. Reduction in the capital costs of thickening and pumping equipment has allowed thickened tailings technologies to be more competitive against conventional tailings and filtered tailings [12].

TTD technology has been applied considering site specific conditions of Chile and Peru, such as: extreme climatic conditions, seismic risks, water scarcity, community demands, and environmental constraints. This review highlights the contribution of several experiences in Chile and Peru, which have chosen TTD technology for reduction of negative environmental impacts, mainly focusing on the following issues: (i) increase of tailings water recovery, (ii) reduction of TSFs footprint (impacted areas), (iii) decrease the risk of physical instability, avoiding the construction of high dams, and (iv) decrease of TSFs seepages.

Finally, this review describes the advantages (benefits) and disadvantages (aspects to improve) of TTD, where nowadays a high degree of dewatering of tailings is seen as a safe option, considering the occurrence of some TSF dam failures recently worldwide, which has resulted in severe environmental pollution. A better environmental perception about TTD of authorities and communities, considering that this technology allows to satisfy the needs of stable and safe TSFs, make the TTD be more acceptable, popular and one of the best available technologies (BATs) for operations with mine tailings.

## 2. Tailings Thickening Plants Development and Advances

Mine operators and thickeners suppliers have gained experiences carrying out projects during the last decades, learning that each particle size distribution (PSD), mineralogy and rheology of tailings exhibit their own unique thickening behavior, making efficient and reliable solid/liquid separation units. Figure 1 shows a tailings continuum concept with a schematic copper tailings thickener classification.

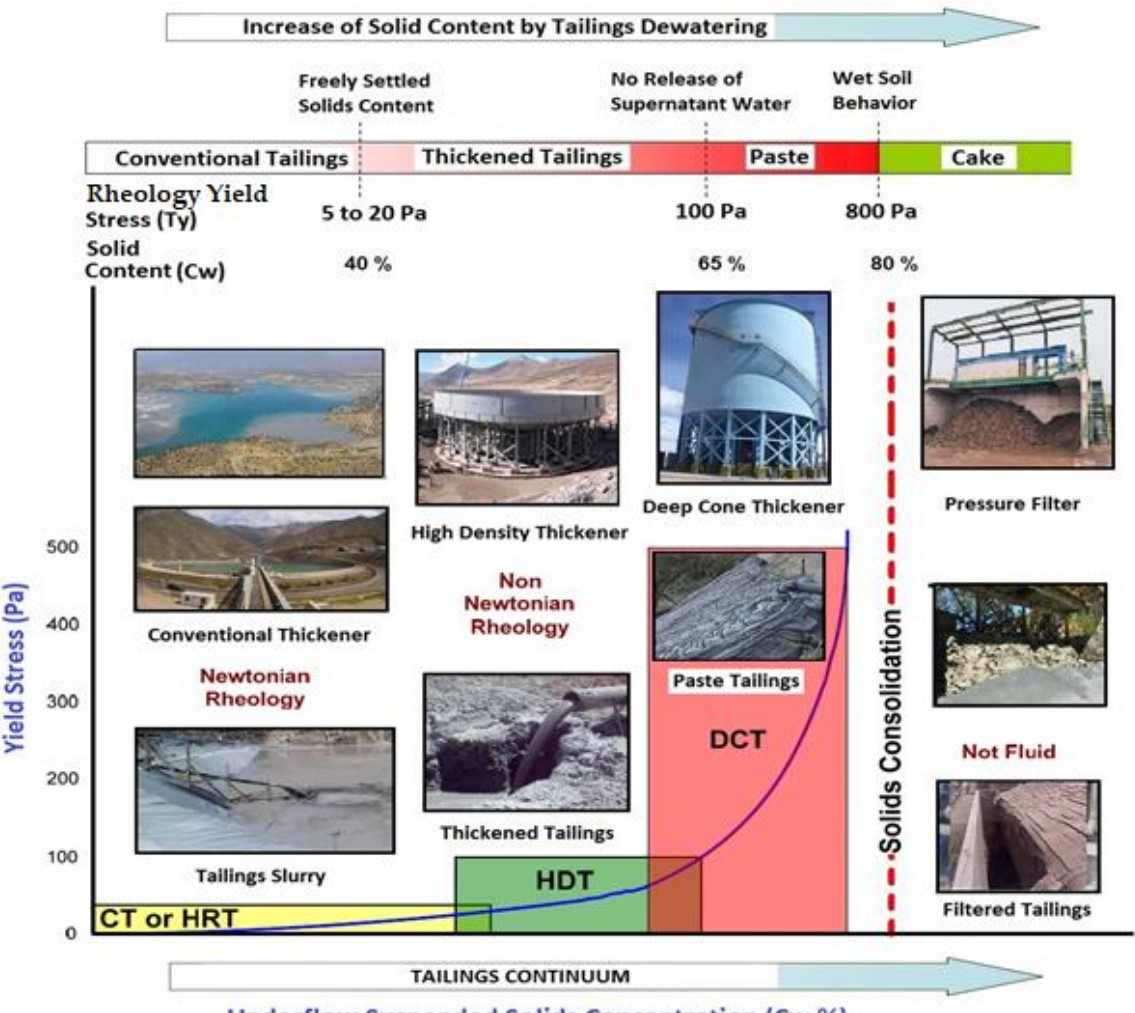

**Figure 1.** Dewatering Tailings Technologies—Tailings Continuum [13].

## 2.1. Conventional Thickeners (CT)

A conventional thickener typically use of flocculent (polymer) is not needed, however, there are some conventional thickeners that use flocculent, to improve overflow clarity, handle a higher tonnage, or aid in achieving the desired underflow density. These units are often fairly simple, due to the relatively large size; they are somewhat forgiving in operation and can have the storage capacity to absorb some plant upsets without affecting downstream operations. Typical features include a drive and rakes, a relatively shallow feedwell, and a bridge to support the feed pipe or launder and allow center access [14].

These units are sized based on the settling flux rate of the smallest particle and segregation of small and large particles is the norm. Typically, the maximum suspended solids concentration in the underflow slurry is defined by operating at less than maximum rake drive torque or at underflow concentrations that are dischargeable to prevent plugging. The underflow slurries produced are at a suspended solids concentration typically exhibiting Newtonian rheology. A few examples of CT applications in Chile are: (i) Chuquicamata Mine (09 units of 91 m diameter), (ii) El Teniente Mine (07 units of 100 m diameter) see Figure 2, and (iii) Candelaria Mine (02 units of 125 m diameter). Some examples of CT equipment in Peru are: (i) Toquepala Mine (02 unit of 80 m diameter), and (ii) Cuajone Mine (03 units of 100 m diameter).

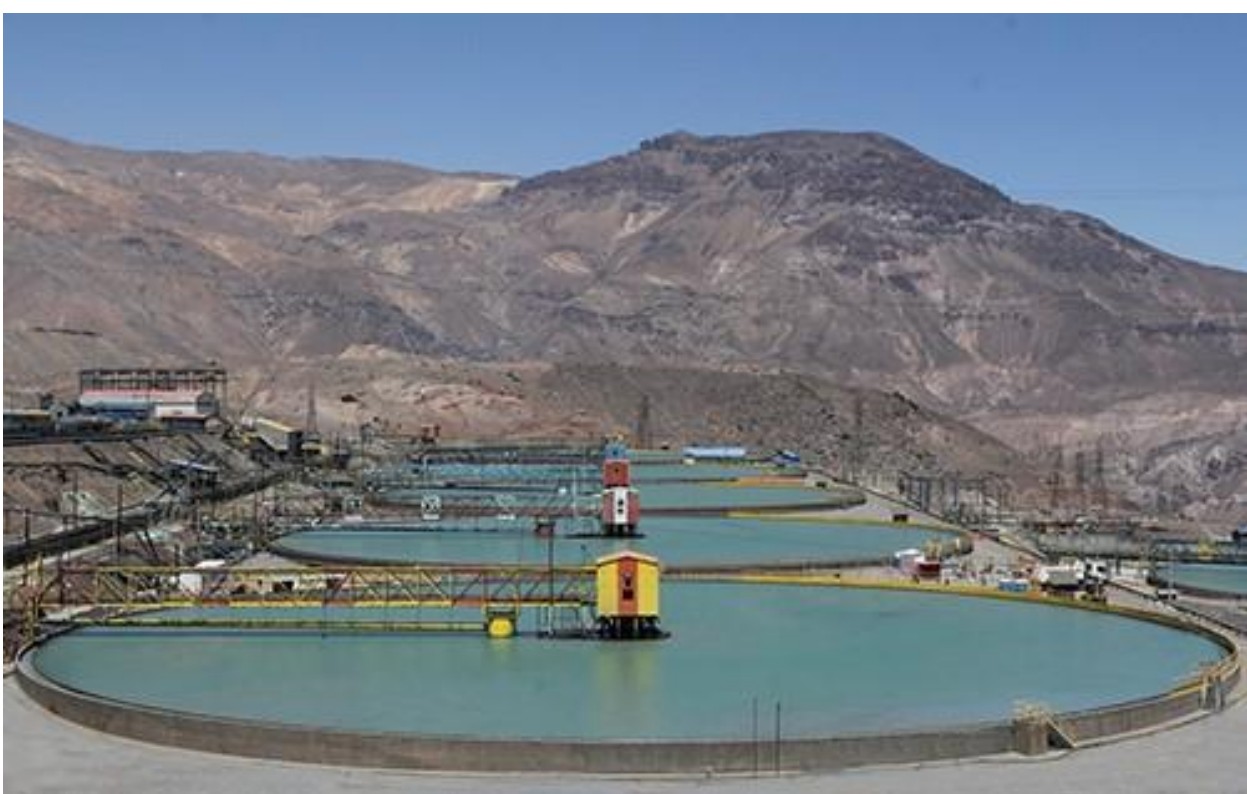

**Figure 2.** Tailings Conventional Thickeners in El Teniente Mine—Chile.

*2.2. High Rate or High Capacity Thickeners (HRT)*

With the advent of synthetic flocculent, the terms High-Rate and High-Capacity emerged as a type of thickener, as the throughput rates for the now flocculated feed slurries were considerably higher than for un-flocculated slurries. These units are sized using flocculants to produce an underflow slurry with minimal particle segregation. The underflow slurries produced are at suspended solids concentrations typically exhibiting Newtonian rheology and a yield stress less than 20 (Pa), avoiding discharge problems [14].

Thickener size or throughput is directly dependent on flocculent dose and feed slurry concentration because of this, most high-rate thickeners use feed dilution systems. The optimum size of these thickeners is governed by capital and the primary operating cost of flocculent, where flocculation is required, and feed slurry dilution systems are often needed for optimal performance. High-rate thickeners are generally small to medium sized bridge type thickeners, although large center column thickeners processing very high tonnage can also fall into this category. Common features of these devices include a deep self-diluting feed well, heavy duty drive, streamlined rake arms, and large effluent launders and underflow outlets. A few examples of HRT applications in Chile are: (i) Collahuasi Mine (02 units of 125 m diameter), (ii) Carmen de Andacollo Mine (01 units of 70 m diameter), and (iii) Los Pelambres Mine (03 units of 125 m diameter) see Figure 3. Some examples of HRT equipment in Peru are: (i) Las Bambas Mine (02 unit of 80 m diameter), (ii) Antapaccay Mine (02 unit of 60 m diameter), (iii) Constancia Mine (01 unit of 75 m diameter), (iv) Toquepala Mine (03 units of 60 m diameter) and (v) Quellaveco Mine (02 units of 120 m).

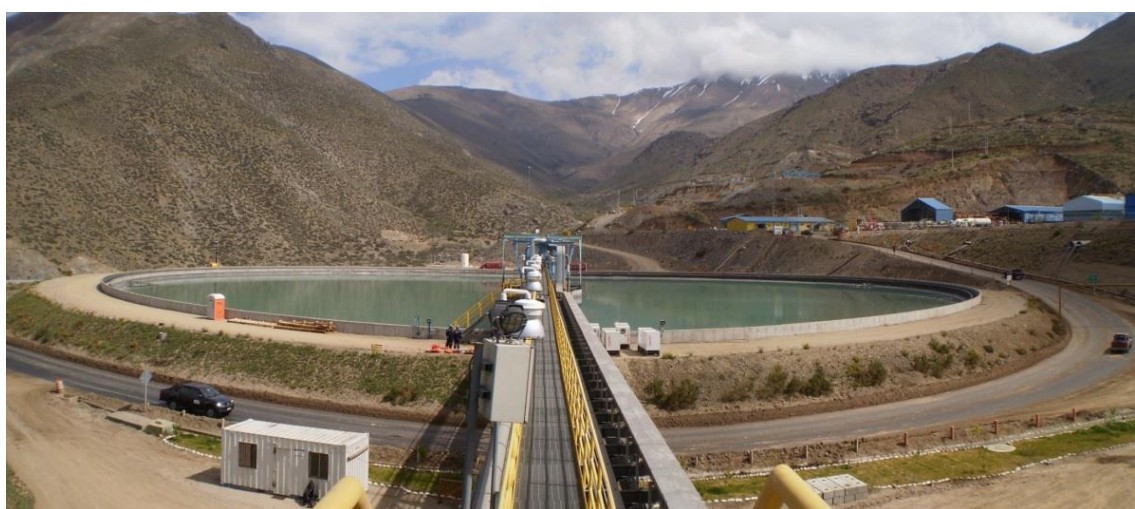

**Figure 3.** Tailings High-Rate Thickeners in Los Pelambres Mine—Chile.

*2.3. High Density Thickeners (HDT) or High Compression Thickeners (HCT)*

This technology is an extension of HRT thickening, utilizing a deeper mud bed to augment the thickening capacity. These devices usually add depth to a high-rate design to aid in increasing the underflow density. Deeper mud beds increase the mud compressive force, reducing the time required for thickening and increasing the underflow density [14]. These units are sized based on optimized flocculation or maximum solids flux rate. HDT's are designed to produce a slurry that can be pumped by a centrifugal pump and typically have all of the physical design features of a paste thickener but simply are controlled to produce an underflow slurry exhibiting a yield stress of about 100 Pa, requiring significantly more torque than HRT's due the increased mud viscosity. The underflow slurries produced are at a suspended solids concentration typically exhibiting Non Newtonian rheology A few examples of HDT applications in Chile are: (i) Caserones Mine (03 units of 45 m diameter), (ii) Cerro Negro Norte Mine (02 units of 40 m diameter) see Figure 4, and (iii) Esperanza Centinela Mine (03 units of 60 m diameter and 03 units of 45 m diameter). Some examples of HDT equipment in Peru are: (i) Toromocho Mine (04 unit of 40 m diameter), (ii) Las Bambas Mine (03 unit of 50 m diameter), and (iii) Cerro Verde Mine (04 units of 80 m diameter).

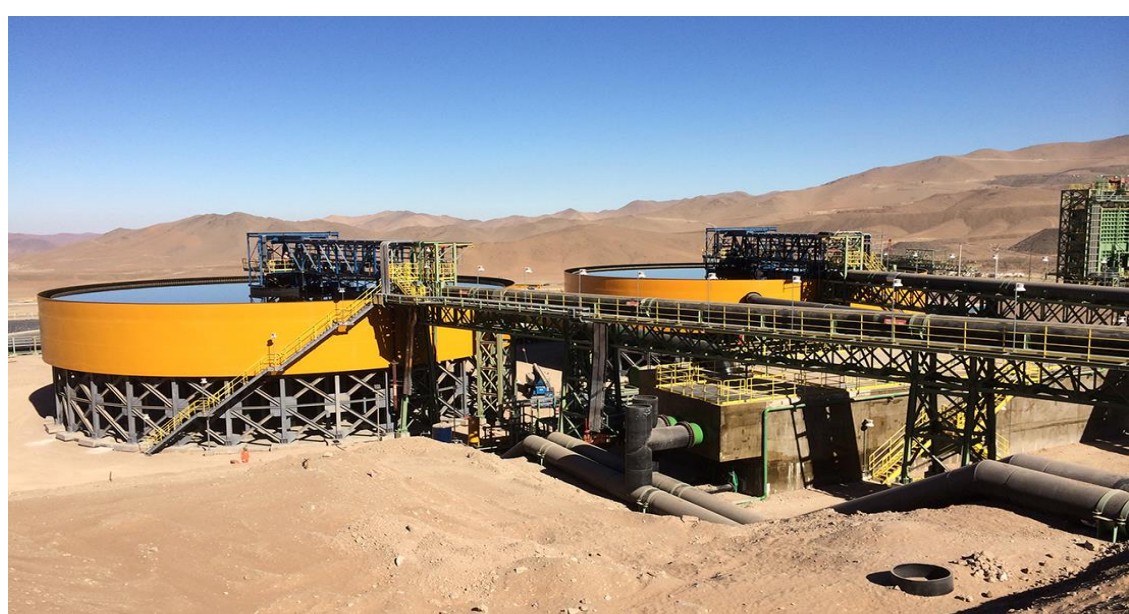

**Figure 4.** Tailings High Density Thickeners in Cerro Negro Norte Mine—Chile.

### 2.4. Paste or Deep Cone Thickeners (DCT)

DCT are designed to produce and discharge underflow slurries exhibiting yield stresses of less than 300 Pa. Paste thickeners are predominately used for paste tailings disposal with the consistency of the underflow slurry determined and controlled to meet design criteria of the disposal site and transport pipeline—high pressure pumping system called positive displacement pumps (PD Pumps). These units are sized based on optimized flocculation or maximum solids flux rate, typically utilize very deep mud beds in order to take maximum advantage of mud compressive forces for dewatering and provide sufficient time for the mud to dewater to a paste consistency. The tank height to diameter ratio is frequently 1:1 or higher. Due to the high underflow viscosities, mechanism torques can be 5–10 times higher than HRT on similar tailings. The underflow slurries produced are at a suspended solids concentration typically exhibiting Non Newtonian rheology A few examples of DCT applications in Chile are: (i) Las Cenizas Mine (01 unit of 17 m diameter) see Figure 5, (ii) Delta plant (01 unit of 12 m diameter), (iii) Collahuasi Mine (01 unit of 22 m diameter), (iv) El Toqui Mine (01 unit of 14 m diameter) and (v) Alhue (01 unit of 17 m diameter). Some examples of DCT equipment in Peru are: (i) Chungar Mine (01 unit of 17 m diameter), (ii) Cobriza Mine (01 unit of 14 m diameter) and (iii) Rumichaca Carahuacra Mine (01 unit of 20 m diameter).

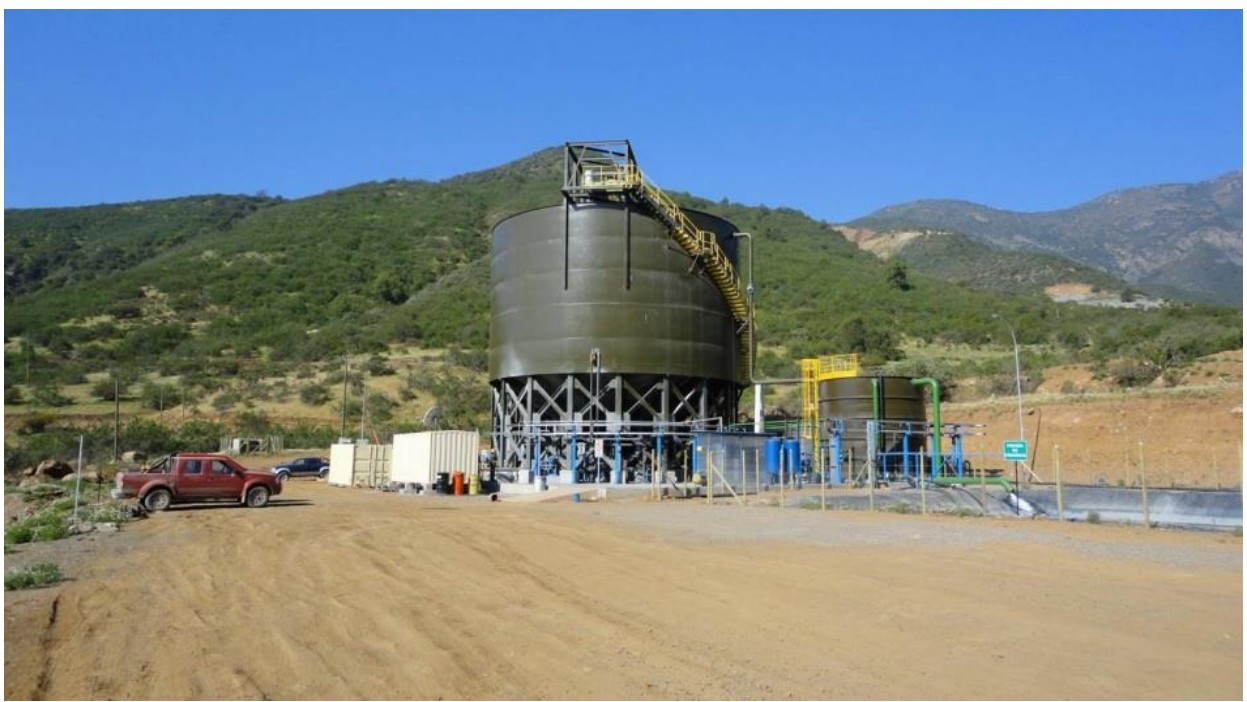

**Figure 5.** Tailings Deep Cone Thickener in Las Cenizas Mine—Chile.

In the case of HDTs and DCTs, the sizing procedures generally must take into account the volume of thick pulp within the thickener, as a substantially longer than normal retention time is necessary for the solids to obtain high concentrations. It is generally more economical to provide this volume by using a relatively deep bed of pulp rather than a greater area with a shallow compaction bed. This increase in depth also provides bed compression, where the weight of solids above helps compress and dewater the mud to higher concentrations [15]. Table 1 shows a characterization of different tailings thickeners applied in Chile and Peru.

**Table 1.** Copper Tailings Thickener Characteristics [16,17].

| Parameter | Units | CT | HRT | HCT or HDT | Paste or DCT |
|---|---|---|---|---|---|
| Solid loading | tph/m$^2$ | (0.02–2.0) | (0.35–1.5) | (0.4–1.0) | (0.3–0.8) |
| Maximum diameter | m | 125 | 100 | 80 | 30 |
| Flocculent dosing use | g/ton | No use | 10–15 | 20–30 | 30–50 |
| Tank thickener height | m | 1–3 | 3–5 | 4–8 | >8 |
| Cone angle | ° | 2–5 | 5 | 5–15 | 15–30 |
| U/F Solid content by weight | % | 30–45 | 45–60 | 60–65 | 65–75 |
| Tailings Bed Depth | m | <1 | 1–2 | 3–4 | 4–10 |
| Residence Time | h | 0.5–1 | 1–2 | 3–6 | 6–12 |
| Yield Stress | Pa | 0 | 0–30 | 30–50 | 50–300 |
| Torque K Factor | kN/m | <50 | <50 | <125 | >200 |
| Shear Thinning System | - | No | No | No–Yes | Yes |

As the thickened mud approaches a limiting concentration, it behaves less and less similar to a fluid, and has little tendency to flow to the underflow withdrawal point. Therefore, steep floors of 30–60 degrees and rakes designed to overcome the yield stress of the mud are used to transport the thickened mud to the outlet. By contrast, standard thickener design calls for a slope of something less than 10 degrees. When the pulp reaches the compression zone, mechanical action, such as by the raking mechanism itself, contributes to the rate of water removal from the compacting mass. Since DCTs operate with pulp depths which generally extend well above the rake structure, the mass of material located in the zone is not exposed to a similar mechanical action. Therefore, it is helpful to add pickets, usually consisting of posts or rods which project into this mass, in order to create channels and assist in water removal [15,18]. The use of low drag designs is important to minimize the torque required. Figure 6 shows different tailings thickeners applied in Chile and Peru.

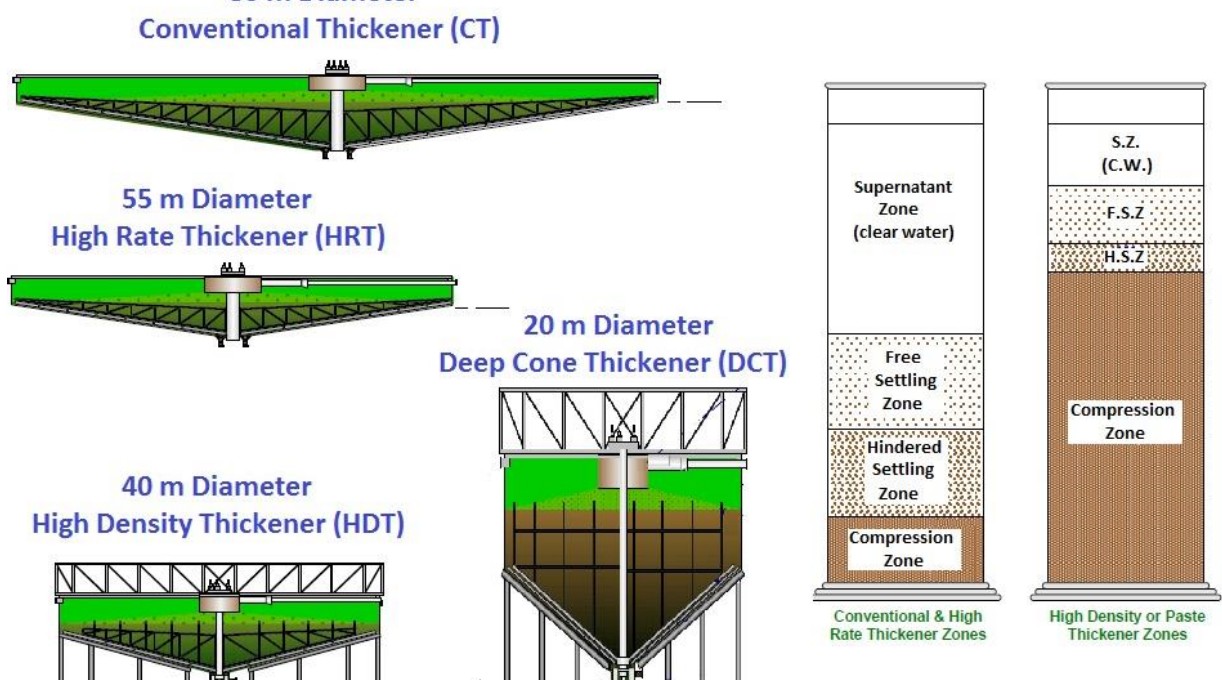

**Figure 6.** Different Tailings Thickeners applied in Chile and Peru Mining Operations.

## 3. Control and Performance of Tailings Thickening Plant

During thickener operation, it is often desired to control the underflow solids concentration to be within a certain range. In the copper mining industry, tailings should be thickened to a sufficiently high solids content to recover water but not too high so as to damage the thickener or underflow pump. Important independent variables in thickener operation include: (i) underflow rate (U/F), (ii) feed solids rate (FR) and (iii) flocculent dosage to the feed tailings stream (FD). Underflow rate will determine the throughput of the thickener affecting the residence time and tailings product quality (density and Cw). Flocculent dosage affects the final settling velocity of tailings particles in hindered settling region as well as the compression of materials in the sediment, thus the underflow solids concentration. These two variables (U/F and FD) are commonly available to be manipulated by an automatic control system to maintain the underflow solids concentration within a desired operating range. However, the feed solids rate (FR), such as copper tailings properties which are determined by upstream processes (concentrator plant) become external disturbances to the thickener, which require attenuation and generally, FR is used only in an emergency to avoid impacting plant production [15].

Instrumentation is a key element in a thickening plant. There are two main functions to control in thickener operation to manage dependent variables including: (i) solids settling rate, (ii) underflow density (viscosity), (iii) solids interface level (bed depth), (iv) rake torque, (v) bed pressure (bed mass) and (vi) overflow turbidity. The thickeners are set up to control the flocculent pump speed to a tonnage ratio set point by the operators. The underflow pump speeds are controlled by the operators to maintain either a bed pressure set point, or a flow rate set point.

The two independent variables that are typically used for control of thickeners are underflow rate and flocculent addition rate. A third variable, the feed rate, is generally used only in an emergency to avoid impacting plant production. Figure 7 shows a schematic diagram of instrumentation control of thickener [19].

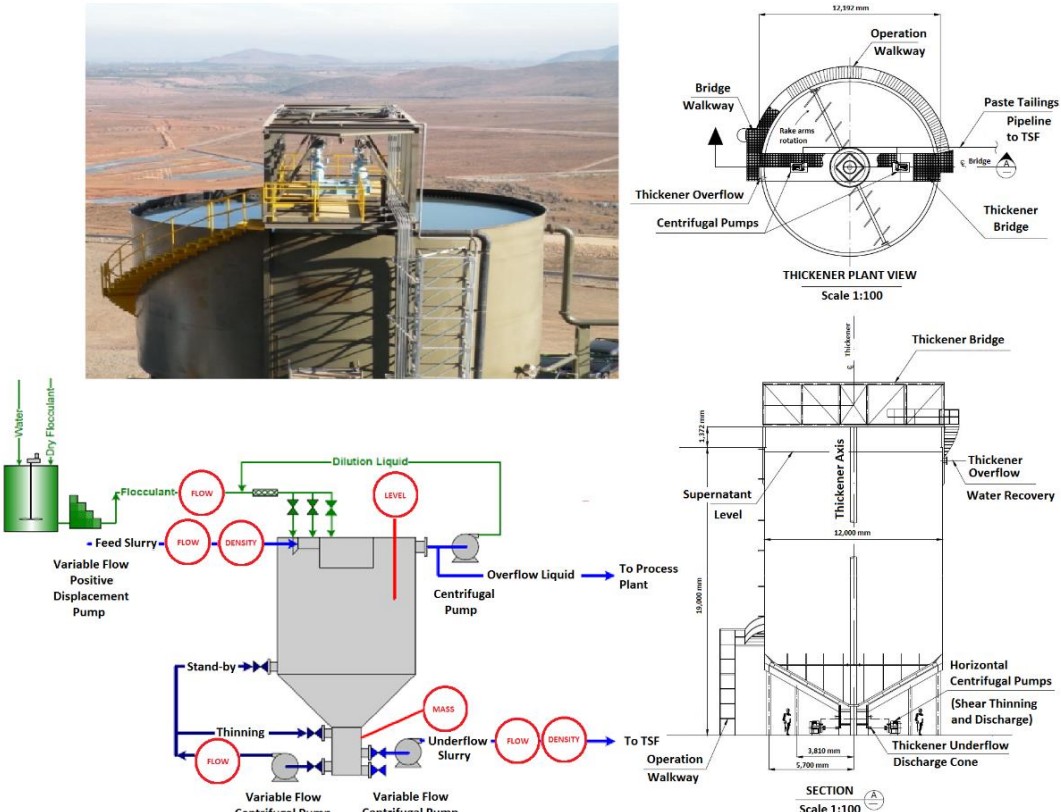

**Figure 7.** Example of paste tailings thickener plant control and performance.

Accurate control of the addition of flocculent to any thickener is important to achieve the best and most economic dewatering performance. This is especially true in a thickener where the thickening operating envelope is being pushed to its limit. The addition of flocculent to a thickener is recommended to be controlled in a direct way. Direct control of the flocculent addition requires a variable speed dosing pump to deliver a pre-determined dose based on the incoming tonnage of solids to the thickener.

It is key that the solid particles of small size present in tailings can adhere to each other to form larger particles, which settle faster, favoring the industrial thickening process. For this purpose, flocculating agents are added, which are polymers or polyelectrolytes with a long chain and high molecular weight and soluble in water. Flocculants adhere to various particles, generally through ionic bonds, giving rise to agglomerates of particles or flocs with a density and size suitable for sedimentation. They can be classified according to their nature as minerals, natural organics and synthetic organics, and also according to their electrical charge as anionic, cationic and neutral. The most used flocculants in the mining industry are organic synthetics, of an anionic nature and of high molecular weight [20] (Table 2).

**Table 2.** Commonly used Flocculants in Thickening of Mining Tailings [21].

| Manufacturer | Flocculent Name |
| --- | --- |
| SNF | Floerger 913-SH<br>Floerger 923-SH<br>Tec-2050 |
| BASF | Magnafloc 1011<br>Magnafloc 155<br>Magnafloc 2025<br>Magnafloc 333<br>Rheomax 1050 |
| Orica | Orifloc AP 2020 |
| Kemira | Superfloc A-110 |

The type of flocculent and the dose in which it is supplied must be carefully chosen through laboratory tests or pilot tests, so that the resulting flocs have an adequate density to minimize the energy consumption of hydraulic transport by pumping of mining tailings to the TSFs and maximize the recovery of supernatant water to be recirculated to the metallurgical mining process. For the next years, an increase in the consumption of flocculants in the mining industry is expected, due to the increase in mining projects and the implementation of the TTD technology [20].

## 4. Thickened Tailings Transport

In general, the consistency (solids content or density) of the tailings delivered to the TSF will be limited by the transportation distance and method of tailings transport (centrifugal pump, positive displacement pump, gravity, etc.). For example, centrifugal pumps are limited to a pipeline pressure of 15 Bar (equivalent to 1.5 MPa). Positive displacement pumps (PD Pumps) will pump much higher density materials at correspondingly higher discharge pressures of 125 Bar (equivalent to 12.5 MPa) (See Figure 8). The cost of installing and operating positive displacement pumps must be evaluated over the lifetime of the project to make a meaningful comparison with a system using multiple centrifugal pumps (or pump stations) required to generate a comparable pump discharge pressure.

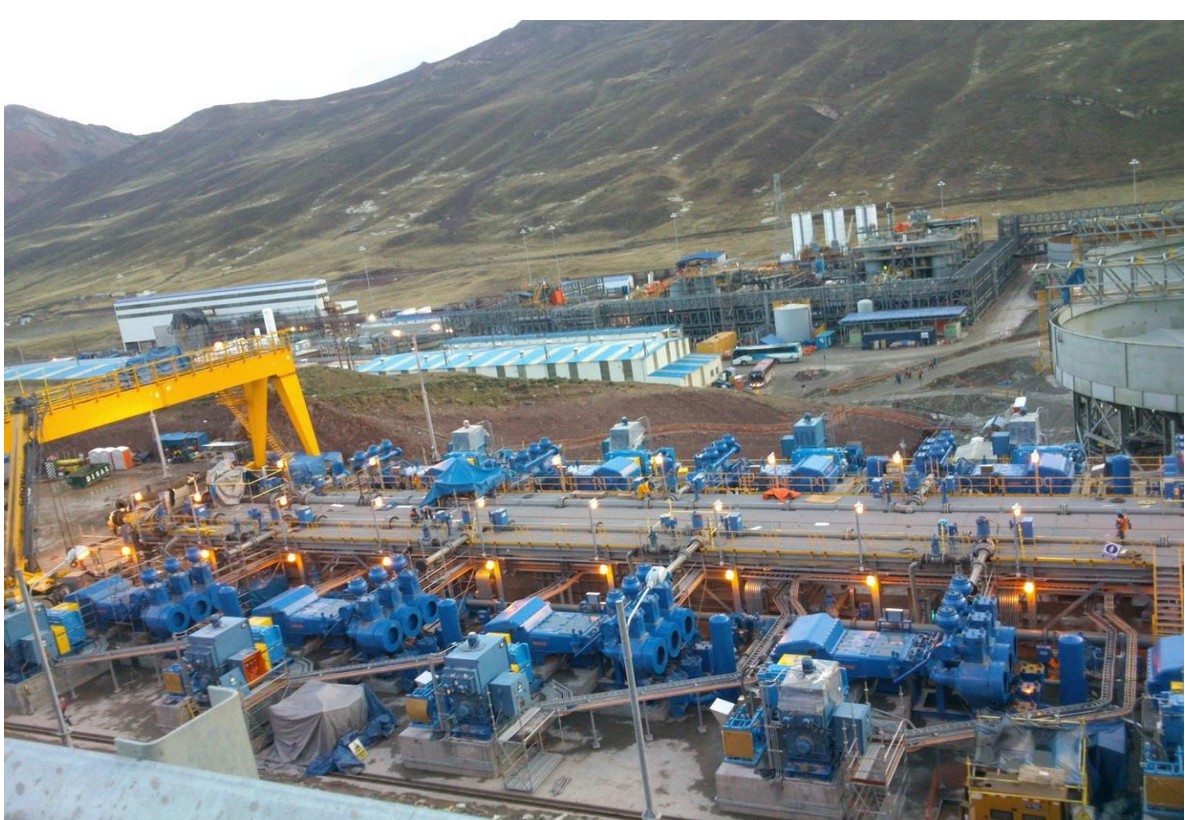

**Figure 8.** Thickened tailings positive displacement pump station—Toromocho Mining Project Peru.

A popular and easy method used in Chile and Peru by the thickened tailings thickening operators for indication of flow behavior is the standard 12 inches (305 mm tall) slump cone developed for concrete slump measurements (ASTM C143/C 143M-00) [22]. This test can be used to obtain a preliminary assessment for the thickened tailings pipeline flow behavior and enables quality control for paste tailings transportation to TSF. Considering only the process performance of copper tailings thickener, experience with pilot and full-scale operations suggests for slumps higher than 9 inches (230 mm), the tailings underflow yield stress will be a pumpable fluid. For slumps between 8 and 9 inches (200–230 mm), yield stress will be high (Ty > 150 Pa) and a shear thinning system should be evaluated as a possibility. For slumps of 8 inches (200 mm) and lower, shear thinning will be required (Ty > 300 Pa). These operating ranges are strongly determined by solids particle size, throughput rate, and pumping/pipeline shear. Thickened tailings have shearing properties in the pipeline, affected by thickener discharge pumping and pipeline flow [23].

The viscous behavior of underflow tailings slurries generated from DCTs can normally be characterized as being of a yield stress non-Newtonian nature. The yield stress of such slurries is usually in the range of 150 to 400 Pa. The DCT itself and downstream processes such as pumping or gravity flow in channels is improved by obtaining a lower yield stress at the same target underflow density, reducing flowability problems. The yield stress of thickened tailings can be modified by application of mechanical shear which breaks the network structure of flocculated paste tailings slurries. Typically, the concentric shear thinning system modifies the yield stress of the underflow paste tailings density from its unsheared thickened state (150–300 Pa) to a sheared yield stress range of 75–150 Pa [24]. Figure 9 shows a typical flow sheet for two alternative processes for shear thinning.

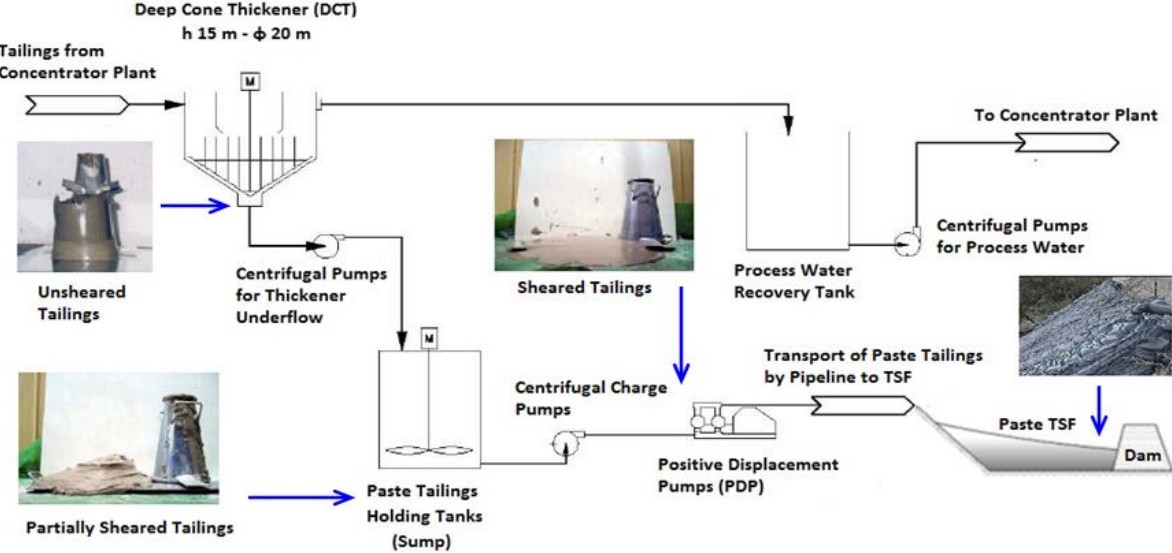

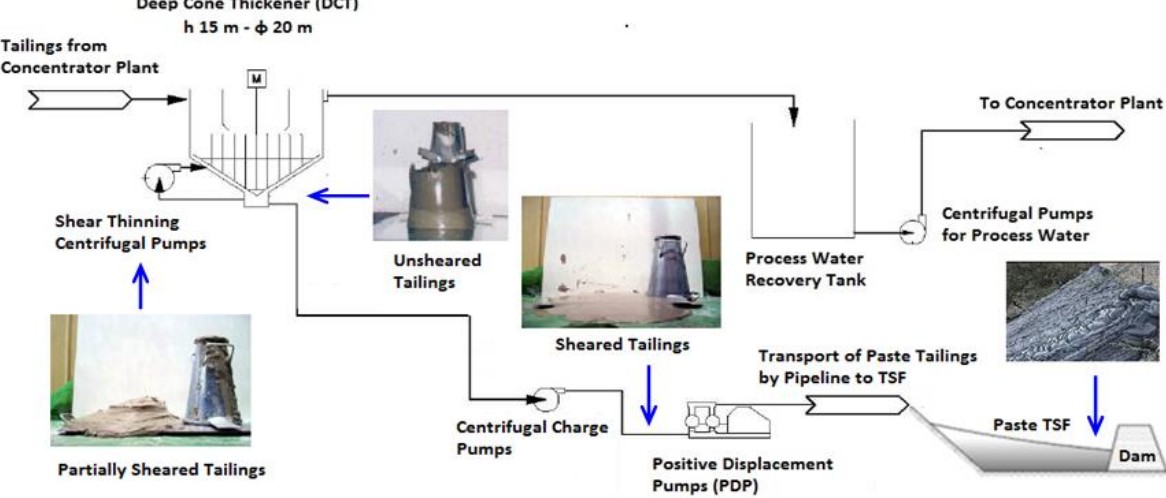

**Figure 9.** Alternative Processes for Paste Tailings Pump Transportation.

## 5. Thickened Tailings Disposal

The most important property of thickened tailings to permit the "stacking" of tailings, in a down-valley discharge (DVD) or cell dyke disposal (CDD), because thickened tailings are non-segregating and not having hydraulic sorting behavior on the beach [25].

One issue of increasing importance in tailings facility design and deposition planning is the ability to more accurately predict the beach slope during initial design before any measurements on the actual beach can be made. The use of thickened tailings adds emphasis to this as slopes significantly steeper than past experience with conventional slurry can be expected. The prediction becomes particularly important where the long beach lengths are planned and thus where a steeper slope could result in a large differential height on the containing structures. If a steep slope can be relied upon then it may be possible to construct a lower height dam at the lower end of the deposit (See Figure 10).

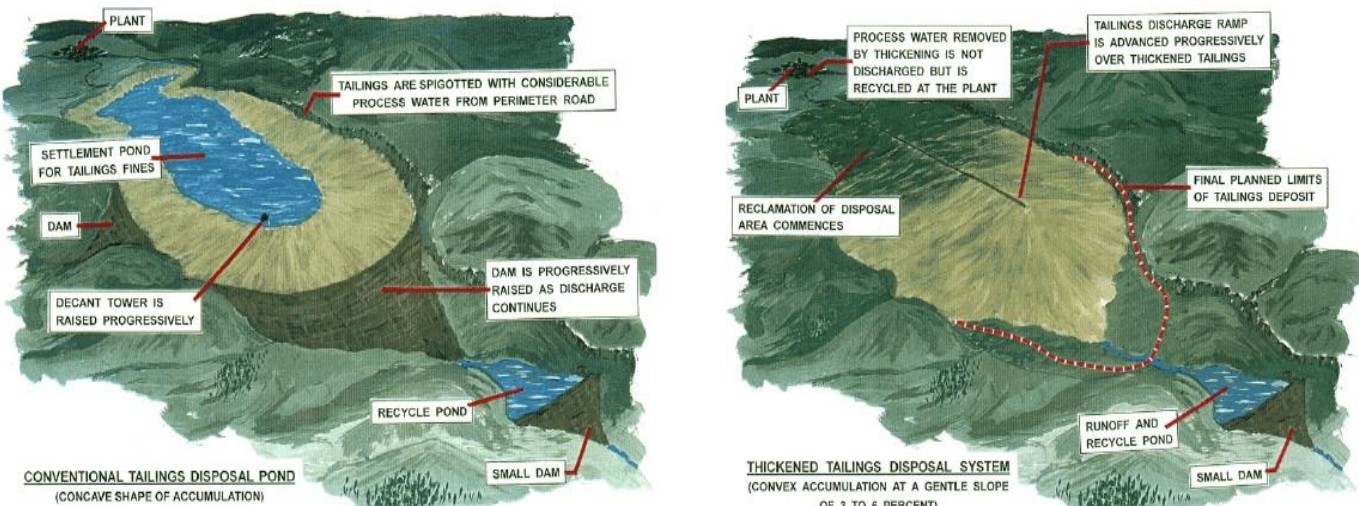

**Figure 10.** Comparison between Conventional tailings disposal method and Thickened tailings disposal system [25].

However, care must be taken to avoid over-predicting the slope angle since if it is not realized the result will be the need for an unexpected rise to the dam structure. Another consideration is seismic stability of the slope and the potential of a steeper beach to become mobilized if the tailings liquefy in an earthquake. Ideally the deposit is managed with a slope that is not mobilized by the design earthquake. However, if there is uncertainty, the tailings facility must be designed to withstand the mobilization if it occurs.

### 5.1. Down Valley Discharge (DVD)

Considering the steep topography of the Andean region of Chile and Peru, it is attractive to use topographical depressions to impound tailings, reducing dam volume since the sides of the valley serve to contain tailings. Typically, valley TSFs are constructed as a single facility, in which the tailings are contained behind a single dam or embankment.

In the case of thickened and paste tailings, if pumping occurs from the thickener to the highest topographical point of the valley, then tailings are discharged from spigots, and flow occurs by gravity down the valley to reach a retaining embankment at the toe of the TSF. This technique is called down-valley discharge and according to the practice, non-segregating slurries (thickened and paste tailings) for a constant percent of solids and flow rate will form a concave beach profile due to variability in thickener performance (See Figure 11).

Thickened and paste tailings are deposited hydraulically, or loosely, and beach, or settle, at somewhat steeper slopes than conventional tailings slurry. In theory, the beach slope can be up to 4.0%, however, in practice steep slopes are only achieved for a short distance and the remaining beach is sloped at less than 2.0%. The steeper beach slope of thickened and paste tailings storage facility, compared to a conventional slurry beach, provides an opportunity to store tailings above the dam elevation, which reduces the footprint and height of the dam [26].

Thickened and paste tailings should be, by definition, largely non-segregating (i.e., fine and coarse particles do not separate during deposition), however minor segregation could still occur depending on the tailings' particle size distribution and solids content at deposition [26].

Compared to conventional tailings storage facilities, less bleed water and consolidation water is released from thickened and paste tailings deposited [26].

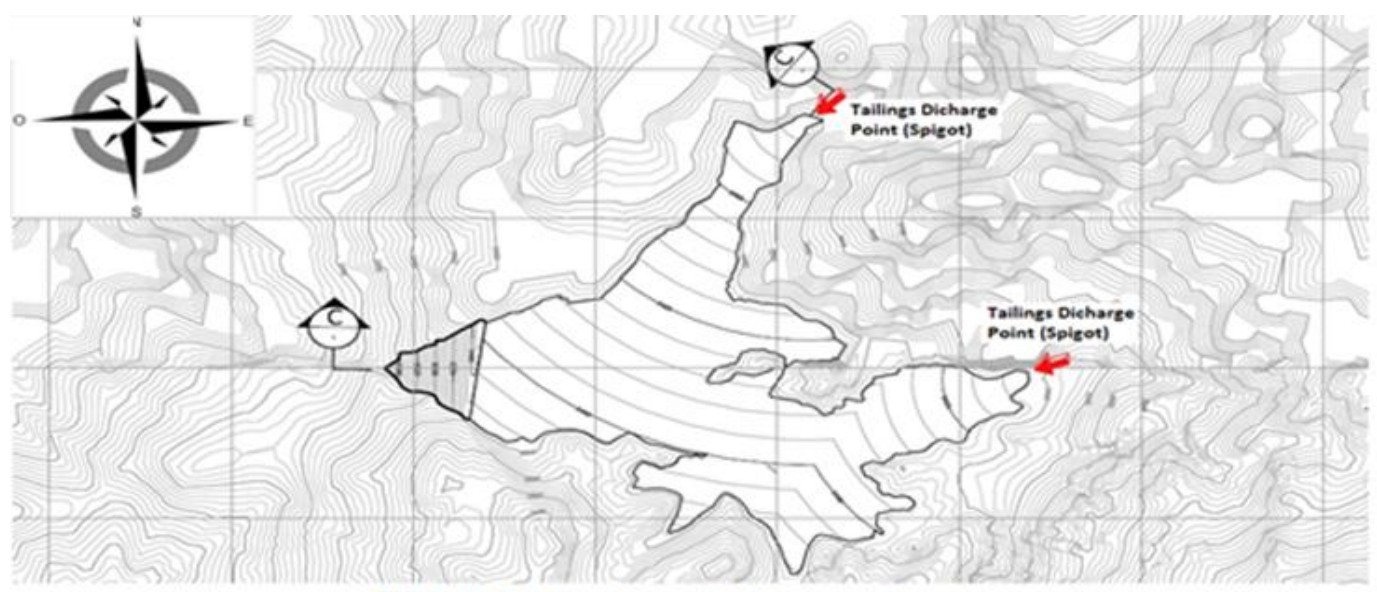

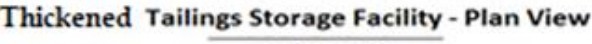

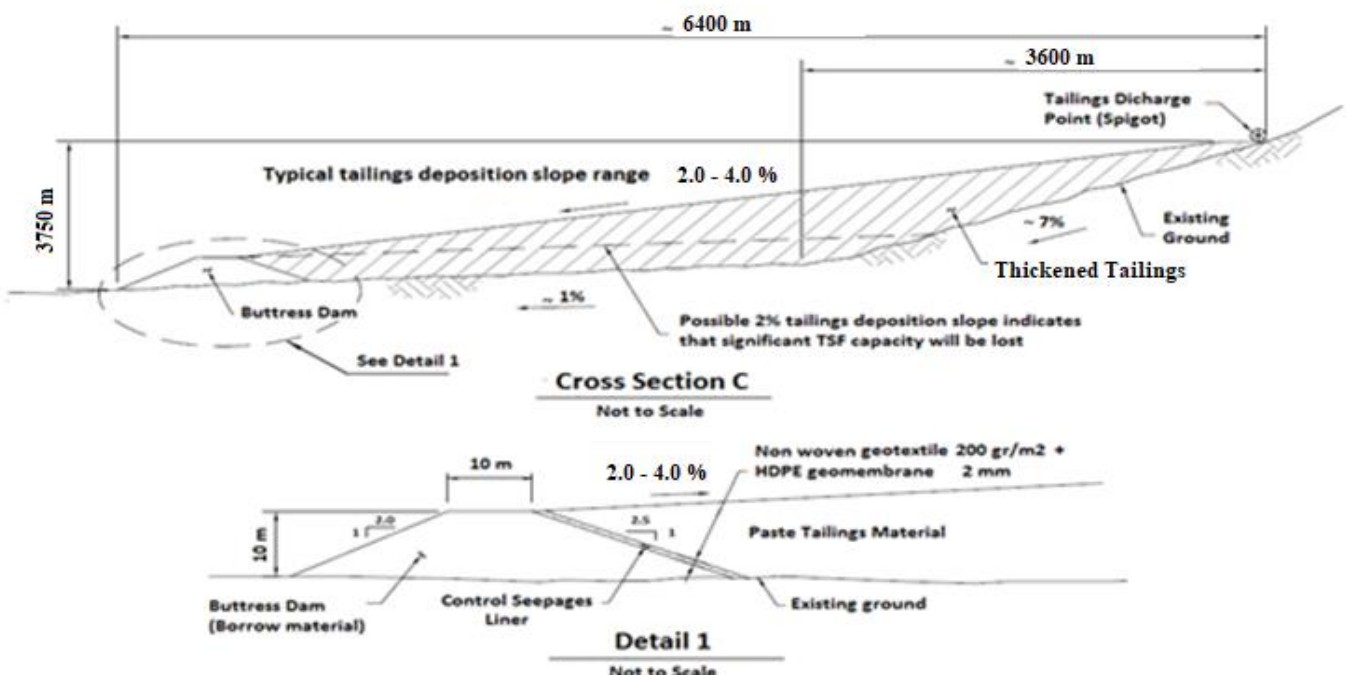

**Figure 11.** Typical Thickened Tailings Disposal Method applied in Andean Region—Down Valley Discharge (DVD).

### 5.2. Cell Dyke Disposal (CDD)

Considering the flat topography of the Atacama Desert region of Chile and Peru, it is necessary to build perimetral dykes and dams to contain tailings. Typically, TSFs are constructed as a multiple cell dyke facility, in which the tailings are contained behind a number of dykes, internal berms or embankments. Dykes constructed typically with mine waste rock divided the available area into cells with several spigots (See Figure 12). By switching between these cells and spigots every one to three days, it is possible to place thin layers of fresh thickened or paste tailings that are then left exposed to the environment up to one and a half months. This exposure allows the tailings to dry and form a desiccated crust. Alternating deposition between cells and limiting lift thicknesses promotes drying and densification of the deposited tailings (See Figure 13).

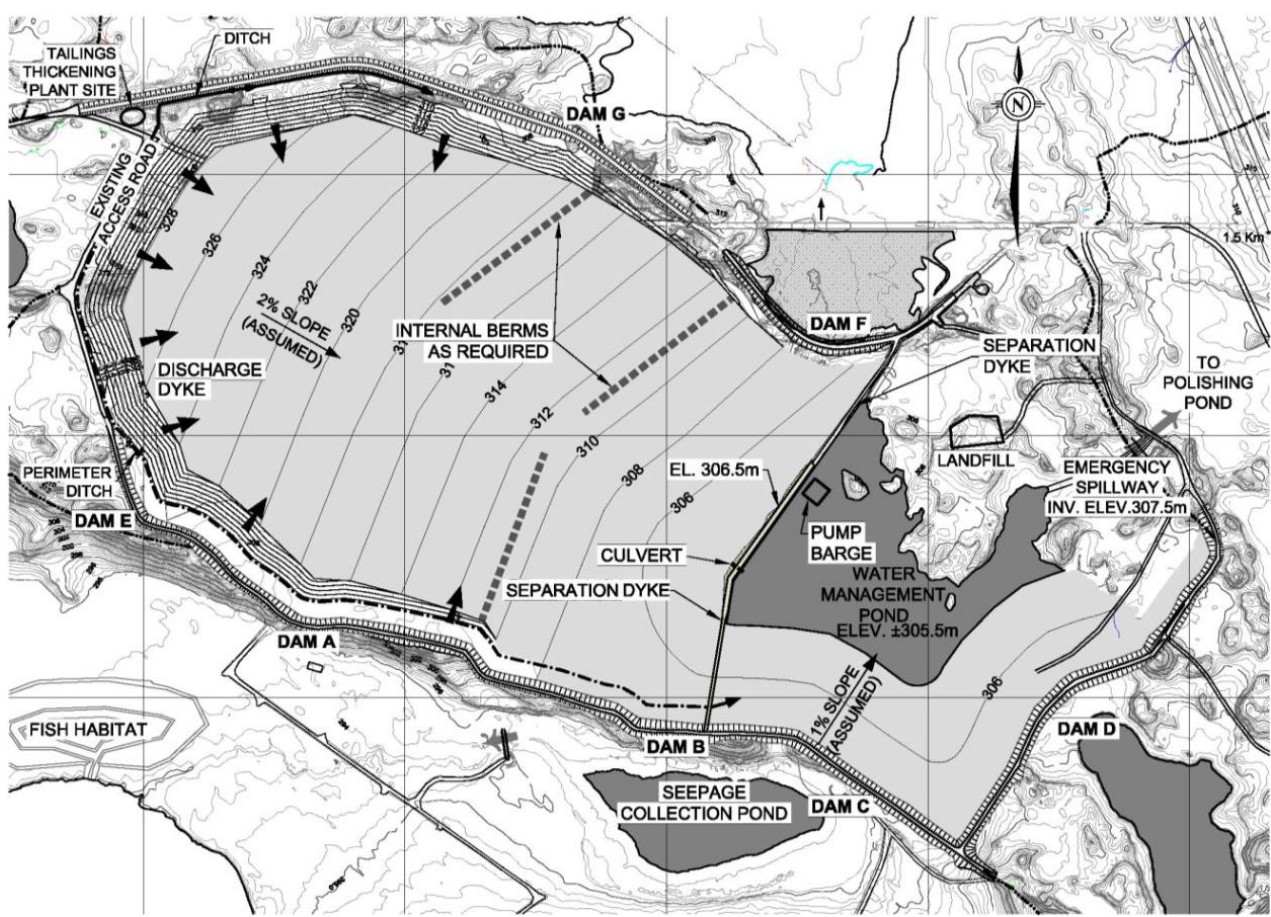

**Figure 12.** Typical Thickened Tailings Disposal Method applied in Atacama Desert—Cell Dyke Disposal (CDD)—Layout View.

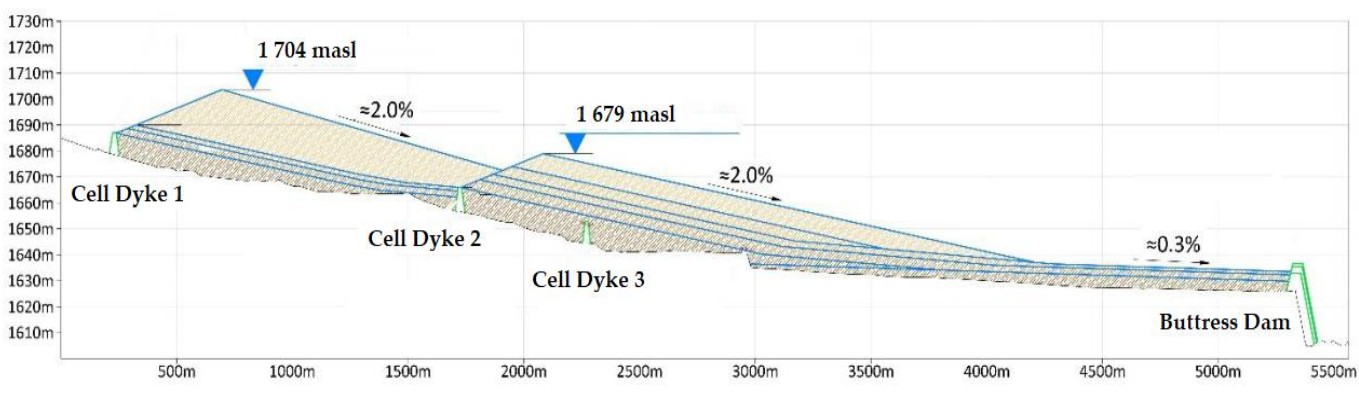

**Figure 13.** Typical Thickened Tailings Disposal Method applied in Atacama Desert–Cell Dyke Disposal (CDD)–Profile View.

Following thickening, pumping, pipeline transport, and beach deposition, the tailings will increase in density by means of different phenomena. First, they settle and consolidate under their own weight. After this initial phase, drying by climate exposure can greatly accelerate densification and strength gain. Afterwards, new layers are deposited over the existing tailings, which consolidates the underlying tailings.

The increase in density and corresponding decrease in void ratio due to desiccation not only reduces the total volume of tailings but also allows for steeper deposition angles and increases the resistance of the TSF to seismic events.

## 6. Thickened Tailings Experiences—State of Practice

In recent years, the improvements in tailings dewatering technologies (thickening and filtering) have allowed an increase in water recovery. These technologies have been successfully applied for production rates up to 100,000 mtpd. There is still a need for more reliable equipment for the thickening processes at large-scale, focused tailings water recovery and reuse in mining processing. Table 3 shows a comparison between different thickened tailings management technologies, considering some mining projects located in wet areas, and dry areas of Chile, Peru and Worldwide.

**Table 3.** Thickened tailings management technologies in Chile, Peru and Worldwide.

| Tailings Storage Facility Name | Mining Company Name | Country | TSF Disposal Parameters | | | | Reference |
|---|---|---|---|---|---|---|---|
| | | | Production Rate (mtpd) | PSD $d_{50}$ (μm) | Solids Content $C_w$ (%) | Tailings Disposal Method | |
| Sarcheshmeh TSF | National Iranian Copper Industries Company | Iran | 96,000 | 57 | 60 (TT) | TTD | [27,28] |
| Kidd Creek TSF | Glencore | Canada | 8000 | 45 | 63 (TT) | TTD | [4] |
| Kimberley TSF | De Beers | South Africa | 25,000 | 45 | 55 (TT) | TTD | [4] |
| Ernest Henry TSF | Evolution | Australia | 20,000 | 55 | 75 (TT) | PTD | [4] |
| Century TSF | New Century Resources | Australia | 12,000 | 60 | 58 (TT) | TTD | [4] |
| Sunrise TSF | BHP | Australia | 10,000 | 50 | 64 (TT) | TTD | [4] |
| Osborne TSF | Ivanhoe Limited's | Australia | 4500 | 55 | 74 (TT) | PTD | [4] |
| Centinela TSF | Antofagasta Minerals | Chile | 95,000 | 45 | 65 (TT) | TTD | [29] |
| Sierra Gorda TSF | KGHM | Chile | 110,000 | 40 | 60 (TT) | TTD | [30] |
| Spence TSF | BHP | Chile | 95,000 | 55 | 52 (TT) | TTD | [31] |
| Talabre TSF | Codelco | Chile | 200,000 | 60 | 57 (TT) | TTD | [32] |
| Talabre TTD TSF (*) | Codelco | Chile | 400,000 | 60 | 67 (TT) | TTD | [32] |
| Los Corralillos TSF | Cerro Negro Norte CMP | Chile | 20,000 | 75 | 65 (TT) | TTD | [33] |
| Carmen Andacollo TSF | Teck | Chile | 55,000 | 70 | 58 (TT) | TTD | [34] |
| Los Diques TSF | Lunding | Chile | 75,000 | 65 | 50 (TT) | TTD | [35] |
| Demo Plant | Collahuasi | Chile | 6000 | 74 | 65 (TT) | PTD | [36] |
| Chinchorro TSF | Las Cenizas | Chile | 2500 | 65 | 70 (TT) | PTD | [36] |
| Delta Plant TSF | ENAMI | Chile | 2000 | 71 | 67 (TT) | PTD | [36] |
| Sector 5 TSF | Coemin | Chile | 8000 | 68 | 70 (TT) | PTD | [37] |
| Alhue TSF | Yamana Gold | Chile | 3000 | 75 | 65 (TT) | PTD | [38] |
| El Toqui TSF | Nyrstar | Chile | 1500 | 68 | 72 (TT) | PTD | [39] |
| Toromocho TSF | Chinalco | Peru | 140,000 | 69 | 65 (TT) | TTD | [40] |
| Antapaccay TSF | Tintaya | Peru | 75,000 | 75 | 58 (TT) | TTD | [41] |
| Las Bambas TSF | MMG | Peru | 140,000 | 75 | 62 (TT) | TTD | [42] |
| Constancia TSF | Hudbay | Peru | 90,000 | 70 | 58 (TT) | TTD | [43] |
| Chungar TSF | Volcan | Peru | 5500 | 60 | 70 (SL) | PTD | [44] |
| Cobriza TSF | Doe Run | Peru | 5000 | 65 | 70 (TT) | PTD | [45] |
| Rumichaca TSF | Volcan | Peru | 6000 | 60 | 65 (SL) | PTD | [46] |
| Huachuacaja TSF | El Brocal | Peru | 18,000 | 65 | 65 (TT) | TTD | [47] |
| Cerro Corona TSF | GoldFields | Peru | 22,000 | 78 | 55 (TT) | TTD | [48] |
| La Quinua TSF | Newmont | Peru | 17,000 | 67 | 67 (TT) | TTD | [49] |

**Note:** The following terms mean: TT: Total Tailings, SL: Slimes (fine particle size distribution of total tailings), PTD: Paste Tailings Disposal, TTD: Thickened Tailings Disposal and (*) Feasibility Project under Study.

Observing Table 3 it is possible to see projects with thickened tailings technology in Canada, Australia, South Africa, Iran, Chile and Peru. It is important to note that the largest number of projects with thickened and paste tailings technology are found in Chile and Peru, with 14 cases and 10 cases, respectively. It is also observed that the cases of Canada, Australia and South Africa have tailings productions of a maximum of 25,000 mtpd, while in the cases of Iran, Chile, and Peru there are projects with productions of 100,000 mtpd, even in some cases such as Chile with productions of the order of 200,000 mtpd. With respect to the contents of solids by weight (Cw) reached, in some cases there are low values of the order of 55% and at most there are values of the order of 70% to 75%.

It is important to mention that the Talabre TTD TSF project with tailings production of 400,000 mtpd with Thickened Tailings Disposal (TTD) technology is in a phase of technical studies, awaiting approval of environmental permits and waiting to be added to the production of tailings from the Ministro Hales and Radomiro Tomic concentrator plants, to the current tailings production of 200,000 mtpd from the Chuquicamata concentrator plant.

Finally, with respect to the tailings disposal method, there is an equitable relationship in all cases considering Thickened Tailings Disposal (TTD) and Paste Tailings Disposal (PTD). These technologies have been successfully applied in Chile and Peru for production rates up to 100,000 mtpd; showing good performance improvements on large-scale projects with high ore production rates. In this scenario, there is still a need for more reliable equipment for paste tailings thickening plants on large scale, focusing in the tailings water recovery enhancing for its reuse in mining processing.

## 7. Successful Cases in Chile and Peru

The development of thickening technologies has changed the criteria's used in evaluating the benefits of increased tailings density. Nowadays, considering more stringent regulatory framework, more concentrator plants apply tailings thickening technologies together with optimized tailings disposal schemes to recover water, minimize TSF footprints and comply with regulations. The following paragraphs present various successful cases in South America specifically in mining projects of Chile and Peru.

### 7.1. Demo Plant Paste Tailings—Down Valley Discharge—Collahuasi—Chile

The Compañía Minera Doña Inés de Collahuasi owned a copper mine (Collahuasi), situated in northern Chile, about 200 km southeast of Iquique, at an altitude of 4400 masl in the Andes Mountains. The climate is typical of this dry region, where annual precipitation is approximately 200 mm with evaporation rates over 2000 mm year. One of the major needs for the mine project is water and thus significant resources have been focused on water recovery from tailings. For this reason, Collahuasi carried out a pilot test and Demo Plant program to enable adequate tailings thickening to maximize water recovery (2008–2010). Table 4 shows Demo Plant performance:

**Table 4.** Demo Plant—Thickening Parameters and Tailings Characterization [50].

| Paste Thickening Process Parameters | | | Cu Tailings Characterization | | |
|---|---|---|---|---|---|
| Parameter | Value | Units | Parameter | Value | Units |
| Tailings throughput | 6000 | tpd | Particle size distribution (P80) | 115 | μm |
| DCT diameter | 22 | m | Fines content (<#200 ASTM) | 65 | % |
| DCT height | 20 | m | Solid Gravity (Gs) | 2.76 | - |
| Solid loading | 0.658 | tph/m$^2$ | Feed solid content (Cw) | 47–53 | % |
| Flocculent dosing use | 20–25 | g/ton | Underflow Solid Content (Cw) | 57–67 | % |
| Overflow water recovery | 127 | m$^3$/h | Unsheared/sheared yield stress | 260/130 | Pa |

The Demo Plant was located on the side of TSF called Pampa Pabellón (Figure 14), below the tailings distribution tank (4275 masl) and parallel to the tailings discharge drop boxes system. The tailings are transported to the demo plant from the concentrator plant, tailings being pumped from the first drop box located after the distributor tank. The demo plant has a 22 m in diameter, 18 m high DCT, equipped with a shear thinning system at the DCT underflow. Using centrifugal pumps, it produces shear rupture of the flocs formed in the thickening and then recirculates portion of this material to the DCT cone by a loop. From this cone, paste tailings are extracted by another centrifugal pump for transport to a network of discharge spigot at elevation 4330 masl.

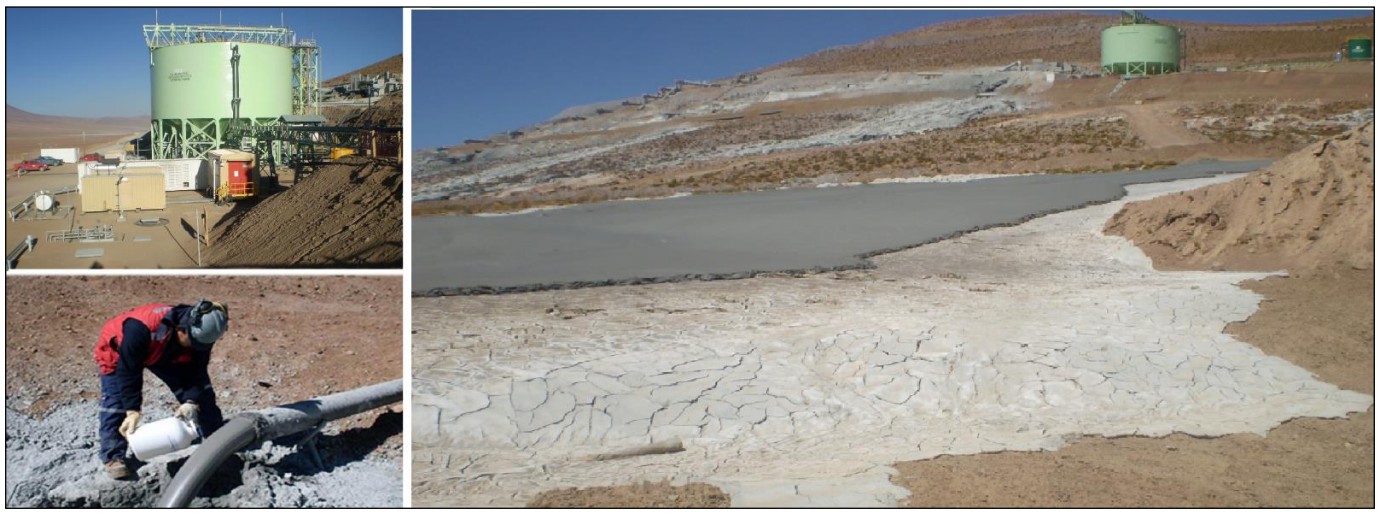

**Figure 14.** Demo Plant overview and paste tailings disposal at TSF [36,50].

*7.2. Las Cenizas Paste Tailings—Down Valley Discharge—Cabildo—Chile*

Las Cenizas Mine is an underground copper mine operation located in the Cabildo valley in the central region of Chile, approximately 120 km north of Santiago city. The climate is typical of a Mediterranean region, where annual precipitation is approximately 200 mm with evaporation rates over 1750 mm per year. Tailings generated in the process plant are discharged to the surface as paste tailings (40% of the time) and are used as paste fill (60% of the time) for underground mining works. Tailings are produced from the process plant with 30% solids and are transported to a 16 diameter HRT Thickener. Underflow tailings from the HRT reach 58–62% solids content and are pumped by PD pump to the paste tailings plant or underground mine such as a backfill.

A 17 m DCT is located in the Chinchorro valley, about 4 km from the HRT thickener (Figure 15), where tailings are then transported for surface tailings disposal. One of the main requirements for the mine project is the reclamation of water and maintaining a good relationship with the community, because there is a severe drought in the area and the community demands the least possible impact from tailings. For this reason, Las Cenizas Mine carried out pilot tests to enable adequate tailings thickening to maximize water recovery (2002). Table 5 shows paste tailings thickening plant performance.

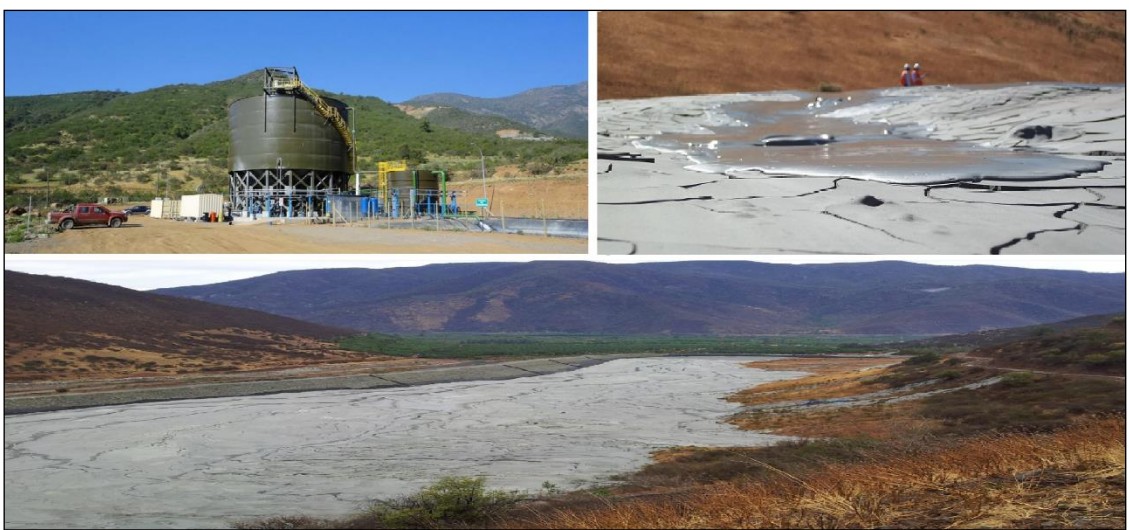

**Figure 15.** Las Cenizas DCT overview and paste tailings disposal at TSF [36,51].

**Table 5.** Las Cenizas Plant—Thickening Parameters and Tailings Characterization [51].

| Paste Thickening Process Parameters | | | Cu Tailings Characterization | | |
|---|---|---|---|---|---|
| **Parameter** | **Value** | **Units** | **Parameter** | **Value** | **Units** |
| Tailings throughput | 2500 | tpd | Particle size distribution (P80) | 110 | μm |
| DCT diameter | 17 | m | Fines content (<#200 ASTM) | 60 | % |
| DCT height | 16 | m | Solid Gravity (Gs) | 2.82 | - |
| Solid loading | 0.460 | tph/m$^2$ | Feed solid content (Cw) | 55–60 | % |
| Flocculent dosing use | 25–30 | g/ton | Underflow Solid Content (Cw) | 68–72 | % |
| Overflow water recovery | 45 | m$^3$/h | Unsheared/sheared yield stress | 120/60 | Pa |

The DCT was commissioned in 2011; the design underflow solids content is 65%, a lower value than might be expected but a reflection of the clayey fines component in the tailings. The estimated slope is between 1.5–2.0%. This is satisfactory at this operational stage, but further improvement in underflow density and beach slope is expected. The main operational challenges comprise flow fluctuation and properties of tailings, control of discharge, measurements and automation control.

*7.3. Delta Paste Tailings—Down Valley Discharge—Ovalle—Chile*

ENAMI (Empresa Nacional de Minería) have a copper processing plant called Delta which is located in Ovalle valley in the northern region of Chile, approximately 330 km north of Santiago city, a mining area with the highest density of copper extraction activities. The climate is typical of an arid region, where annual precipitation is approximately 125 mm with evaporation rates over 2050 mm per year. Tailings are produced from the process plant with 35% solids and are transported by centrifugal pumps to a 12 m diameter DCT Thickener. Underflow tailings from the DCT reach a paste tailings consistency and are then pumped by centrifugal pump to the TSF. The underflow tailings solid content ranged between 55–65% soon after commissioning in 2011. More recently (2015) solid contents of 65–70% were measured (See Table 6).

**Table 6.** Delta Plant—Thickening Process and Tailings Characterization [5].

| Paste Thickening Process Parameters | | | Cu Tailings Characterization | | |
|---|---|---|---|---|---|
| **Parameter** | **Value** | **Units** | **Parameter** | **Value** | **Units** |
| Tailings throughput | 2000 | tpd | Particle size distribution (P80) | 105 | μm |
| DCT diameter | 12 | m | Fines content (<#200 ASTM) | 70 | % |
| DCT height | 8 | m | Solid Gravity (Gs) | 2.90 | - |
| Solid loading | 0.737 | tph/m$^2$ | Feed solid content (Cw) | 30–35 | % |
| Flocculent dosing use | 15–20 | g/ton | Underflow Solid Content (Cw) | 65–70 | % |
| Overflow water recovery | 114 | m$^3$/h | Unsheared/sheared yield stress | 90/45 | Pa |

Despite the start-up and early operating difficulties that occurred, the DCT, when judged by the general appearance and behavior of the underflow streams and supported by on-site yield stress tests, appears to be performing remarkably well. The beach slope is flatter than the design value, which can be explained by the low percent solids and the fine particle size distribution of the current tailings. The slope can be expected to steepen as operating difficulties are sorted out (See Figure 16).

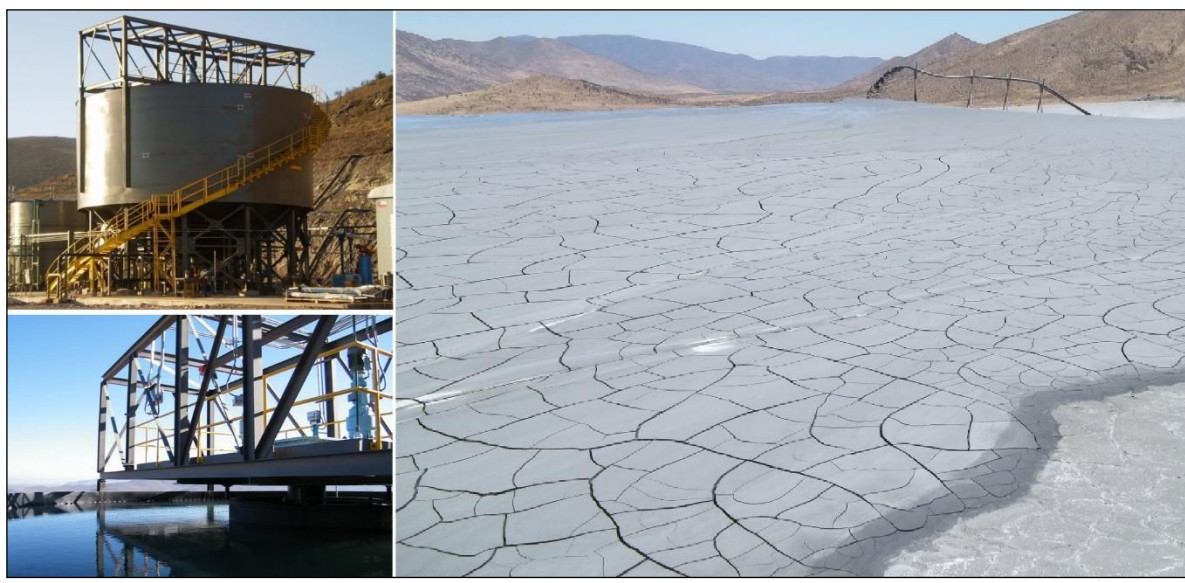

**Figure 16.** Delta paste tailings thickening plant and paste tailings disposal [5,36].

### 7.4. El Toqui Paste Tailings—Down Valley Discharge—Coyhaique—Chile

The El Toqui mine is located in Chile's Region XI 1350 km south of Santiago city and 120 km northeast of Coyhaique, in a region with a well-known history of poly-metallic mineralization. The mine plant processes mainly gold, zinc and silver and produces approximately 1500 tons of tailings per day. The climate is typical of the Patagonia region, where annual precipitation is approximately 1200 mm with evaporation rates over 1000 mm per year. Tailings generated in the process plant are discharged to the surface as paste tailings (50% of the time) and are used as paste fill (50% of the time) in the underground mining works. Tailings are produced from the process plant with 30% solids and transported to a 14 m diameter DCT Thickener. Underflow tailings from the DCT reach 70–75% solids content and are pumped by a PD pump to the surface disposal or underground mine. Table 7 shows paste tailings thickening plant performance.

**Table 7.** El Toqui Plant Thickening Process and Tailings Characterization [39,52].

| Paste Thickening Process Parameters | | | Zn, Au, Pb, Ag and Cu Tailings Characterization | | |
|---|---|---|---|---|---|
| Parameter | Value | Units | Parameter | Value | Units |
| Tailings throughput | 1500 | tpd | Particle size distribution (P80) | 160 | μm |
| DCT diameter | 14 | m | Fines content (<#200 ASTM) | 60 | % |
| DCT height | 10 | m | Solid Gravity (Gs) | 3.20 | - |
| Solid loading | 0.406 | tph/m$^2$ | Feed solid content (Cw) | 30–35 | % |
| Flocculent dosing use | 25–35 | g/ton | Underflow Solid Content (Cw) | 70–75 | % |
| Overflow water recovery | 103 | m$^3$/h | Unsheared/sheared yield stress | 150/75 | Pa |

El Toqui operated surface disposal with filtered tailings for some years. However, problems with management of tailings were experienced due to frequent precipitation, because the moisture content of filtered tailings exceeded 20%. For this reason, tailings are actually disposed on surface such as paste (See Figure 17) [53].

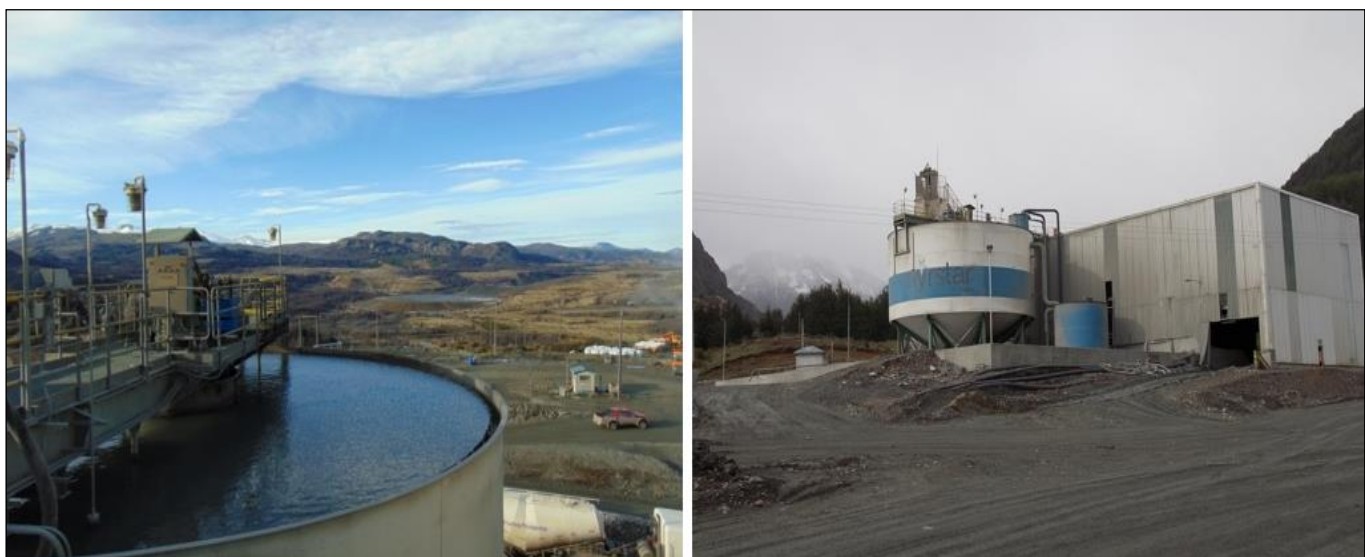

**Figure 17.** El Toqui paste tailings thickening plant [39,52].

*7.5. Alhué Paste Tailings—Down Valley Discharge—Yamana Gold—Chile*

Alhué mine is an underground gold-silver mine located 180 km south of Santiago city in central Chile, situated in an area with moderate to rugged topographic relief characterized by narrow valleys and high hills. Mining began in 1986 at a processing rate of 300 tons per day, producing a gold, silver and zinc concentrate for off-site treatment. The plant actually treats 6000 tons per day of ore, incorporating flotation, leaching and electrowinning processes for onsite doré metal production and generation of a zinc concentrate for off-site treatment. Retreatment of historic tailings started in 2013 and is expected to increase annual production for five years [38].

Alhué needed to ensure operational continuity, because the current TSF is near the end of its useful life, requiring construction of a new TSF. For the disposal of tailings resulting from the process of current operations, the design of new TSF includes the use of modern practices with paste tailings disposal. Alhué mine has chosen to improve environmental standards and increase recovery of tailings water for reuse in the process. For this reason, Alhué mine carried out a pilot test program to enable adequate tailings thickening to maximize water recovery and the tailings disposal slope (2013). Representative tailings samples were thickened in a 1.5 m diameter deep cone paste pilot plant thickener using freshly prepared flocculent at the correct dosage rates. The thickener pilot plant and test campaign used a combination of pilot plant data and channel scale tests in order to determine the right parameters for the design of the paste thickener and TSF [54].

A deposition beach slope of paste tailings of 3.0% for Alhué was estimated from the pilot plant campaign and channel tests. Pilot plants have become a practical tool and are ever more used to support engineering design decisions. Considering the good performance of the paste pilot plant Alhué mine decided to approve paste tailings disposal. Tailings are produced from the process plant with 30% solids and are transported to a 17 m DCT. Underflow tailings from the DCT reach 60–70% solids content and paste is then pumped by PD pumps to the TSF (See Figure 18) [37].

The TSF facility and DCT were constructed in 2015. The DCT plant was commissioned at the end of 2015. Although the design underflow density of tailings was 65% solids, a lower value was obtained, which is a reflection of the clayey fines component in the tailings. The estimated beach slope was between 1.5–2.0%. This is satisfactory at this operational stage (startup stage), but further improvement in underflow density and beach slope is expected. Table 8 shows paste tailings thickening plant performance.

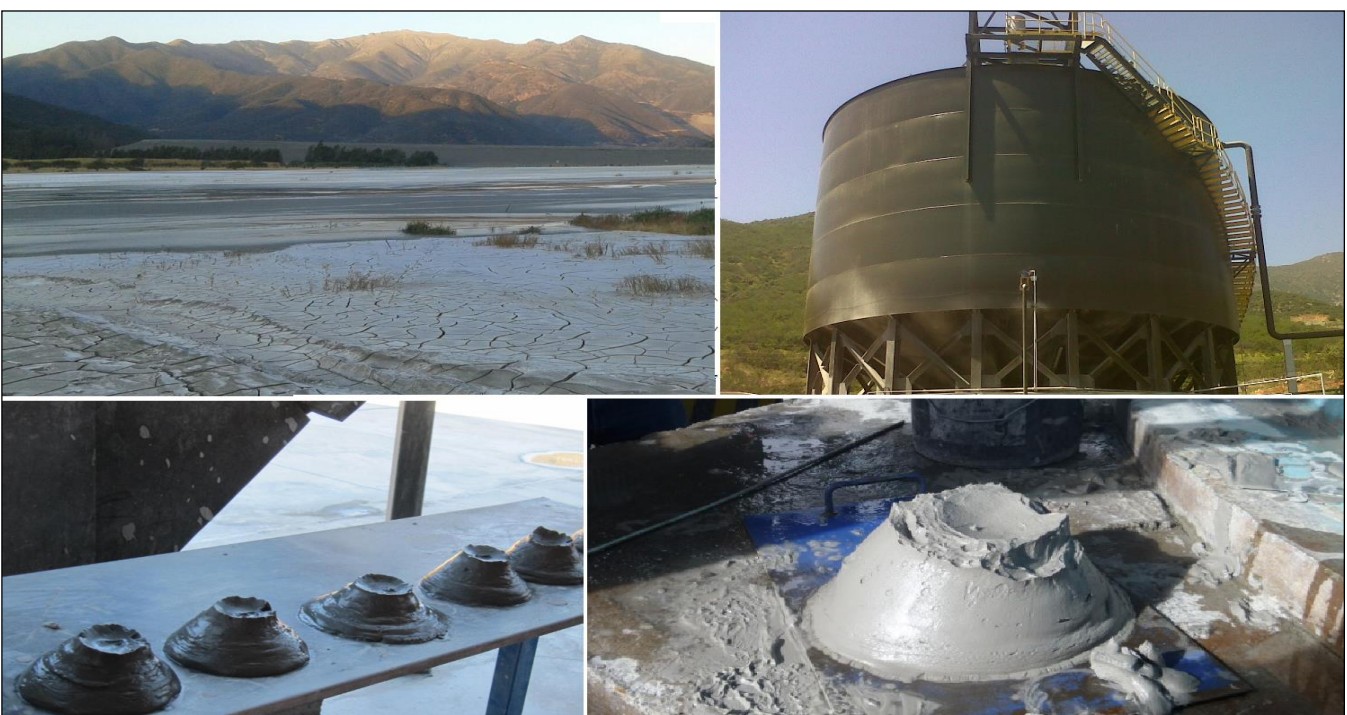

**Figure 18.** Alhué paste tailings thickening plant and paste TSF [38].

**Table 8.** Alhué Plant—Thickening Process Parameters and Tailings Characterization [38].

| Paste Thickening Process Parameters | | | Au and Zn Tailings Characterization | | |
|---|---|---|---|---|---|
| Parameter | Value | Units | Parameter | Value | Units |
| Tailings throughput | 3000 | tpd | Particle size distribution (P80) | 125 | μm |
| DCT diameter | 17 | m | Fines content (<#200 ASTM) | 55 | % |
| DCT height | 12 | m | Solid Gravity (Gs) | 2.72 | - |
| Solid loading | 0.550 | tph/m$^2$ | Feed solid content (Cw) | 20–25 | % |
| Flocculent dosing use | 25–35 | g/ton | Underflow Solid Content (Cw) | 60–70 | % |
| Overflow water recovery | 320 | m$^3$/h | Unsheared/sheared yield stress | 130/65 | Pa |

*7.6. Centinela Thickened Tailings—Cell Dyke Disposal—Antofagasta Minerals—Chile*

Minera Centinela was established in 2014 from the merger of the Esperanza and El Tesoro mining companies. It is located in the Antofagasta region of Chile, 1350 km north of Santiago, in an important mining area with sulfide and oxide deposits. Centinela produces copper concentrates through a grinding and flotation process in the sulfide line, obtaining mining tailings; and copper cathodes using a solvent extraction and electrowinning (SX-EW) process on the oxide line.

Currently, Minera Centinela represents 40% of the production of Grupo Minero Antofagasta Minerals, with a mineral production of around 100,000 mtpd. In the coming years, the development of new operations that will turn Minera Centinela into a large mining district is expected, including the exploitation of the Encuentro and Esperanza Sur deposits. This development will allow the company to extend the useful life of its extractive operations by 25 years, with a production of close to 400 thousand tons of fine copper per year [55].

The thickened tailings deposit called Centinela TSF stores the mining tailings produced by Minera Centinela, and is located in the commune of Sierra Gorda, Antofagasta Region, at an approximate elevation of 2100 masl. The tailings are thickened to reach a concentration of solids by weight (Cw) in the order of 65% through 3 high density thickeners (HDT) of 60 m diameter each one and 3 high compression thickeners (HCT) of 45 m diameter each one (Figure 19) [56].

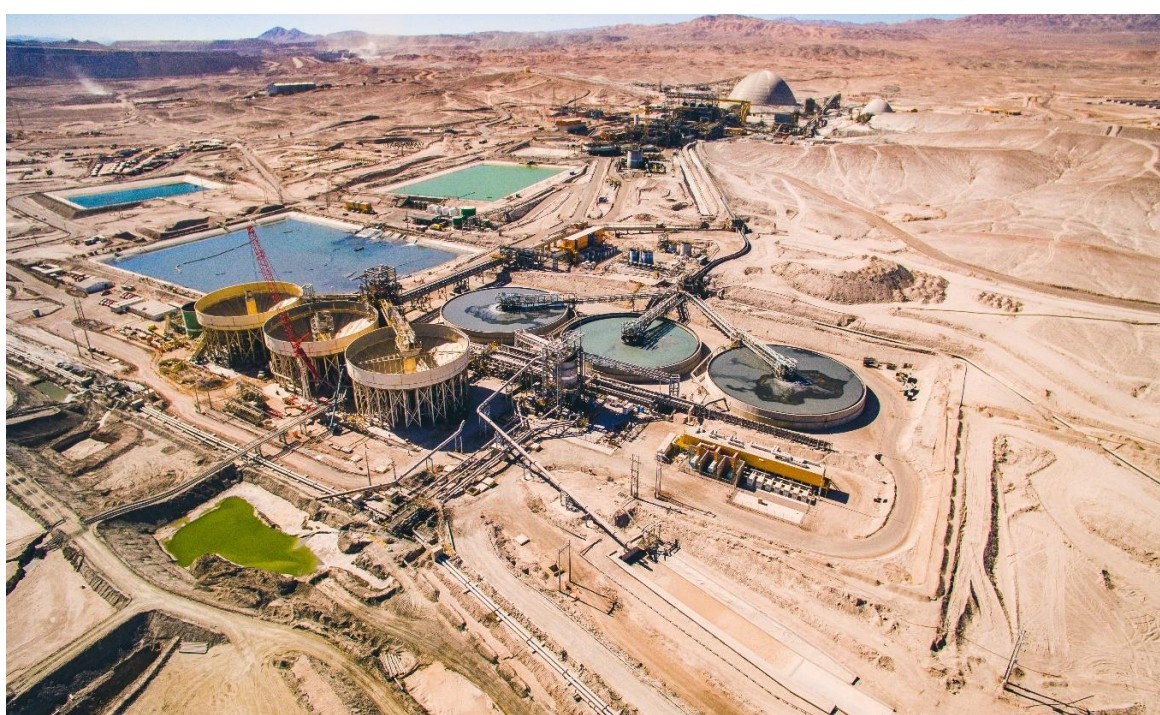

**Figure 19.** Thickening Process Plant in Centinela Mining Project.

The thickened tailings are pumped by centrifugal pumps into a series of cells, where the tailings are contained by a series of small dams or dykes. The tailings pumping system is made up of two trains of three centrifugal pumps, each pumping tailings from a tank (Hammer Tank), which receives the tailings from the thickeners (HDT and HCT), through HDPE pipes inside of the deposit [29,57].

The deposition system considers the deposition of tailings in 5 sectors or fields or cells, which correspond to the High sector, wall J2, wall J1, Main wall, and Caracoles. As mentioned above, the Laguna Seca sector corresponds to the low-concentration tailings dam (low Cw). In Figures 20 and 21 the deposit sectors are presented and then the description of the discharges by the different sectors is presented [29,57].

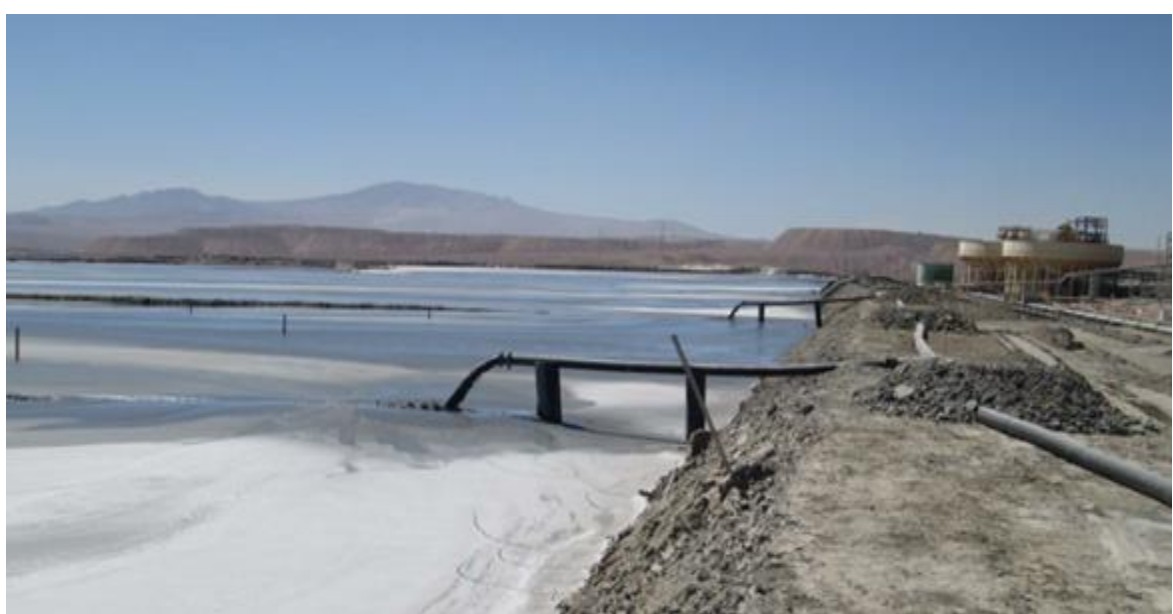

**Figure 20.** Thickened Tailings Discharge in Spigots in Centinela Mining Project.

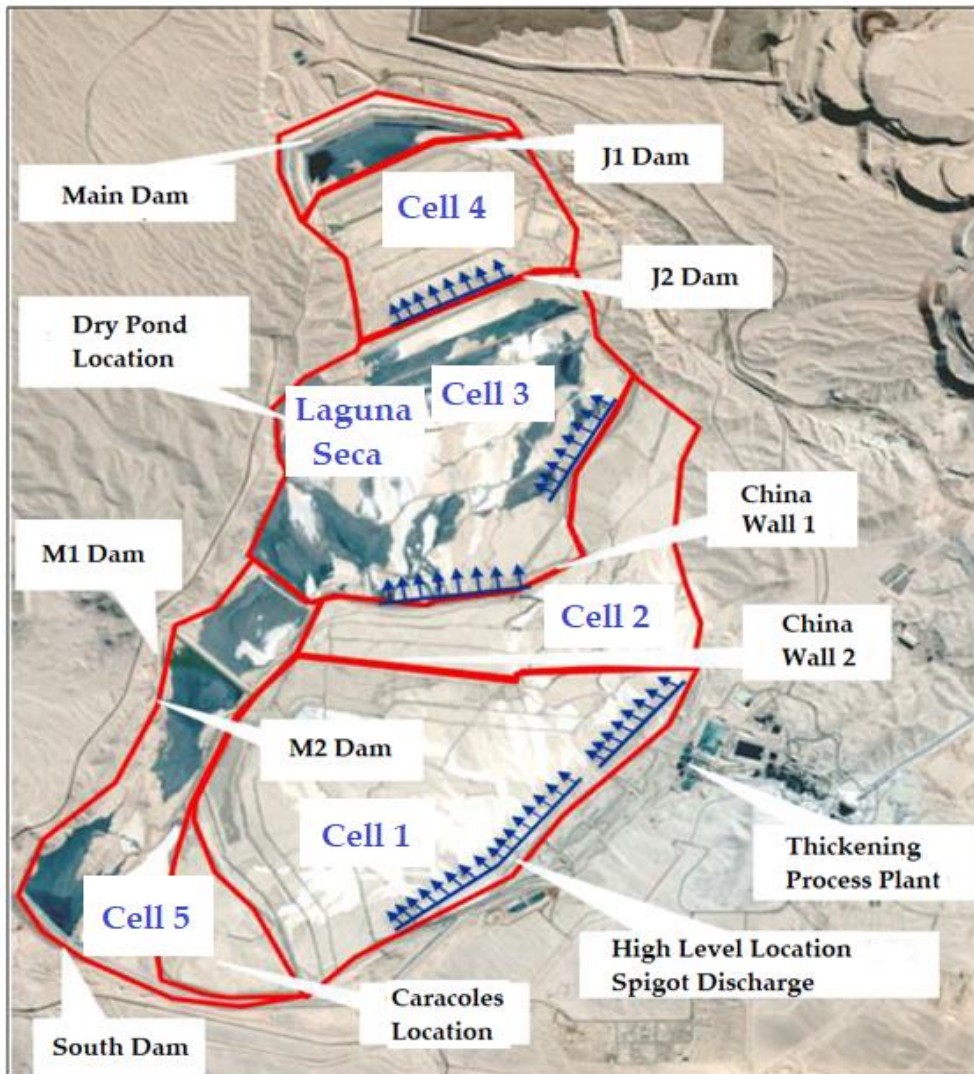

**Figure 21.** Layout View Centinela Thickened Tailings Storage Facility.

The following paragraph indicates the operation of the tailings deposit scheme in the thickened tailings deposit Centinela:

- Cells 1 and 2: Discharges from dykes to the upper sector of the TSF through spigots.
- Cell 3: Discharge from Chinese Wall 1 towards the J2 dyke basin through spigots.
- Cell 4: Discharge through tailings pipes inside the deposit towards the J1 dyke basin. The J1 dyke basin is subdivided into five sub-deposits by means of 4 dykes built inside it, which allow the deposition to be alternated. Work is constantly being carried out on installing spigots from the J2 dyke crest.
- Cell 5: Discharge through a tailings channel through the Caracoles sector to the South dike.

*7.7. Sierra Gorda Thickened Tailings—Cell Dyke Disposal—KGHM—Chile*

Sierra Gorda Sociedad Contractual Minera (SGCM) is a copper, gold and molybdenum mine operation located approximately 60 km from the city of Calama in Northern Chile (II Antofagasta Region). The mine has been operating since 2014, with proven and probable ore reserves of 800 million tons containing grades of 0.35% copper, 0.04 g/t gold and 0.01% molybdenum. Current operations are 110,000 mtpd (increasing to 230,000 mtpd in Phase II) with the produced copper and molybdenum concentrate being sent by train to Sierra Gorda SGCM's port facilities located in Antofagasta. Sierra Gorda SGCM operates with the use of sea water from the cooling systems of a power plant in the town of Mejillones. Instead of being recycled to the sea, the water is pumped to the processing plant through a

144 km pipeline located at an elevation of 1700 masl and storage in a lined pond (Figure 22). The mine contemplates a useful life of 20 years [30].

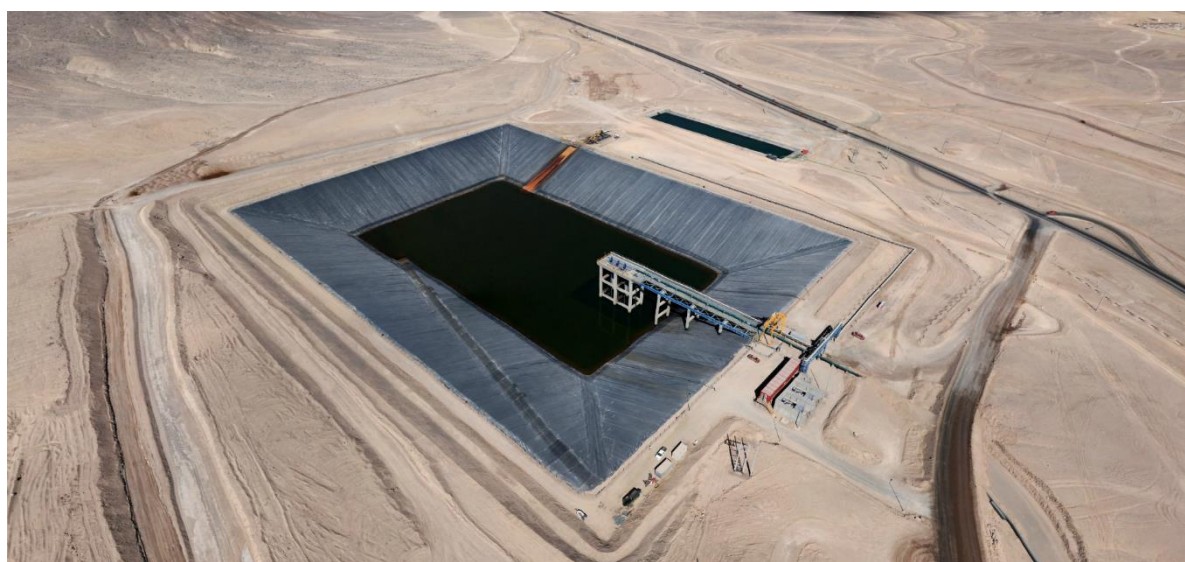

**Figure 22.** Sea water reservoir in Sierra Gorda Mining Project.

As part of the design of Sierra Gorda SGCM, thickened tailings disposal (TTD) was considered. The original tailings storage facility (TSF) design considered discharge of tailings at 62% solids (Cw), having an average achievable beach slope of 1.0% and a storage capacity of 1350 million tons of tailings over the 20-year life of the operation. During commissioning of the processing plant in 2014, the design solids concentration of the tailings was not consistently achieved by the three 86 m diameter high-rate tailings thickeners (Figure 23). The principal reasons relate to the variations in particle size distribution of the current tailings, which are slightly finer, and the torque limitations of the thickeners. Since 2014 to date, tailings are discharged at 58–62% solids (Cw) with beach slopes of 0.5% being achieved (via single point discharge) [30].

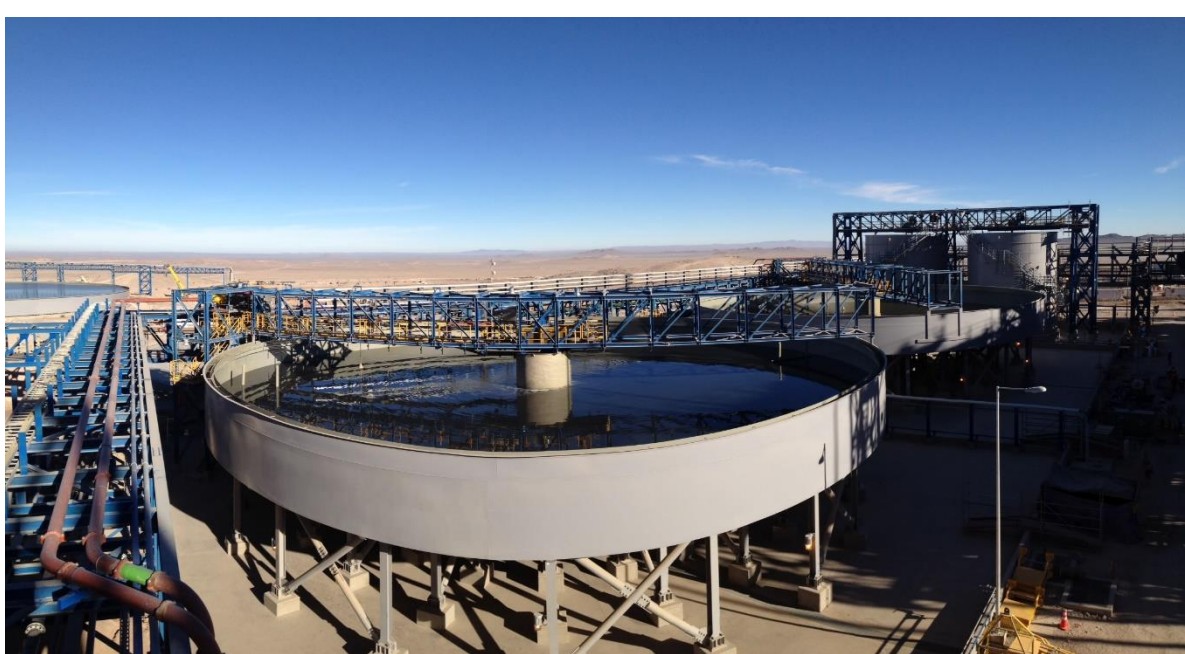

**Figure 23.** Thickening Process Plant in Sierra Gorda Mining Project.

Until February 2017, the TSF was designed to operate by depositing tailings sub-aerially via single point discharge from predominately near the center of the 2300 Ha TSF via a 1.2 m diameter, 2.4 km long pipeline from the processing plant. A second single point discharge was used to an emergency area nearer to the processing plant. The design of the tailings transport system considers gravitational discharge from the main underflow starter box adjacent to the three tailings thickeners. The only thickened tailings pumps considered in the design are located in the underflow of the tailings' thickeners, reporting to the starter box initiating 100% gravitational transport of tailings to the TSF. Operation of the single point discharge system has resulted in the tailings flowing as single channels downstream to the main embankments of the TSF without developing a beach over the natural ground. The reasons primarily being that the tailings formed a flatter beach than expected, less than the natural ground slope [30].

The goal of Sierra Gorda SGCM's tailings management strategy is to not only disposal the tailings via the spigot distribution system, but also to optimize the in situ density and capacity of the TSF reducing the elevations of the large perimeter dykes at final capacity. In the short-term, and due to the reduced freeboard during 2017, no tailings were to be discharged towards the main dykes so as to allow the already deposited tailings to dry and consolidate, as well as allow time to install underdrain extensions and subsequent raising of the main dykes (Figure 24). During this time, temporary internal dykes were constructed to store tailings while the various spigot systems were commissioned [30].

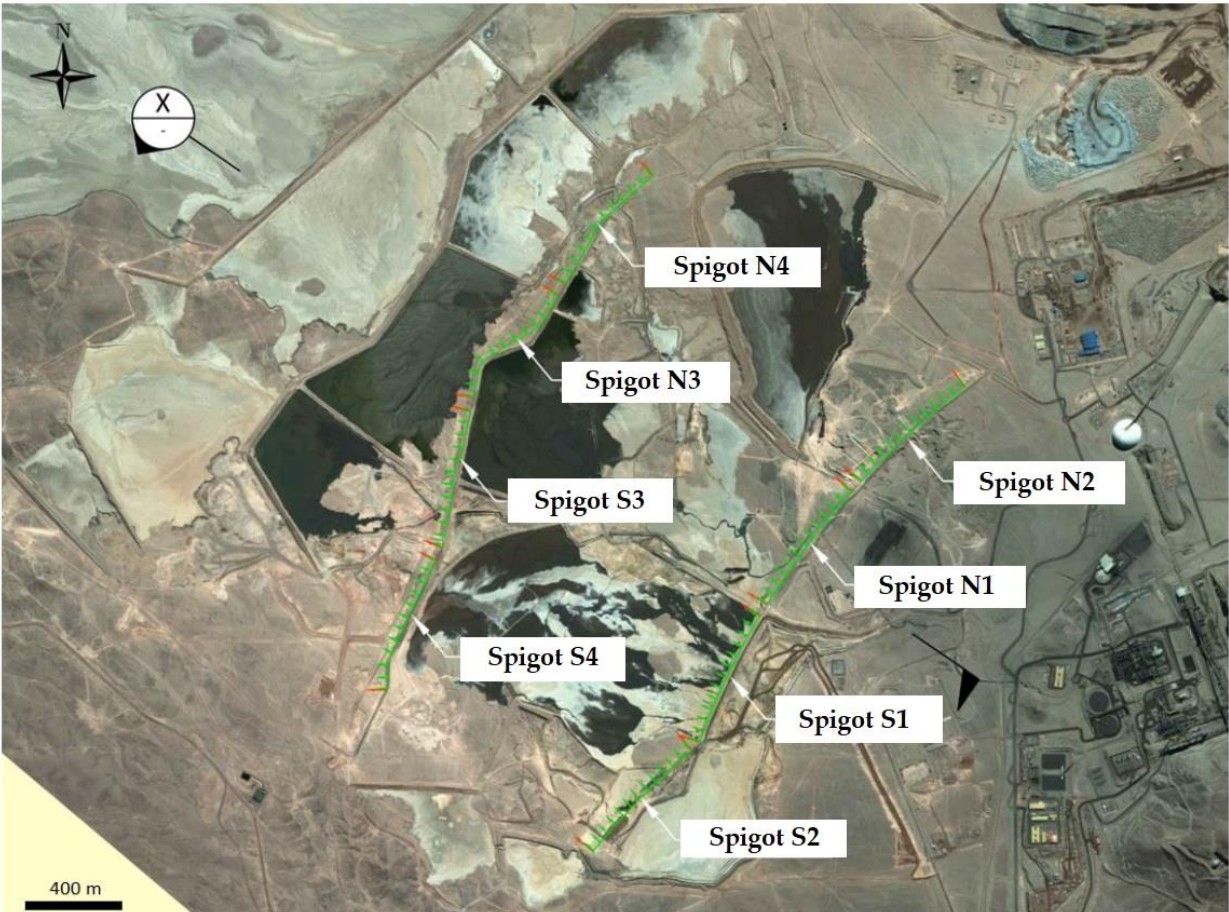

**Figure 24.** Layout View Sierra Gorda Thickened Tailings Storage Facility.

These internal dykes are considered temporary in that they will eventually be covered by tailings discharged from the spigots. Interwall drop pipes were considered to transfer flow from one side of a dyke to the other lower side thereby enabling controlled filling of the TSF as the beach from the spigots developed [30,58].

### 7.8. Spence Thickened Tailings—Cell Dyke Disposal—BHP—Chile

Spence is a mining project located in the region of Antofagasta, commune of Sierra Gorda, close to the Centinela and Sierra Gorda mining projects. Spence is a mining project operated by the mining company BHP Billiton. The Spence mining deposit began mining operations exploiting oxide mining resources through hydrometallurgical processes in leaching heaps, today it has begun to exploit sulfide mining resources, where a mineral processing concentrator plant with a capacity of 95,000 mtpd has been designed and built (Figure 25). The concentrator plant has the processes of crushing, grinding, flotation, thickening of tailings and concentrates, and filtering of concentrates. This metallurgical plant mainly recovers copper and molybdenum concentrates. The Spence project also includes the use of industrial quality desalinated water, for which the construction and operation of a desalination plant is being contemplated with a production capacity of up to 1600 L/s. This plant would be built in two stages: the first one to produce and bring 800 L/s required by the mining project, and a second stage of 800 L/s to supply other BHP Billiton projects.

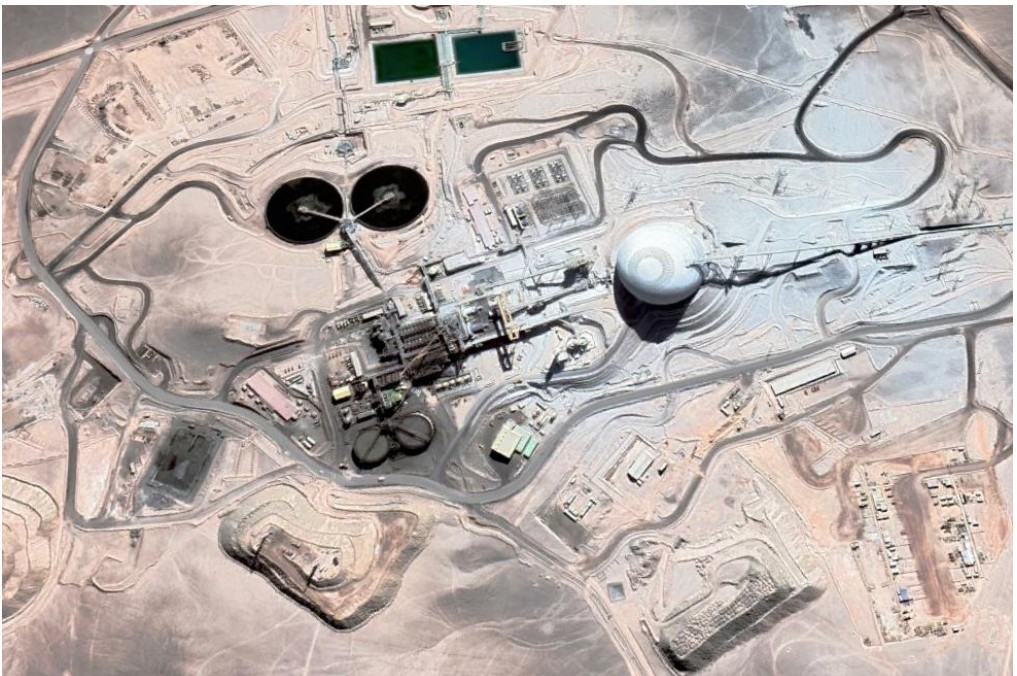

**Figure 25.** Layout View Spence Concentrator Plant.

Mining tailings are generated through the mineral concentration process, which are thickened by 2 high-rate type thickening equipment with a diameter of 100 m. The mining tailings reach a thickening level with a concentration of solids by weight (Cw) of the order of 52%. The thickened tailings are transported in pipelines with the help of centrifugal pumps. The mining tailings transportation system considers a series of spigots or tailings discharge points, with the aim of forming a tailings beach with a deposition slope that allows the consolidation and sedimentation of the tailings over time, locating the supernatant waters of the tailings in the vicinity of the water collection reservoir.

The tailings storage facility is located in the vicinity of the concentrator plant in a relatively flat topography, for which it has been necessary to build dikes with borrow material from the sector and with waste rock material from the mine (Figure 26). The mining tailings are arranged in 3 cells built by perimeter dikes, where the tailings consolidate, settle and solid-liquid separation occurs, where the supernatant water is collected in a water reservoir attached to the tailings deposit cells (Figure 27). The water collected and recovered from the tailings is recirculated through pumps and pipes to the mineral concentration process.

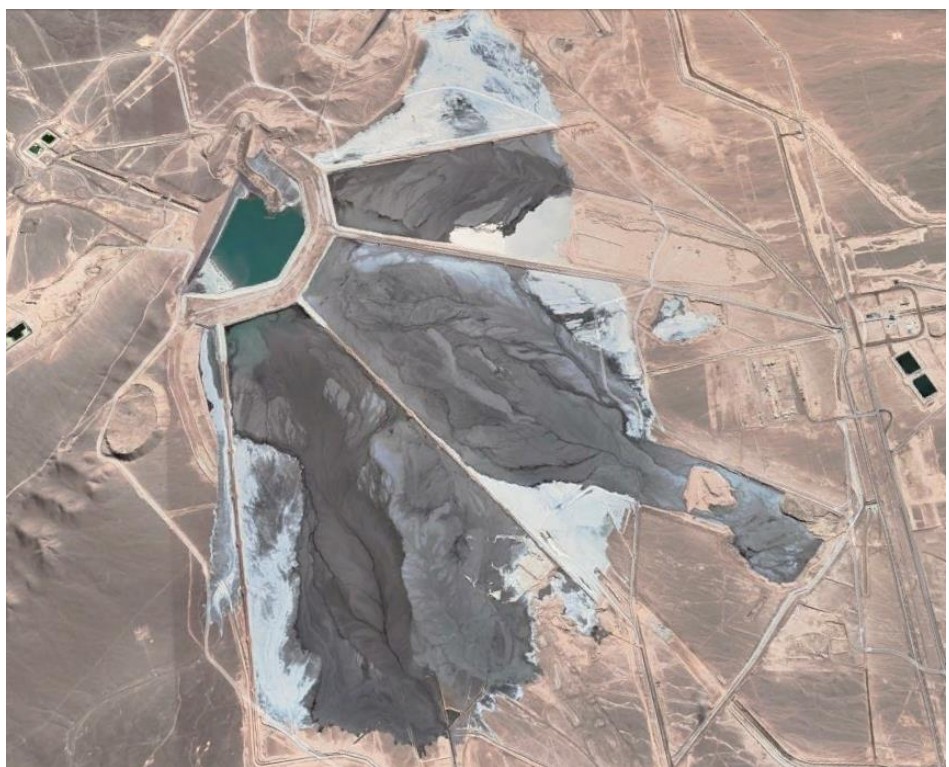

**Figure 26.** Layout View Spence Tailings Storage Facility.

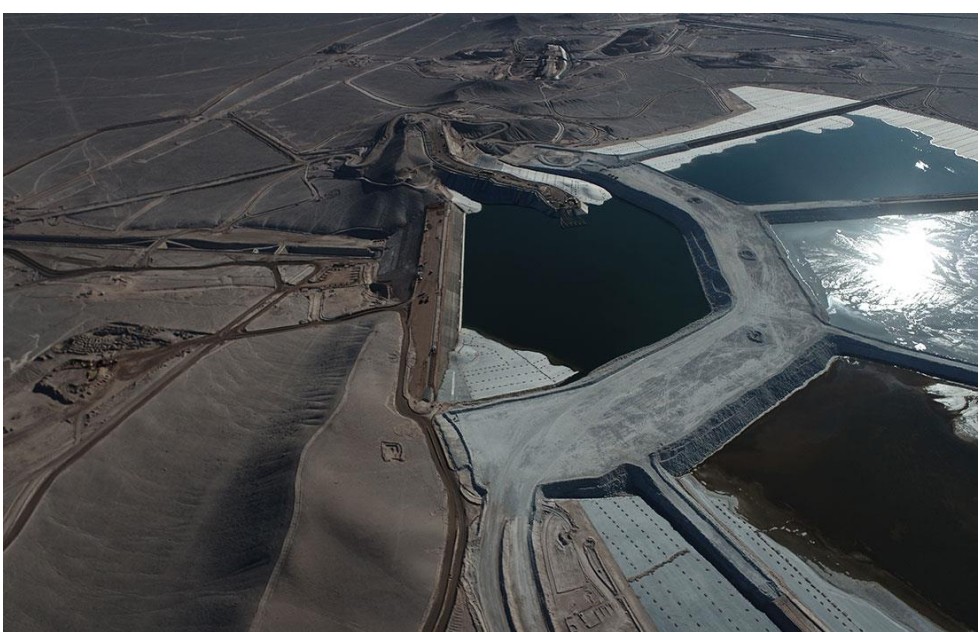

**Figure 27.** Supernatant Water Collected in Water Reservoir in Spence Tailings Storage Facility.

*7.9. Talabre Thickened Tailings—Cell Dyke Disposal—Codelco—Chile*

Chuquicamata mine is presently the biggest open pit copper mine worldwide. It is located 15 km north of Calama City and about 245 km northeast of Antofagasta City. Mining processes began in 1915 and since 1971 (Codelco Chile) Chuquicamata have had a remarkable growth throughout time, involving a large generation of mine tailings that have been safely stored in the tailings deposit named Talabre, which is located 15 km northeast of Calama City (Figure 28). The Chuquicamata mine has over 100 years of operation and still has reserves for the next 40 years and beyond and is operated by Codelco (National

Copper Corporation of Chile). Talabre TSF is currently one the world's largest conventional tailings disposal in terms of area, reaching more than 50 km$^2$ [32].

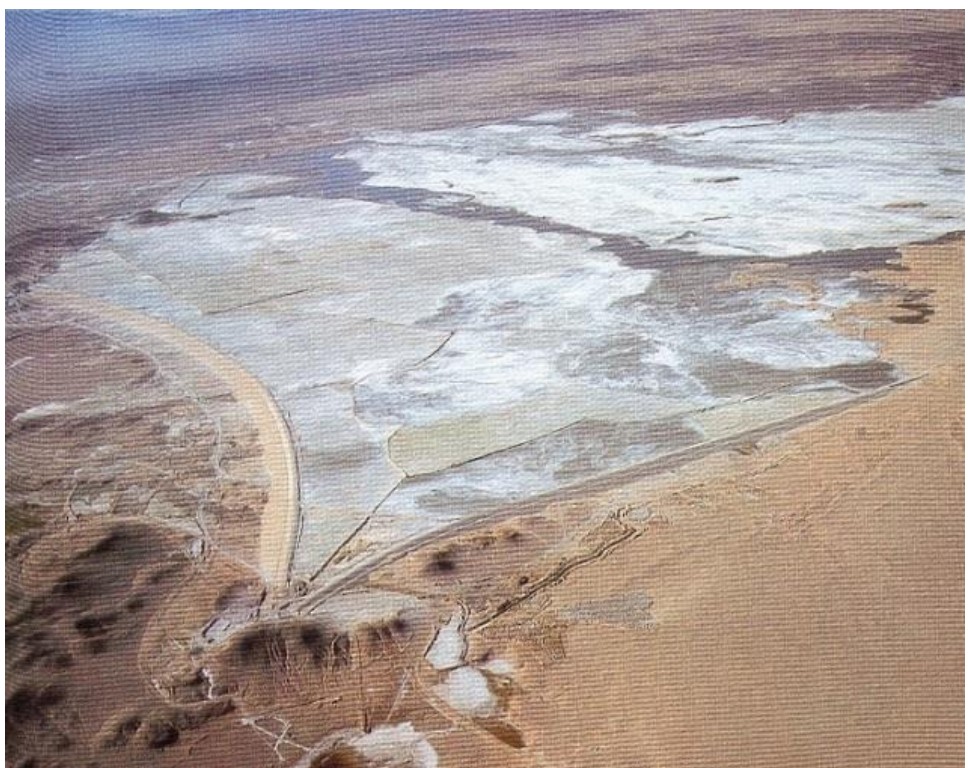

**Figure 28.** Panoramic View of Talabre Tailings Storage Facility.

Talabre tailings storage facility mainly consists of three resistant dams called North, South and West Dams. These three dams have a total length of 11 km and a maximum height of 45 m (2490 masl). Initially the dams were constructed using the downstream construction method, compacting cycloned tailings sand up to elevation 2485 masl. Later, the construction of these dams was modified by the center line construction method, using compacted mine waste rock material obtained from the open pit. Both upstream and downstream slopes in the dam, in the part made of cycloned tailings sand, are 3:1 (H:V) [32]. The deposit of mining tailings is currently carried out in the Talabre TSF in a series of cells where the tailings are contained by interior dikes built with mine waste rock material.

Currently at the Chuquicamata concentrator plant, mining tailings are thickened in 9 conventional type thickener units with a diameter of 91 m (Figure 29). Tailings are currently being deposited at a rate of 200,000 mtpd, having an average solids concentration by weight (Cw) of 57%, which will increases in the near future with the Talabre TTD TSF project to a rate of 400,000 mtpd, considering the additional production of tailings from the new concentrator plants of Ministro Hales and Radomiro Tomic (both operated by Codelco), reaching an average solids concentration by weight (Cw) of 67% [32].

The mining tailings thickening process at Talabre TTD TSF project will be carried out in a high-density thickening plant that includes 13 high-density thickeners (HDT) with a diameter of 65 m each, and a unit treatment capacity of 35,000 mtpd, having an average solids concentration by weight (Cw) of 67% [59]. The new plant will use an approximate area of 20 Ha and its facilities will be:

- Flocculent plant: facilities for the preparation of reagents and distribution and injection pumps to the tailings feed lines.
- Electrical room.
- Control and operation room, offices.
- Recirculating water reservoir (120,000 m$^3$ capacity).

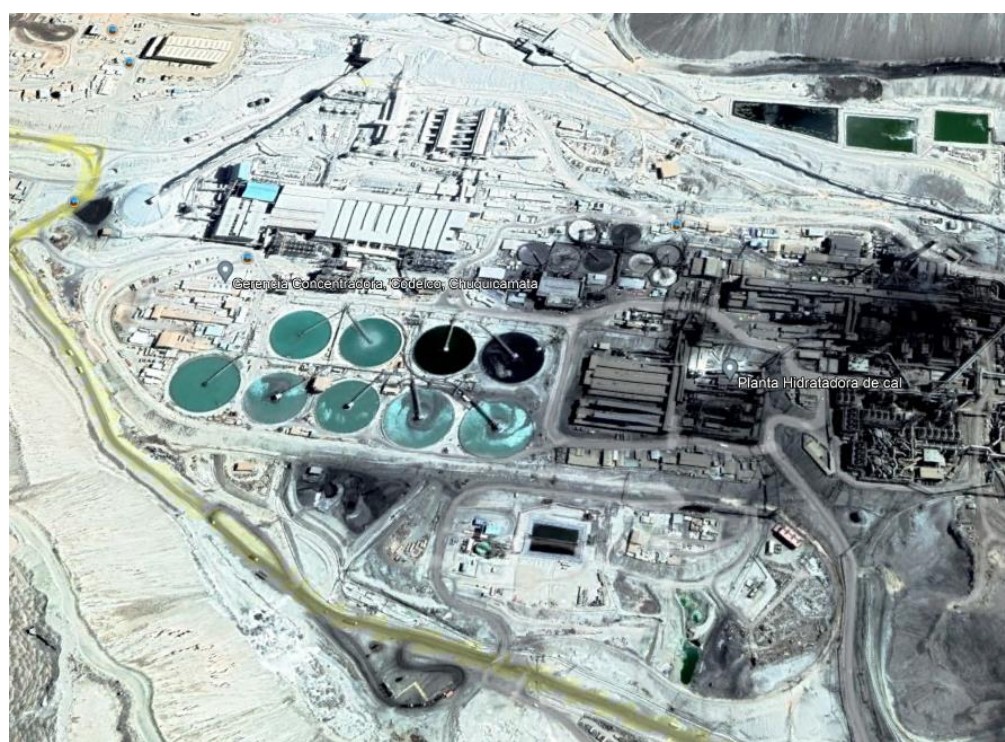

**Figure 29.** Layout View of Tailings Thickeners in the Current Chuquicamata Concentrator Plant.

The Talabre TTD TSF project considers the operational continuity in the long term, by transforming Talabre TSF from conventional to thickened TSF. The new scenario is for a treatment rate of approximately 400,000 mtpd, reaching a total surface over 90 km$^2$ in 40 years. Currently, this is the world's biggest project of thickened tailings disposal, with tremendous technical, economic, environmental and social challenges and benefits associated. The conversion from conventional to thickened tailings involves a huge amount of studies and considerations that give much to the knowledge of the art of thickened tailings and paste disposal in brownfield projects [32].

Conversion of Talabre from conventional into thickened tailings disposal (TTD), has a number of technical challenges and complexities that must be evaluated. Within these, the geotechnical evaluation of materials, to establish the technical feasibility of this transformation is crucial. Due to the high seismicity in Chile, it is very important to know very well the expected properties of the materials involved, among which are the thickened tailings and conventional tailings that make up the foundation soil. Thus, it is possible to be sure of the evaluation of the seismic response of the TSF against a great seismic event [32].

The feasibility studies of the Talabre TTD TSF project shows that the key issues, in terms of stability, should focus on underlying conventional tailings, since thickened tailings have better geotechnical conditions and can be controlled in the operation, according to design specifications. The investigations that are being developed in this line, achieve a greater understanding of the phenomenon of geotechnical resistance in brownfield projects that intend to develop changes in their tailings' disposal technology [32].

### 7.10. Los Corralillos Thickened Tailings—Down Valley Discharge—Cerro Negro Norte CMP—Chile

Cerro Negro Norte CMP is an iron mine located on the southern border of the Atacama Desert, approximately 800 km north of Santiago, Chile. This region is characterized by scarce precipitations and almost nil vegetation except in the Copiapó Valley, located approximately 30 km from the mine. This valley has, economically-speaking, two main areas: (i) mining, and (ii) agriculture. The most relevant agricultural activity is the production of grapes for export. As a consequence, the agricultural activity competes with mining for the

water resource which is very scarce and comes almost completely from snow melt off the Los Andes mountains.

The scarcity of the water resource has made it ever more necessary that the mining process diminish its consumption of fresh water in order to obtain the environmental permits, reduce the dependence on sources (mostly groundwater wells), diminish investment costs to obtain water from other sources, and, finally, reduce the risk of running out of water [33].

Even though the mining companies are the owners of legal rights to the water, this does not mean they are allowed to omit the optimization studies to reduce water losses, or what is equivalent, on improving water reclaim in the floatation and tailings generation processes [33].

This chapter of the paper presents the performance of the tailings management at the Cerro Negro Norte Mine owned by Cia. Minera del Pacífico (CMP). This is where one of the critical aspects is presented by the design of a tailings impoundment that combines the following characteristics: (a) high water reclaim rate, (b) low investment and operational costs, and (c) the application of technology to minimize the risk. One of the critical aspects is the estimate of the concentration limit of the tailings that allows it to be pumped by centrifugal pumps without risks. Another important aspect is the estimate of the tailings beach slope what affects the size of the dam embankment and the elevation of the ultimate discharge point [33].

The mining project considers a tailings production of the order of 20,000 mtpd, where the tailings are thickened to a concentration of solids by weight (Cw) of the order of 65% in 2 thickeners of the high-density type of 40 m in diameter (Figure 30). After being thickened, the mining tailings are pumped through pipes to be taken to the upper part of the tailings deposit, where the spigots allow the tailings to be discharged under the method called down valley discharge (Figures 31 and 32) [60,61].

The tailings deposit dam considers the use of mine waste rock construction material from the mine, which will be placed by mining truck. A dam with a maximum height of 88 m with a length of 2500 m and a use area equivalent to 2500 hectares is projected. Finally, it is expected to have a maximum tailings filling capacity in the deposit of the order of 110 million tons considering a thickened tailings deposition slope of the order of 1.5 to 2.5% [60,61].

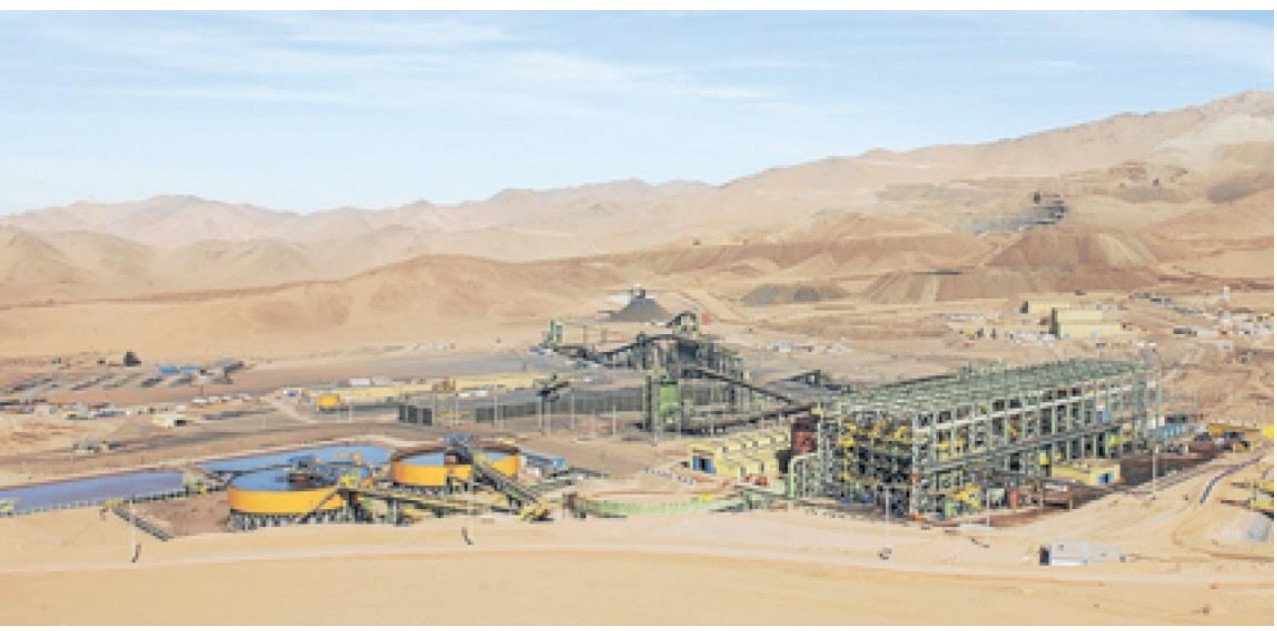

**Figure 30.** Tailings Thickening and Pumping Station—Cerro Negro Norte Process Plant.

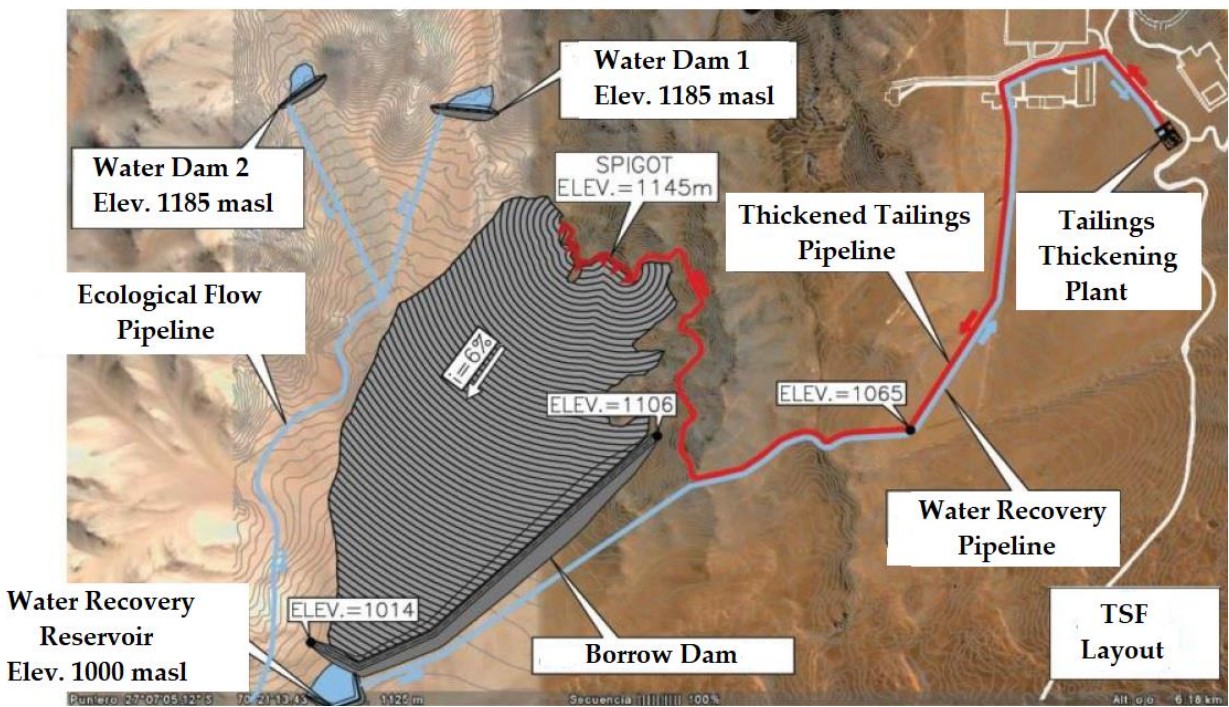

**Figure 31.** TSF Los Corralillos Layout View in Cerro Negro Norte Mining Project.

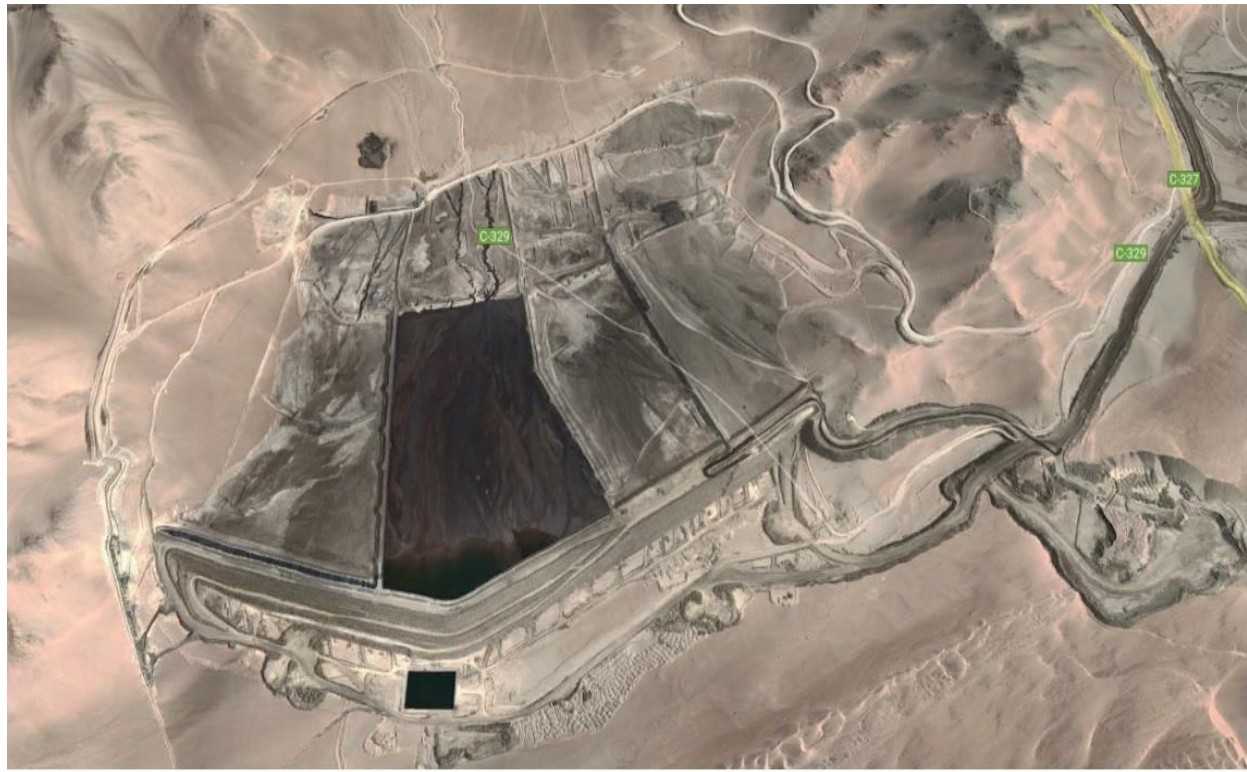

**Figure 32.** Los Corralillos Thickened TSF in Cerro Negro Norte Mining Project.

*7.11. Carmen de Andacollo Thickened Tailings—Down Valley Discharge—Teck—Chile*

Carmen de Andacollo is located in the Coquimbo Region of central Chile at an elevation of 1000 metres, approximately 350 kilometres north of Santiago, near the southern limit of the Atacama Desert. Teck owns a 90% interest in the mine. Empresa Nacional de Minería (ENAMI) holds the remaining 10%.

Carmen de Andacollo is an open pit mine, producing copper in concentrates from the hypogene portion of the orebody. Copper cathode production from the supergene portion of the orebody is currently approaching completion. The majority of mine personnel live in the town of Andacollo, immediately adjacent to the mine, or in the nearby cities of Coquimbo and La Serena [34,62].

The mining project considers a tailings production of the order of 55,000 mtpd, where the tailings are thickened to a concentration of solids by weight (Cw) of the order of 58%. The TSF is called the Carmen de Andacollo tailings dam and its development contemplates 6 stages of growth, to satisfy the design useful life of 21 years, with a storage capacity of 416 million tons—297 million m$^3$ according to current permits. The tailings dam is made up of the west, north, northeast, east and south dams, including the structure called the south dump and a small dam (called the Closing dam) that is built during stage 6, to the west of the south abutment of the eastern end dam (Figure 31) [34,62].

All the dams are built with mine waste rock material loaded by mining truck, and built by the downstream method, and waterproofed with a 1.5 mm thick HDPE geomembrane, on the internal slope of Stage 1 of the dams (up to El. 1117 masl), in addition to the south dump and in all stages for the south dam and the south section of the east dam. The rest of the dams (including the south dump) are covered with low-permeability borrow material (transition fill) on the internal slope. The base of each dam includes a drainage system (French type) designed with a safety factor SF = 5.0, whose effluent is returned to the concentrator plant process (Figure 33) [34,62].

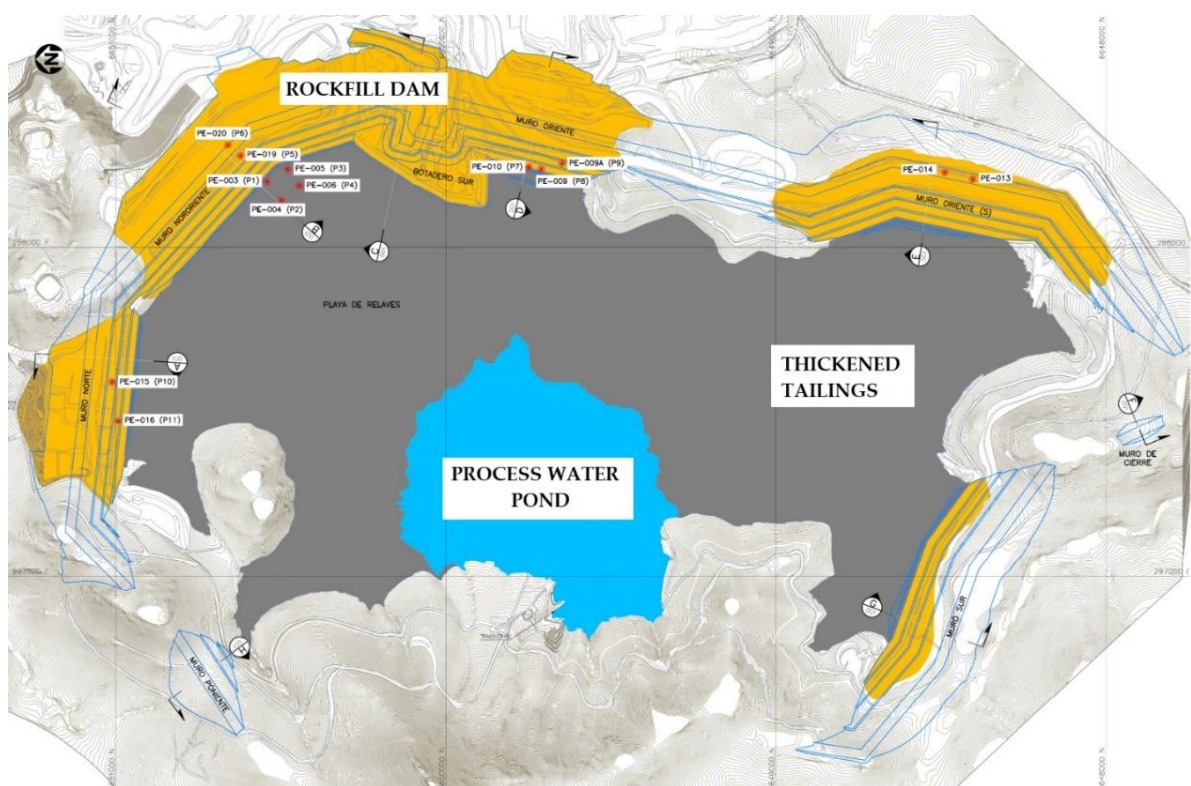

**Figure 33.** Layout View of Andacollo Thickened TSF.

The TSF operates within the limits defined by stage 3 of growth, receiving a total of 16.94 million tons of tailings during the period, through the nine (9) discharges distributed at the crest of said stage (El. 1149.5 masl) accumulating a total of 131.6 million tons as of 2018. The tailings dam is divided into three deposition sectors, the northern sector that concentrated 37% of the annual tailings discharge and the central and southern sectors that received 27% and 36%, respectively [34,62].

The volume of the clear water pond reached a maximum of 1.4 million m$^3$ in mid-May 2017, the date on which the minimum distance from the pond to the dam of approximately 300 m was verified, the elevation of the water in the pond varied between the elevations El. 1132.994 masl and El. 1136.934 masl (3.94 m increment) (See Figure 34). It is expected that the tailings deposit slope on the beach is of the order of 1.0% considering the deposit scheme mentioned above [34,62].

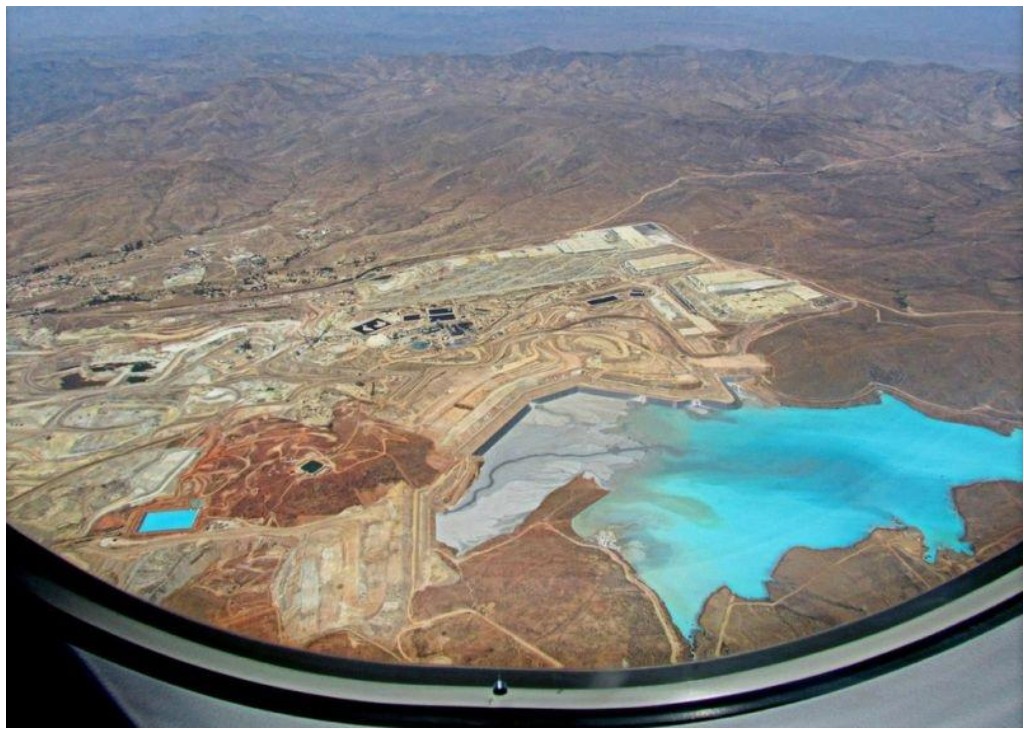

**Figure 34.** Panoramic View of Andacollo Thickened TSF.

### 7.12. *Los Diques Thickened Tailings—Down Valley Discharge—Candelaria—Chile*

The Candelaria Mine is an active open pit copper mine operated by Compañía Contractual Minera Candelaria (CCMC) located in the Atacama region of Chile, 20 km south of Copiapó city. The Candelaria Mine has been in operation since 1993 and is currently operated by CCMC. The key waste and water management facilities include the Candelaria TSF, the more recently commissioned Los Diques TSF, and several mine waste rock disposal sites including the North Waste Dump (Figure 35) [35,63,64].

The Candelaria TSF includes a main embankment and two saddle embankments (North and South) and a seepage collection system (SCS). The SCS collects drainage via an underdrain system excavated into the underlying alluvium at the Starter Embankment and is conveyed towards the cut-off trench downstream of the Main Embankment. Seepage water is recovered by pumping wells (named Pique Mina) located upstream of the cut-off trench and transferred to the plant for use as process water [35,63,64].

In 2018, the initial construction phases of the new Los Diques tailings storage facility were completed and the facility received its first tailings during the first quarter as part of commissioning. The Los Diques tailings storage facility can now receive 100% of the flotation tailings from the Minera Candelaria processing plant. The Los Diques tailings storage facility is located to the southwest of the open pit and plant sites and will have an approximate designed capacity of 600 million tons. Flotation tailings from the processing plant continue to be deposited in the Candelaria tailings storage facility similar to a contingency case. The Candelaria flotation tails are produced at 75,000 mtpd, thickened by two high-capacity thickeners and then pumped to the Los Diques tailings storage facility at an average solids concentration (Cw) of 50% (See Figure 36) [35,63,64].

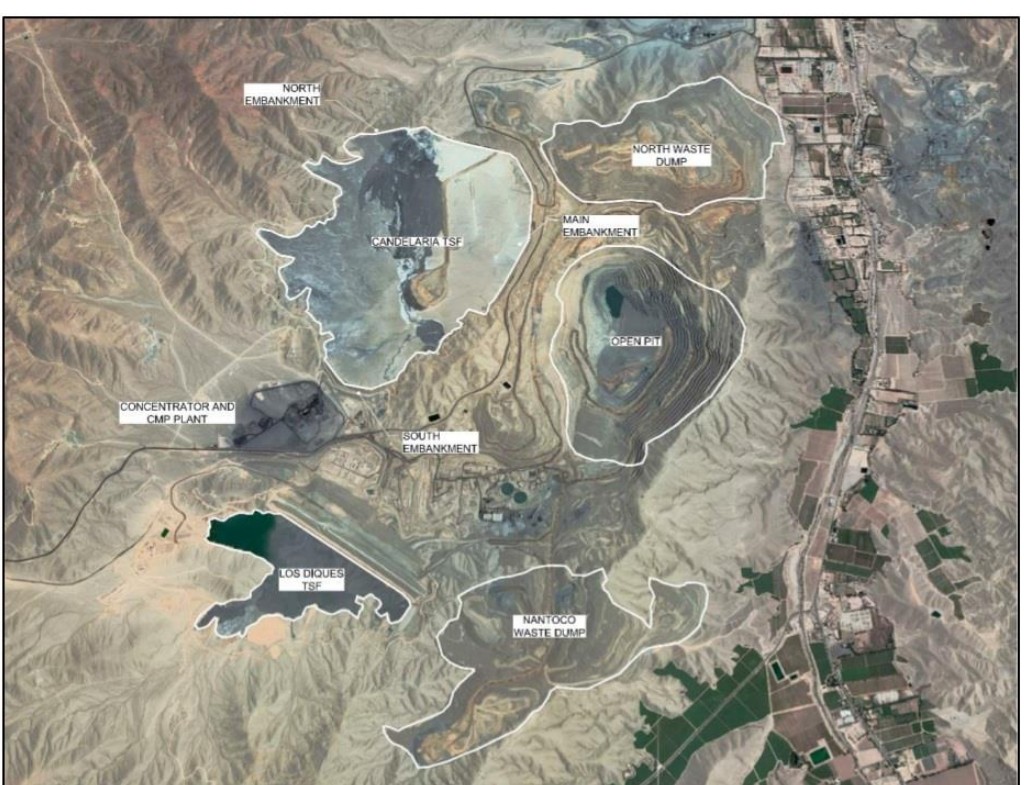

**Figure 35.** Layout Candelaria Mining Project.

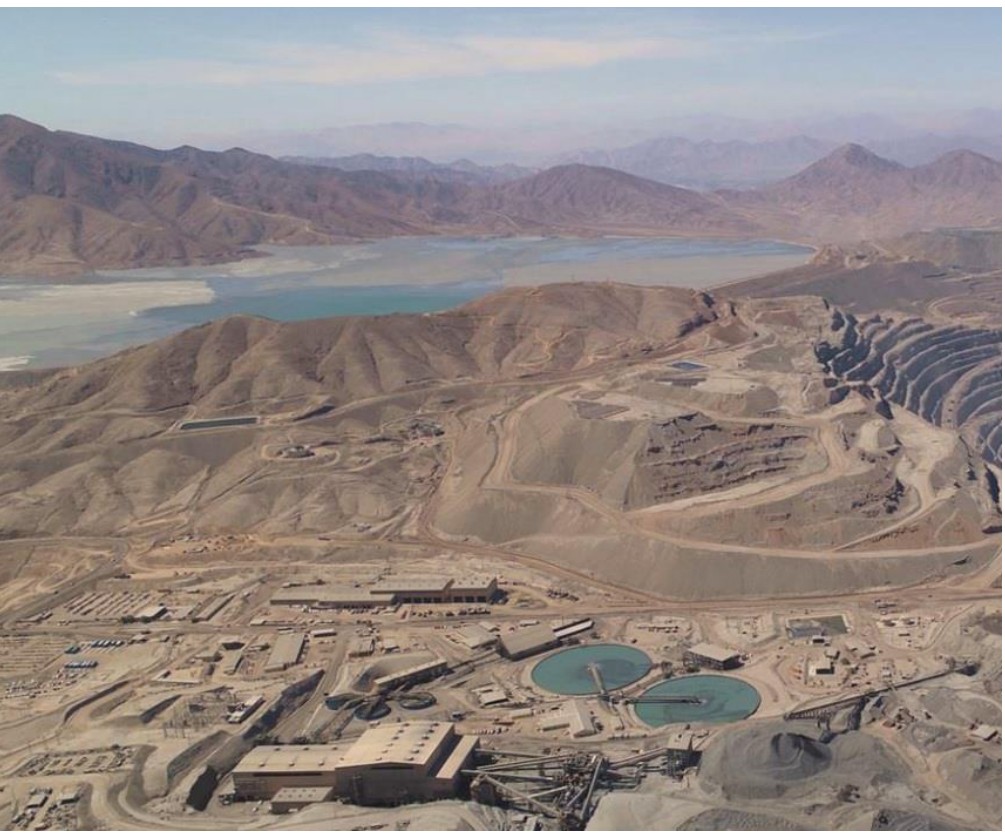

**Figure 36.** Candelaria Concentrator Plant.

The new tailings storage facility called Los Diques (Figure 37) is designed with three embankments, all of them built from mine waste rock material, with transition and filter

zones built from engineered fill borrowed from inside the containment area. The main embankment has underdrains to facilitate water recovery. The design includes a geomembrane on the upstream slope and a grout curtain for the north and south embankments only. In addition, a cut-off wall and drain wall along with extraction wells collect seepage water for recirculation to the processing plant [35,63,64].

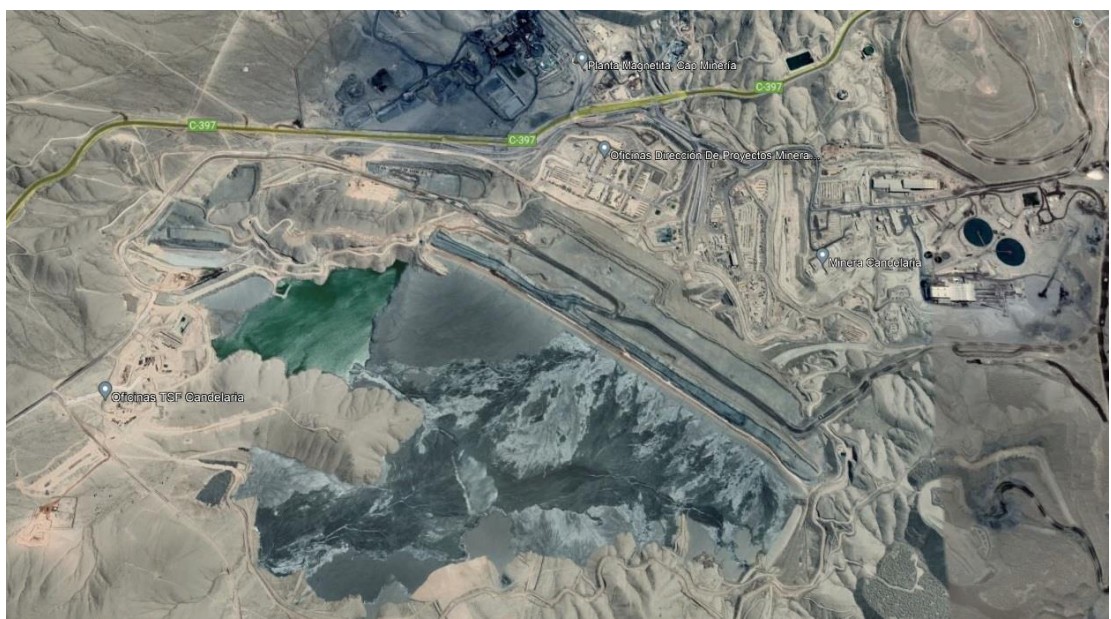

**Figure 37.** Los Diques Thickened TSF.

The tailings will be deposited from the dams through spigots, and also from the southern perimeter of the basin from a single discharge point (See Figure 37). The tailings disposal modeling is based on the objective of maintaining a single pond that maintains a trajectory towards the North dam and then towards the East side of the basin. The tailings disposal should be designed to maximize the existing beach distance between the supernatant pond and the dams, respecting the projected trajectory of the pond [35,63,64].

*7.13. Sector N°5 Paste Tailings—Down Valley Discharge—Coemin—Chile*

Cerrillos Plant, owned by Compañía Exploradora y Explotadora Minera Chileno Rumana S.A, is located 30 km. south of the city of Copiapó in Chile and produces copper concentrates through a conventional flotation process. The ore comes from Mina Carola, which is mainly composed of chalcopyrite and copper ore [37,65].

Within the production process, the operation of a state-of-the-art tailings thickening equipment called a deep cone thickener stands out, which allows depositing paste tailings instead of conventional tailings, recovering and reusing over 85% of the process water, in addition to provide greater physical and chemical stability to the mining deposit. Construction of a paste tailings reservoir that receives the residues from the metallurgic process of the Cerrillos plant, thus disposing it in a safe manner and minimizing its environmental impact. It is considered that the mineral concentrator plant produces around 7000 mtpd of mining tailings, which are thickened to a concentration of solids by weight (Cw) around 70% in a Deep Cone Thickener (See Figure 38) [37,65].

Implementing a paste tailings cover over the basin in order to close the existing TSF 1, using the thickening facilities and big part of the existing infrastructure for the conduction of thickened tailings (Figure 39). Extend the useful life of deposit TSF 1 by filling it with paste tailings and the growth of the sand wall up to a height of 662 masl in conjunction with the fitting out of a shoulder with modular gabions in the north-west abutment. In addition, it is considered to enable a rainwater management system for the diversion of possible runoff [37,65].

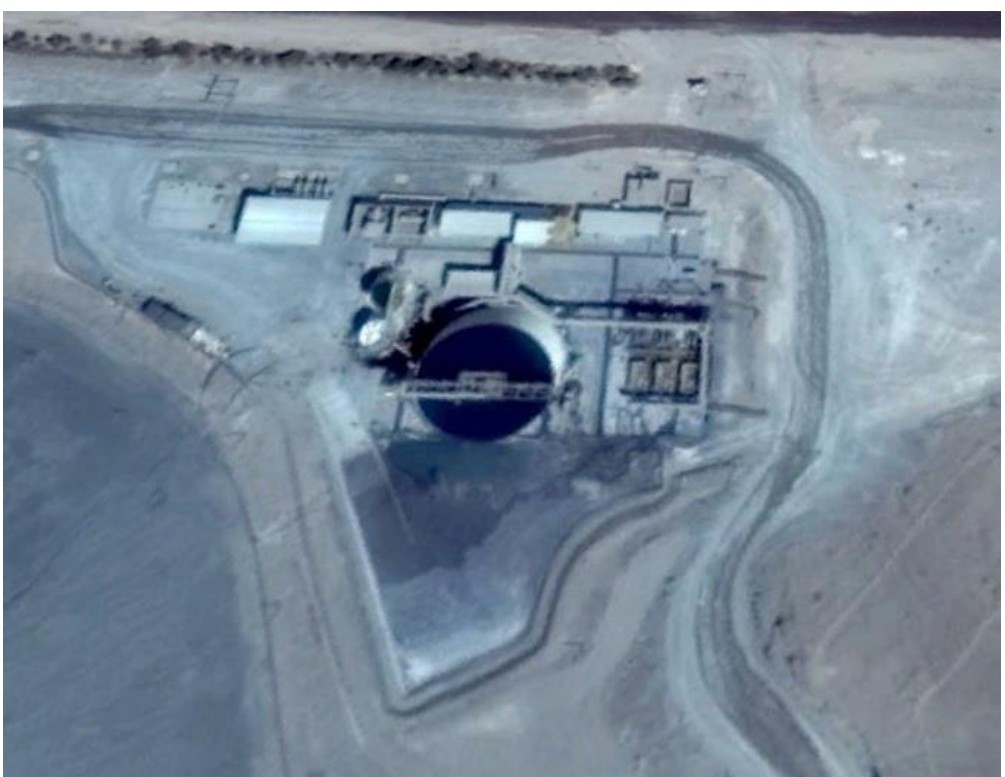

**Figure 38.** Coemin Tailings Thickening Plant.

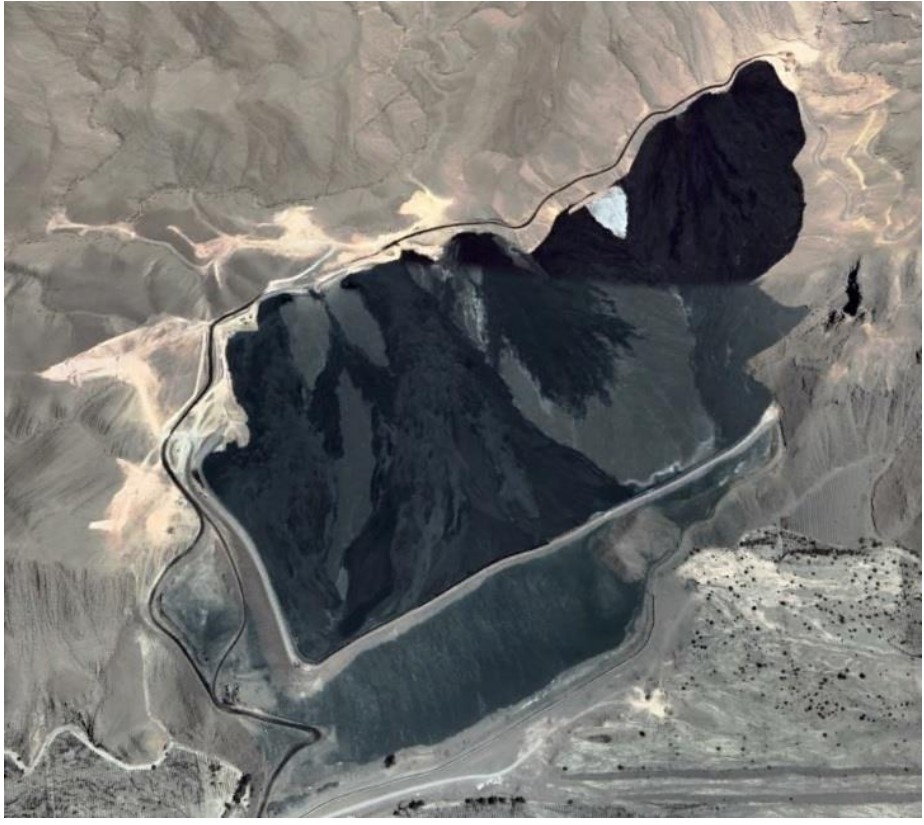

**Figure 39.** Coemin Thickened TSF 1.

Due to the fact that the maximum tailings storage capacity of the TSF 1 deposit is about to be reached, Coemin has a new deposit called Sector N°5 Coemin TSF (Figure 40)

which will be built with a small retention dam and completely filled with paste tailings. It is expected to discharge the tailings through spigots from the upper parts of the basin and thus achieve a deposition slope of the order of 2.0 to 3.0%. Considering the level of thickening of the tailings and the distance between the thickening plant and the Sector N°5 TSF tailings deposit, it will be necessary to pump the tailings with a positive displacement pumping system (Figure 41) [37,65].

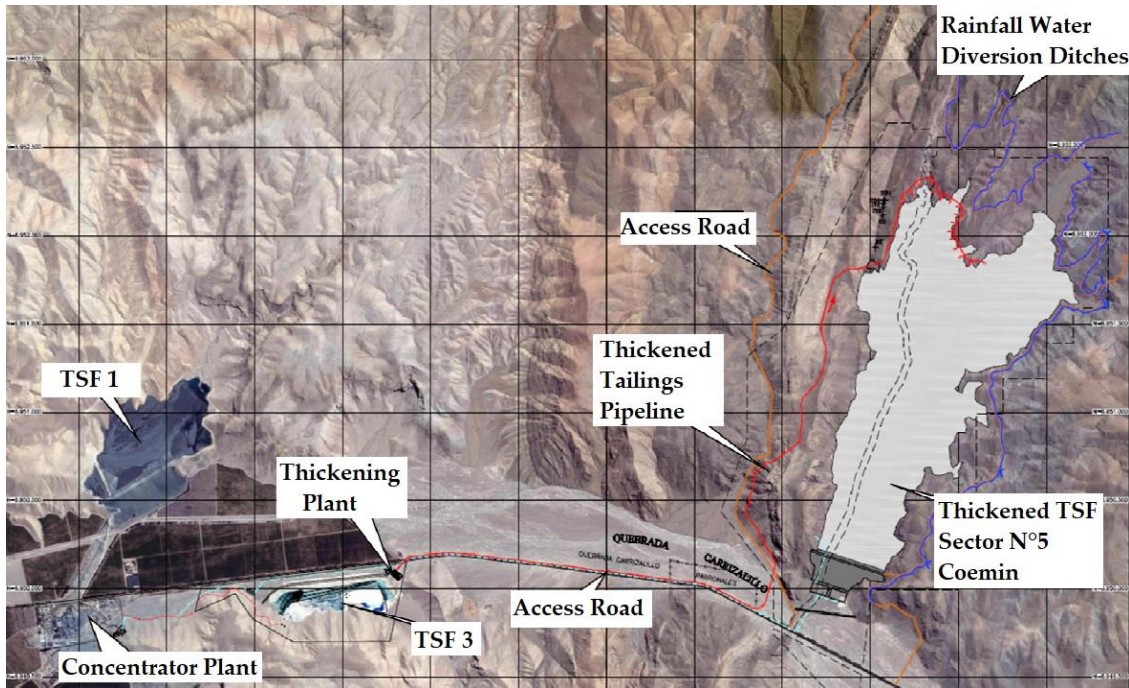

**Figure 40.** Coemin Thickened TSFs—TSF 1 and TSF Sector N°5 Coemin Layout View.

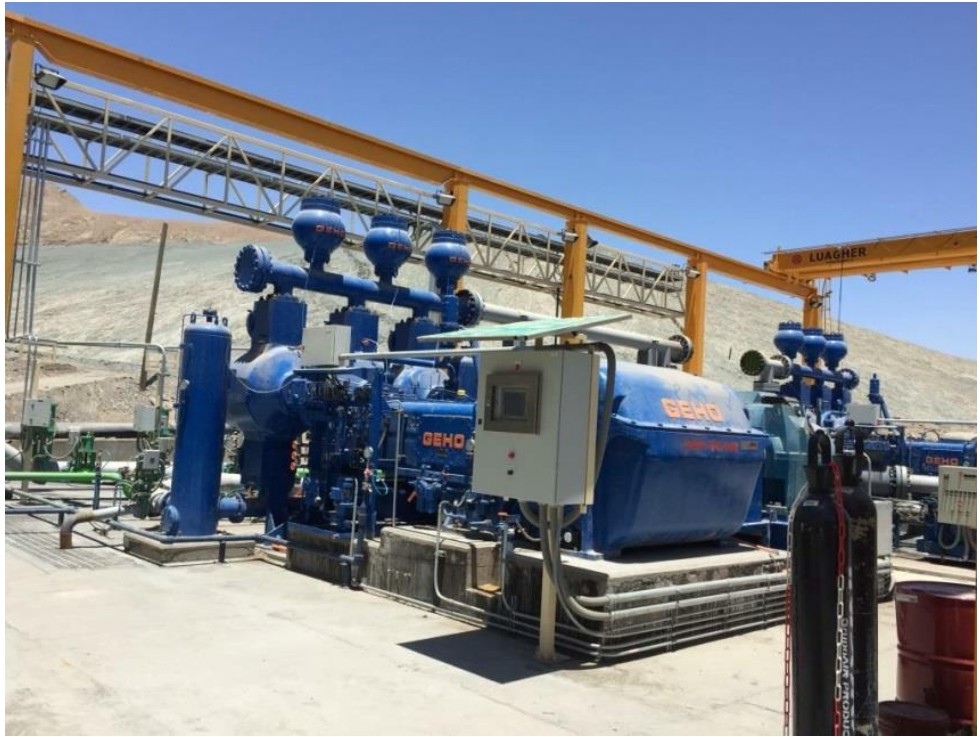

**Figure 41.** Coemin Positive Displacement Pump Station.

### 7.14. Toromocho Thickened Tailings—Down Valley Discharge—Chinalco—Peru

The Toromocho copper-molybdenum project owned by Chinalco is located in the Morococha mining district of Peru north-east of Lima. The site is located between 4500 masl and 5000 masl in a moderately wet climate that receives about 851 mm of precipitation per year. The resource exceeds 1.5 billion (metric) tons mined in an open pit. The ore is processed by milling and flotation to produce a concentrate which is shipped off site for mineral extraction.

The average flotation tailings production is 140,000 mtpd. The tailings are thickened to a slurry density of about solid concentration by weight (Cw) 65% in 4 high density thickeners of 40 m diameter each one (Figure 42).

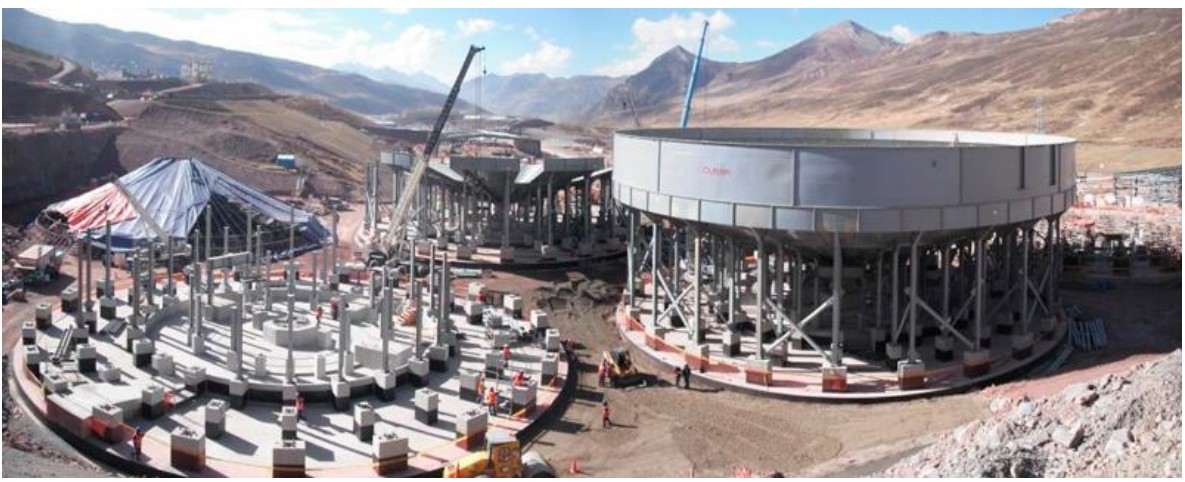

**Figure 42.** Tailings Thickeners under Construction—Toromocho Mining Project.

The transport of the tailings from the concentrator plant to the tailings deposit is carried out through pumping with positive displacement type pumps, with a total of 10 units, these being the largest slurry transport pumps in size currently in the world (Figure 43) [66].

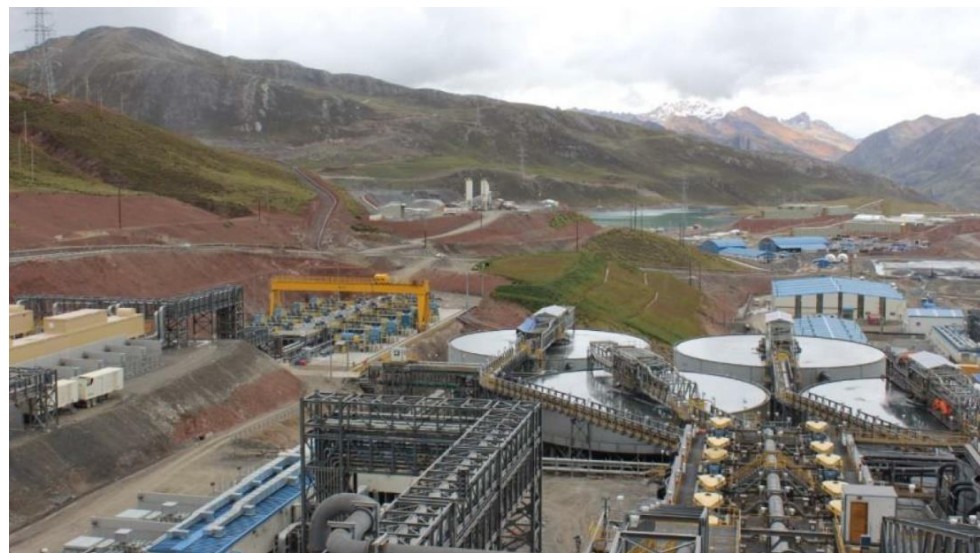

**Figure 43.** Tailings Thickening Plant and Positive Displacement Pump Station—Toromocho Mining Project.

Approximately 1500 million tons of tailings will be discharged via a series of pipelines and spigots into the Tunshuruco tailings storage facility located immediately north-east of the mill plant [67,68].

Thickened tailings (non-segregating) were selected based on the following factors:

- The tailings pond is small and easy to manage during operations and upon closure;
- A homogeneous tailings deposit has lower susceptibility to oxidation due to a higher degree of saturation maintained in the tailings deposit;
- Lower seepage rates from the base of the facility because of the homogeneity and density of the tailings;
- A lower overall dam height; and
- Significantly reduced risk compared with the other alternatives because there is no pond trapped on top of the tailings to transport the tailings long distances in the event of a dam failure.

The deposition concept involves disposal of thickened tailings from the upper side of the Tunshuruco valley sloping downwards towards the tailings dam with a 3.0% slope during the first 2 years of operation and 5.0% slope thereafter when the beach area is larger (Figure 44).

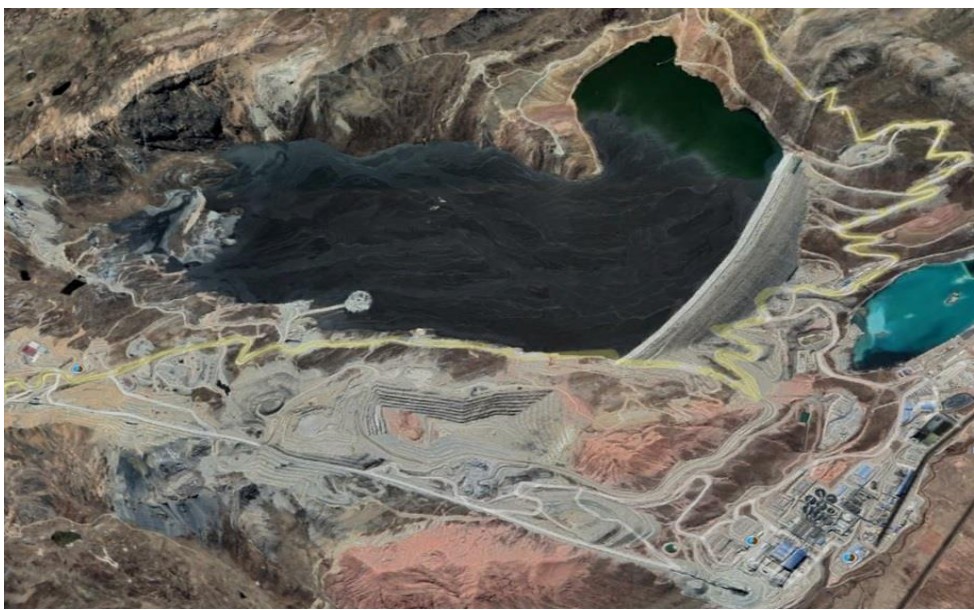

**Figure 44.** Toromocho Mining Project Layout.

At an anticipated deposited dry density of about 1.70 t/m$^3$, the thickened tailings to be deposited in the Tunshuruco valley will occupy a volume of about 750 million m$^3$ [40,56,69].

Thin-lift deposition, achieved by cycling between discharge locations, is an essential component of the deposition plan. Adequate rest times between discharge cycles are required to ensure the tailings gain shear strength and attain the required void ratio before burial by the next lift. The flatter 3.0% slope specified during the initial 2 years considers the rapid rate of rise and smaller surface area lower in the valley. Tailings distribution pipelines and multiple spigots are provided around the majority of the perimeter of the basin to provide depositional flexibility [40,56,69].

The current Tunshuruco valley tailings disposal concept includes a main dam for tailings containment and one dam for water containment: the reclaim pond dam and the seepage collection pond dam. The dams are constructed of durable, non-acid generating mine waste rock with extruded concrete curb and bituminous membranes on the upstream face to inhibit seepage. The construction method of the tailings dam with non-acid-generating rockfill (mine waste rock) material considers transportation and placement with mining trucks, to be later compacted by bulldozer and roller compactor equipment. In addition, the growth will be in stages during the useful life and filling of the tailings deposit, where the construction method is in the direction downstream of the dam. Suitable bedding and filter zones are provided (Figure 45) [40,56,69].

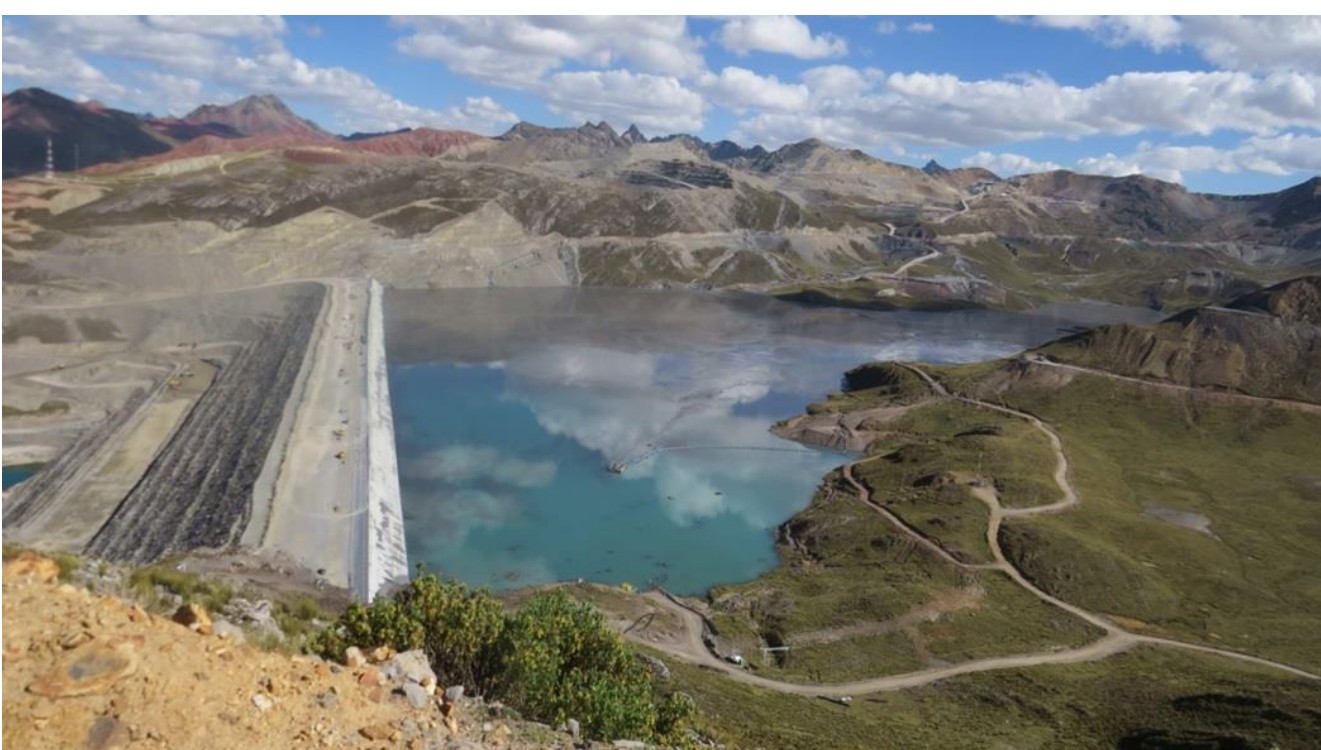

**Figure 45.** Toromocho Thickened TSF.

*7.15. Antapaccay Thickened Tailings—In Pit Tailings Disposal—Tintaya—Peru*

Compania Minera Antapaccay is the owner of the Antapaccay Expansion Tintaya—Coroccohuayco Integration mining project, located in the district and province of Espinar, Cusco region. The cities of Cusco and Arequipa are located approximately 255 km by road to the northwest and southwest of the Project, respectively, while the Yauri town center is located 15 km to the north [41,70].

Antapaccay, which has a useful life of more than 20 years, will benefit from Tintaya's current administrative and logistics infrastructure, as well as the experience of its staff (Figure 46). The site is located between 3800 masl and 4100 masl. In Antapaccay, a skarn-porphyry type deposit will be exploited and will produce a copper concentrate that will be transported by trucks to the port of Matarani. Reserves are estimated at 720 million tons of copper with a grade of 0.56%. In November 2012, Antapaccay began production of copper in commercial grade concentrate and made its first shipment to the port of Matarani for subsequent shipment to customers around the world. The operation will have two open pits and a concentrator plant (single line) with the capacity to treat 75,000 mtpd of sulfide ore through crushing/grinding/concentration. The depleted Tintaya open-pit mine (Figure 46) will serve to deposit mine tailings from Antapaccay, which will considerably reduce the project's capital costs [41,70].

The tailings generated by the Antapaccay project are thickened in 2 high-rate thickeners to reach a solids content by weight (Cw) of the order of 58% (Figure 47). These thickened tailings are later transported by pipeline to the old Tintaya open pit, where it is progressively filled, being its tailings deposit. This tailings deposition methodology is a novelty for the South American region, being unique in its kind, thus marking a reference for the application in other mining operations [41,70].

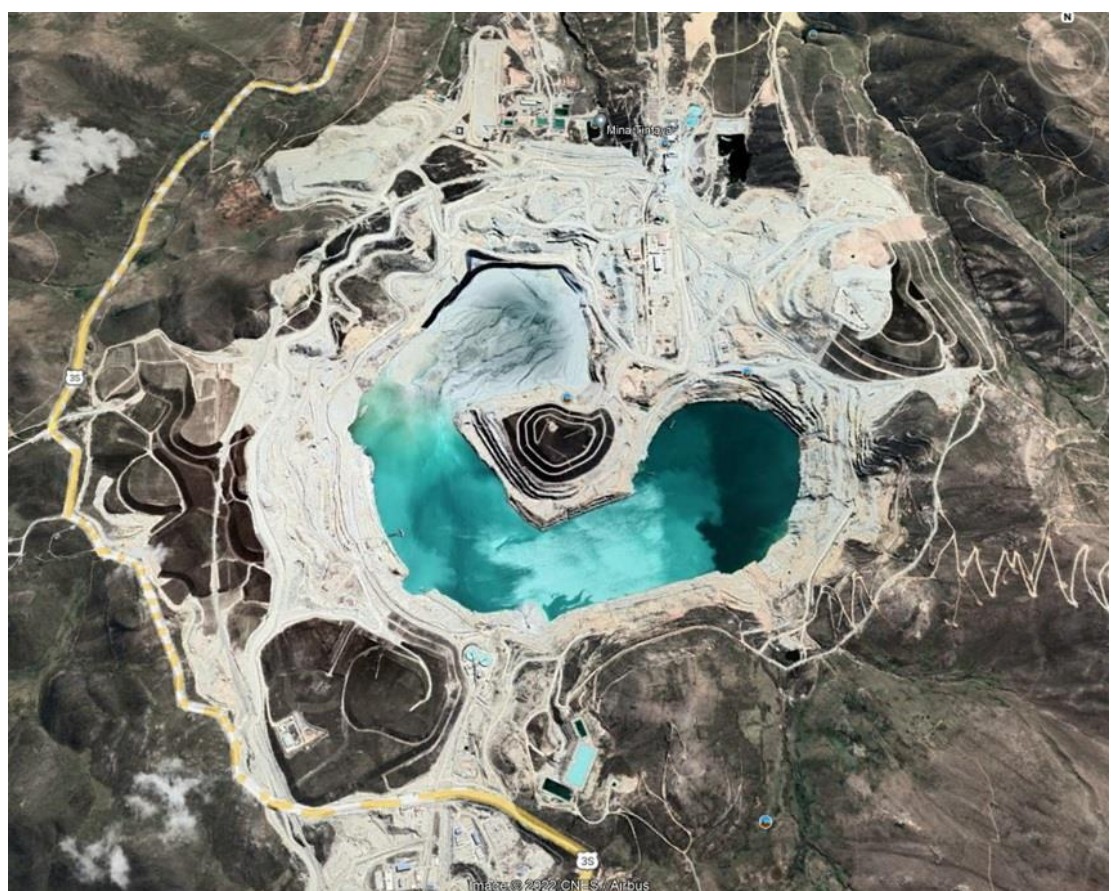

**Figure 46.** Layout View of Tintaya Open Pit.

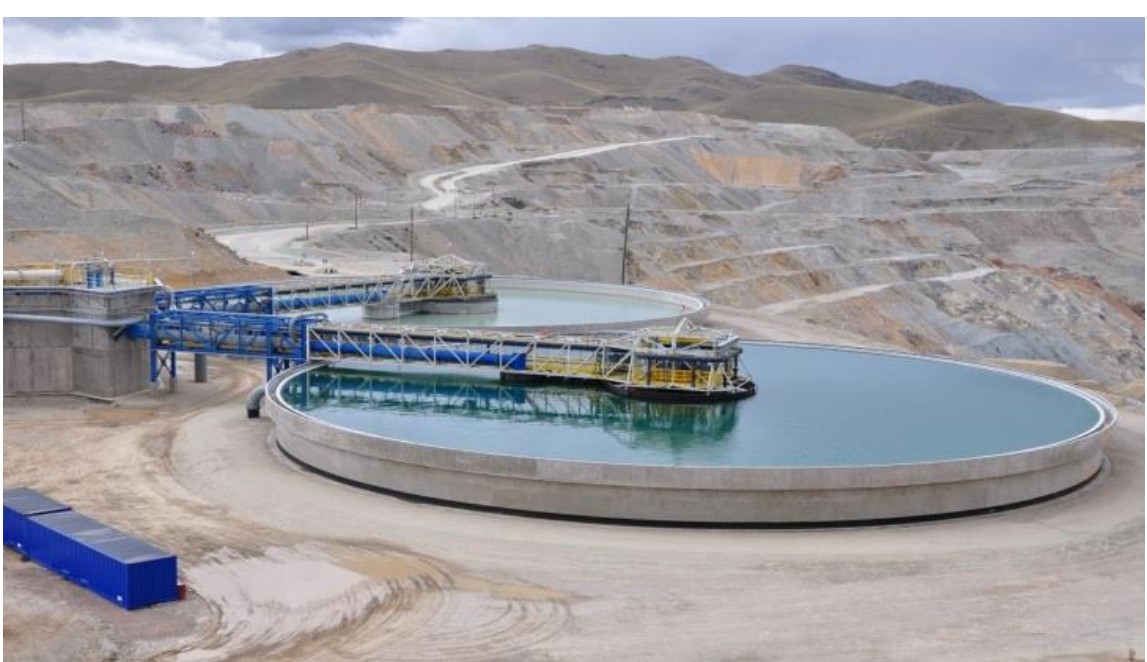

**Figure 47.** Antapaccay Tailings Thickeners Located Close to Tintaya Open Pit.

Part of the water released by the tailings when the consolidation phenomenon occurs and the water collected by rain in the open pit is recovered through pumping equipment and recirculated to the process plant (Figure 48) [41,70].

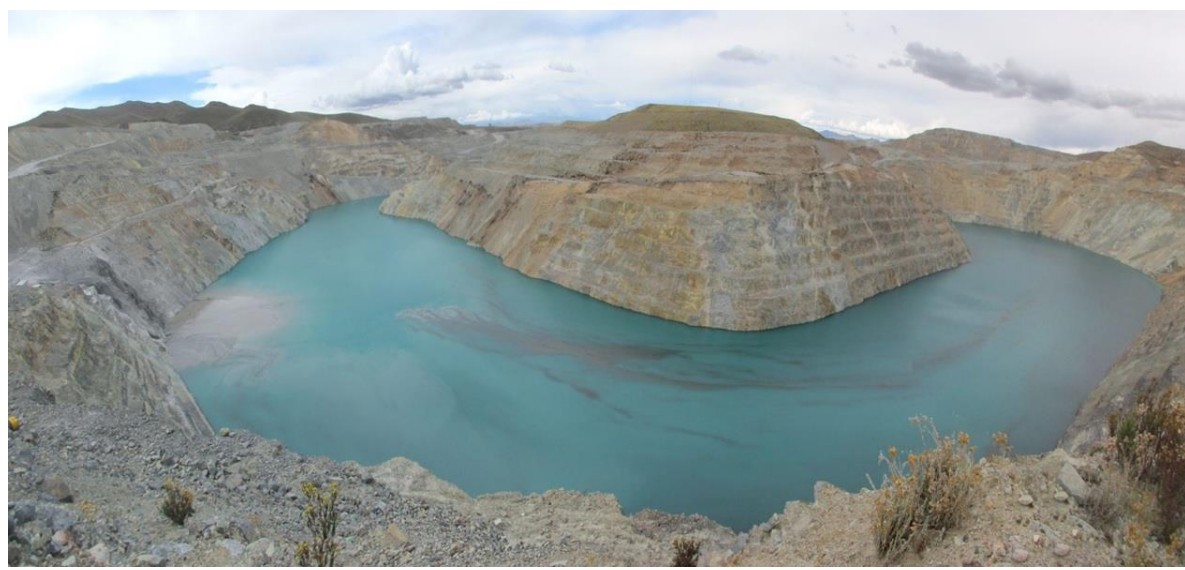

**Figure 48.** Tintaya Open Pit filled with Antapaccay Thickened Tailings.

### 7.16. Las Bambas Thickened Tailings—Down Valley Discharge—MMG—Peru

Las Bambas mining project operated by MMG is located in the Ferrobamba sector in the Apurimac region at an altitude of around 4000 masl.

With an estimated mine life of 18 plus years, the operation produces copper concentrate, with by-products of gold and silver, as well as molybdenum concentrate through conventional processing methods. Ore at Las Bambas is mined from an open pit. The ore is crushed and transported on a 5.5 km overland conveyor to a conventional flotation plant where copper concentrate is produced, then to a molybdenum plant for further processing (Figure 49). Concentrate is then transported by truck and rail to the Port of Matarani in the Arequipa region, where it is shipped to customers worldwide [42,71].

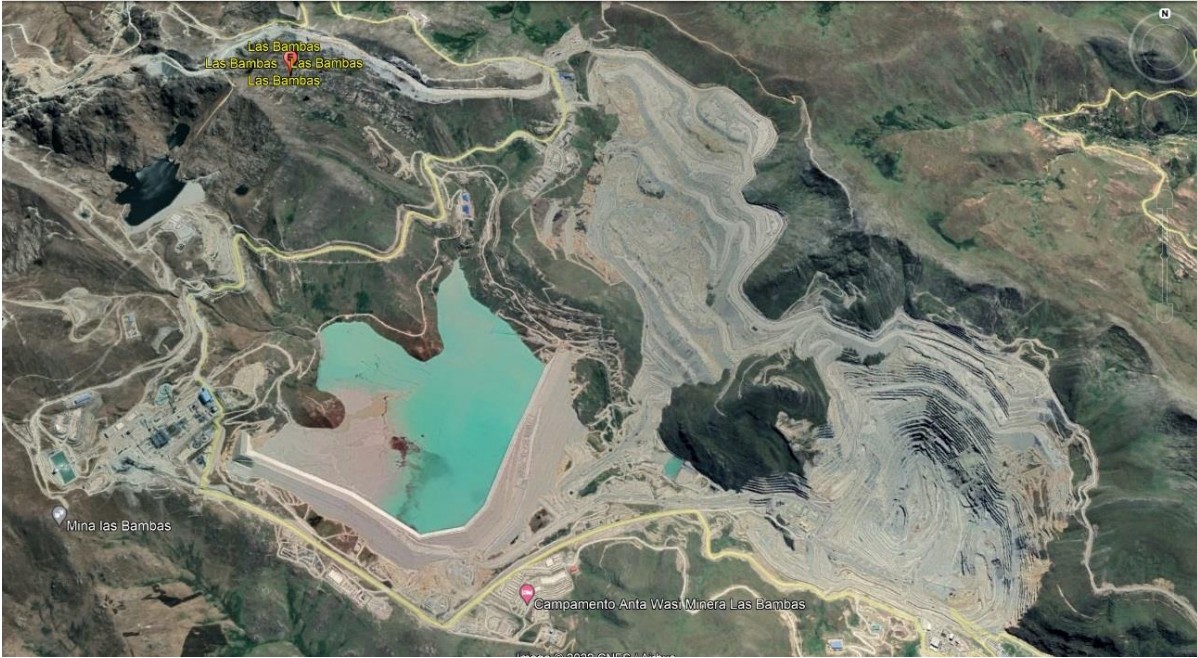

**Figure 49.** Las Bambas Mining Project Layout.

Tailings production is 140,000 mtpd and is processed to be discharged as high-density thickened tailings slurry, with 62% solids concentration (Cw). Water recovered from the thickening overflow will be recirculated to the plant. For the initial operation of the TSF, tailings were discharged by gravity to the upper end of the TSF near the plant. The plant design includes provision for pumping the thickened tailings, but the pumps will not be installed until the rheological characteristics of the tailings are established through actual operation. The planned tailings placement method, which is subject to change depending on actual rheological characteristics of the tailings, is to discharge the thickened tailings from the upstream end of the impoundment (west) and from the impoundment sides (north and south) by means of multiple "spigots" consisting of discharge pipes with lengths of up to about 300 m. This design considers that the tailings will form a final average beach slope of 0.5% sloping from the discharge spigots towards a supernatant water pond [42,71].

The tailings storage facility (TSF) site is located in the upper part of the Ferrobamba Valley, immediately east of the processing plant. The general TSF layout is shown in Figure 50. The TSF will have capacity to store 582 million m³ of tailings, equal to 960 million tons of tailings [42,71].

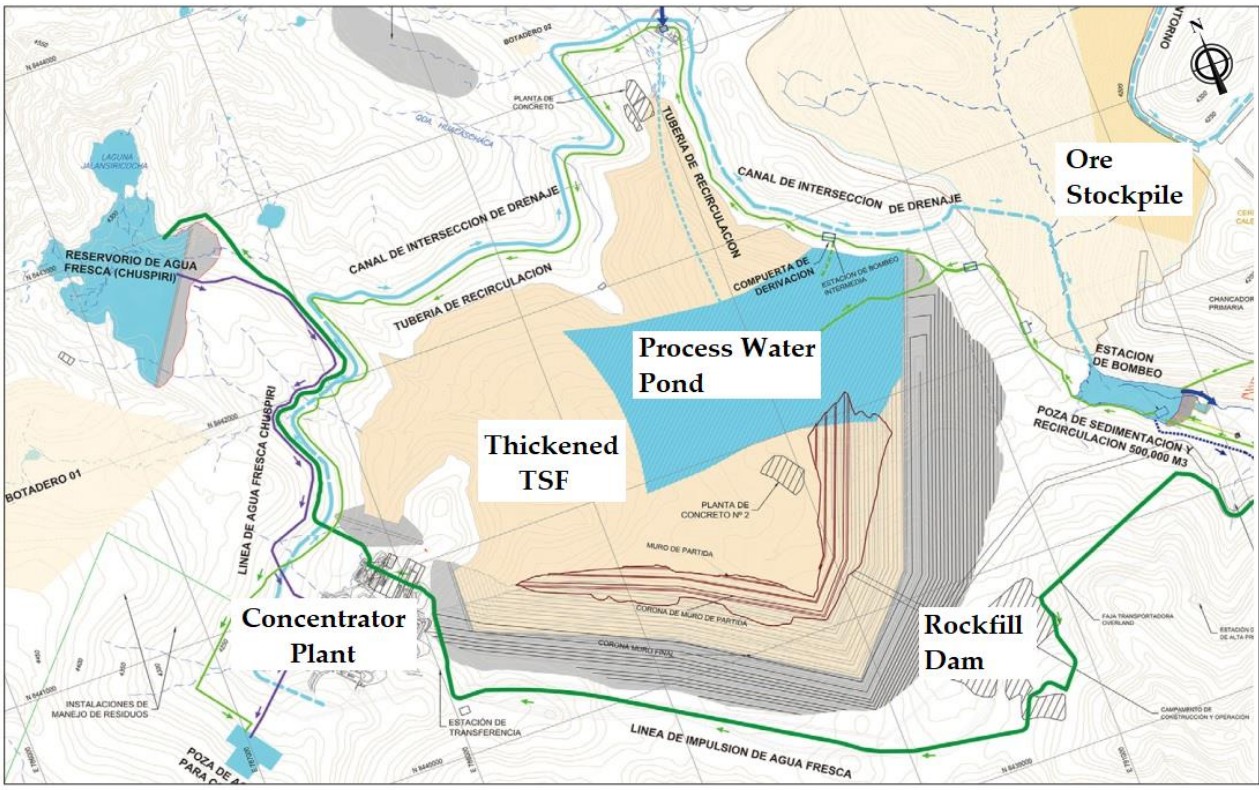

**Figure 50.** Layout of Las Bambas Thickened TSF.

The containment dam is constructed with waste rock from the Ferrobamba pit. It will have a maximum height of 220 m and will be raised sequentially by the downstream method, starting from an 80-m high starter embankment. The upstream face will have a slope of 1.75H:1.0V. The downstream slope will be 1.75H:1.0V. The upstream slope will be covered with a polyvinyl chloride ("PVC") geocomposite liner installed on concrete curbs. A concrete plinth is placed in a trench excavated to competent rock on the upstream toe of the dam and a cement-grout cut-off curtain established under the plinth to control infiltration from the impoundment (See Figure 51) [42,71].

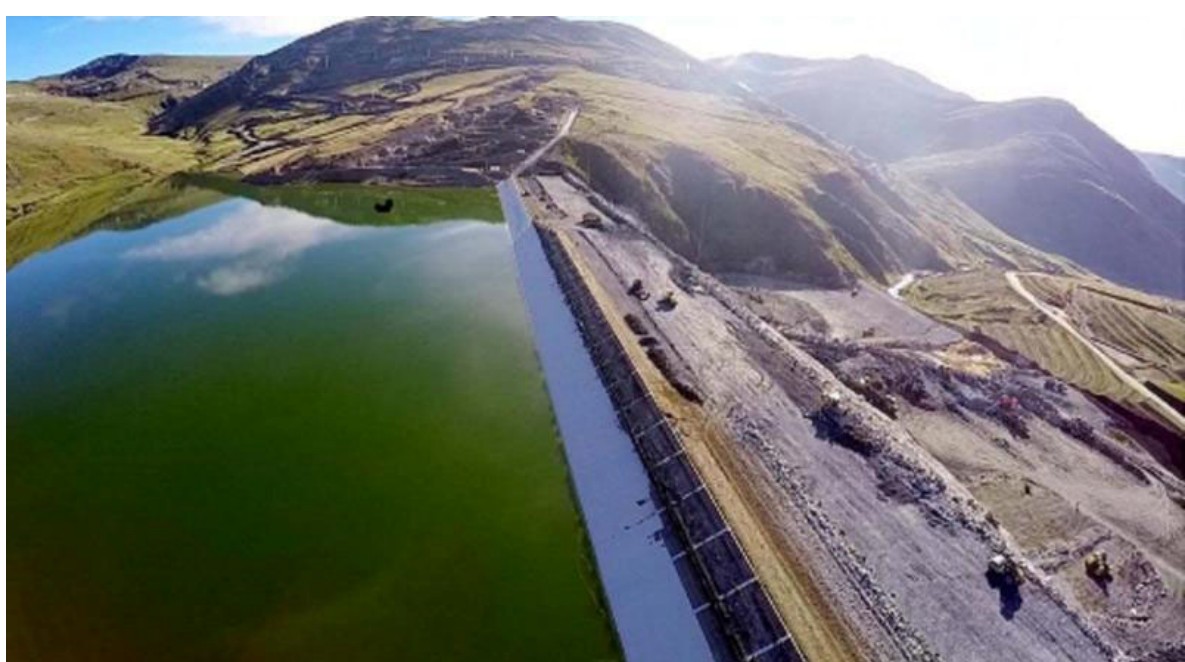

**Figure 51.** Dam Crest View of Las Bambas Thickened TSF.

The water associated with the TSF includes direct precipitation on the impoundment area, water expelled from the tailings due to consolidation. Contact water also includes runoff from upslope areas not intercepted by the non-contact water diversion channel during major storm events. Water will be collected in a supernatant water pond located in the central part of the impoundment toward its northeast corner, against the upstream slope of the containment dam. Water in the impoundment dam will be pumped to the plant using a floating barge pump in the supernatant pond [42,71].

### 7.17. Constancia Thickened Tailings—Down Valley Discharge—Hudbay—Peru

The Constancia open-pit copper mine, owned and operated by Hudbay Perú SAC, is located in southern Perú, approximately 100 km (km) south of Cusco city, at elevations ranging from 3900 to 4500 masl. The Constancia deposit is a porphyry copper-molybdenum system, emplaced in multiple phases of monzonites and monzonite porphyry. Milling rates are in the range of 80,000 to 90,000 mtpd. The original mine plan considered 525 million tons of ore and 538 million tons of mine waste rock to be produced over a 17-year life of mine [43].

The process plant employs a conventional grinding and flotation circuit, from which tailings are thickened to a solid concentration by weight (Cw) of 58% and then tailings are pumped approximately 5.3 km through a high-density polyethylene tailings delivery line and spigotted into a partially-lined tailings storage facility [43]. Figures 52 and 53 shows the layout of the TSF, processing plant, open pit, and other facilities at the site.

In the TSF, tailings are contained by an cross-valley tailings dam with multiple small saddle dams, all constructed of rockfill (mine waste rock), that will have a crest length of 4.1 km at design height. The TSF was designed to store approximately 290 million cubic meters of tailings at a maximum dam height of 170 m (m), corresponding to a crest elevation of 4160 masl. Construction of the TSF was initiated in 2013, and the facility was placed into operations in late 2014, with full production achieved in early 2015. Since then, the dam has been incrementally raised to a height of 107 m as of December 2017. Prefeasibility designs have been prepared to raise the ultimate height to 200 m (crest elevation 4190 masl) to accommodate additional reserves, and final design studies for expansion are in progress [43].

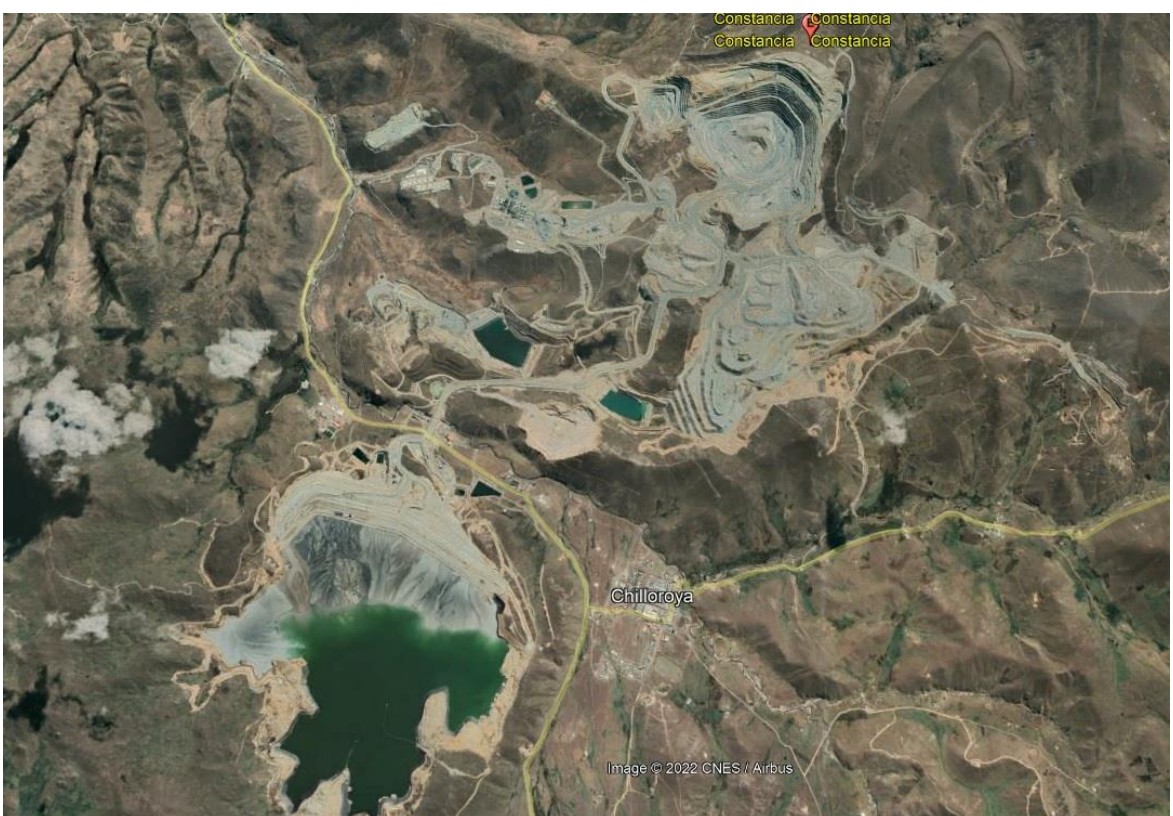

**Figure 52.** Constancia Mining Project Layout.

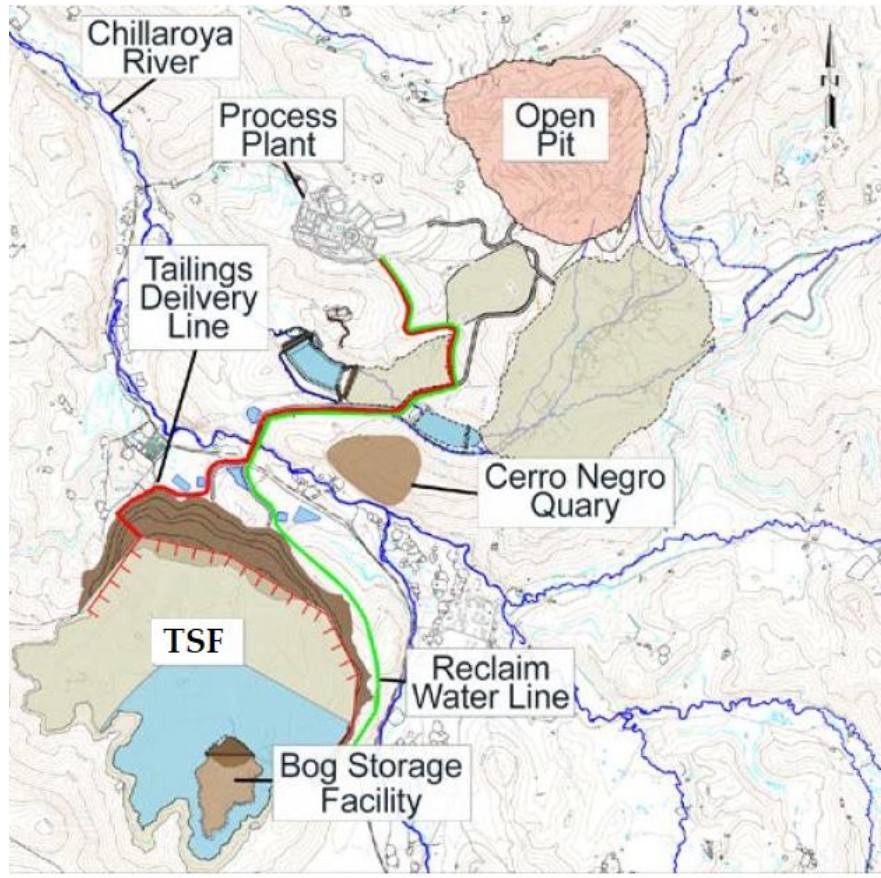

**Figure 53.** Constancia Mining Process Layout.

The dam comprises an initial downstream-method rockfill embankment with sloping upstream core and chimney drain, with subsequent conversion to a centerline-method, supported by an upstream rockfill platform to provide support to the vertical section of core and filter/drain zones (Figure 54) [43].

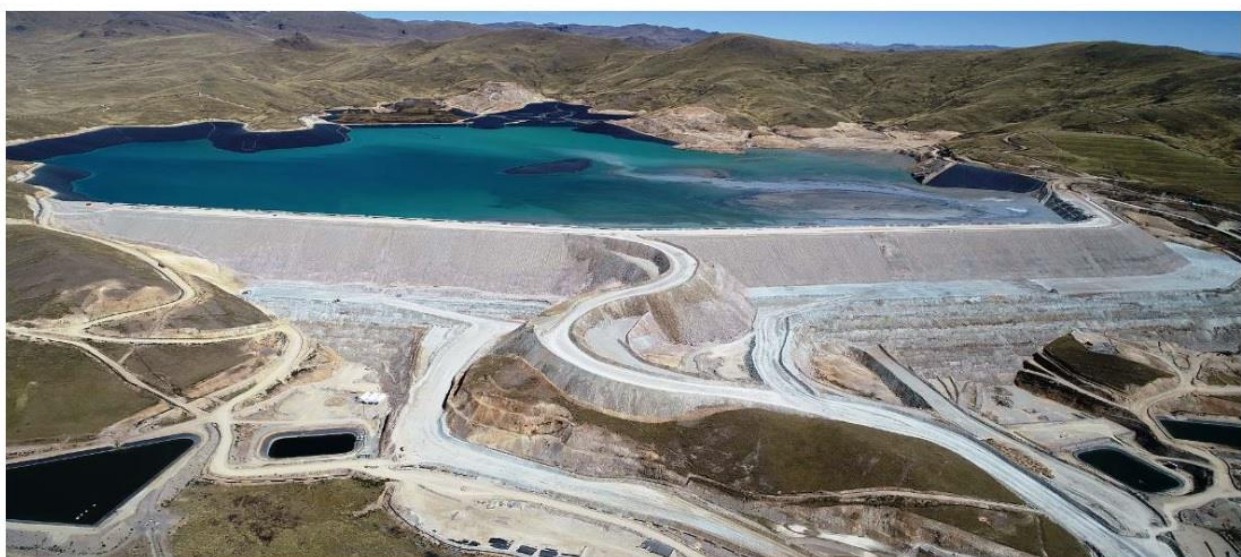

**Figure 54.** Front view Constancia Thickened TSF Mine Waste Rock Dam.

The tailings deposition scheme consists of a series of spigots located on the crest of the dam, which allows the formation of a tailings beach and the distancing of the pond. These controlled tailings discharges allow for the formation of a tailings deposition slope of the order of 1.0 to 2.0% [43].

The water associated with the TSF includes direct precipitation on the impoundment area, water expelled from the tailings due to consolidation. Contact water also includes runoff from upslope areas not intercepted by the non-contact water diversion channel during major storm events. Water will be collected in a supernatant water pond located in the central part of the impoundment. Water in the impoundment dam will be pumped to the plant using a floating barge pump in the supernatant pond [43].

*7.18. Chungar Paste Tailings—Down Valley Discharge—Volcan—Peru*

The Chungar Mine is located in the Department of Pasco, Province of Cerro de Pasco and District of Huayllay, Peru. It is a polymetallic deposit that contains copper, silver, zinc, and lead, an underground working mine, and whose workings are at the 4100 masl [44,72].

The process begins with the pumping of the flotation tailings towards the hydraulic fill hydrocyclone battery, which is carried out with a pumping line of 02 pumps in series with 150 HP of power and another line is on standby. The hydrocyclone battery is composed of 05 hydrocyclones, the underflow of the hydrocyclones is stored in 02 silos of 220 and 240 m$^3$ capacity, to later be sent to an underground mine to be used as hydraulic fill. The hydrocyclone overflow is diverted to transfer tank "A" where it is mixed with water from the underground mine. The overflow mixture plus underground mine water (slurry) is sent to the deep cone thickener 17 m in diameter × 21 m in height (Figure 55) through 01 pumping line (12" diameter HDPE pipe) composed of 03 pumps in series of 200 HP power each, in addition, it has a standby line [44,72].

The discharge of the tailings in paste from the thickener is carried out by means of 02 centrifugal pumps of 150 HP of power, the other a duplex peristaltic pump of 50 HP of power, also, the discharge is carried out by gravity (depending on the characteristics of the tailings in paste), a through the conduction and transport lines with a 10" diameter HDPE pipe, disposing of it in No. 2 and No. 3 TSF (Figure 56) [44,72].

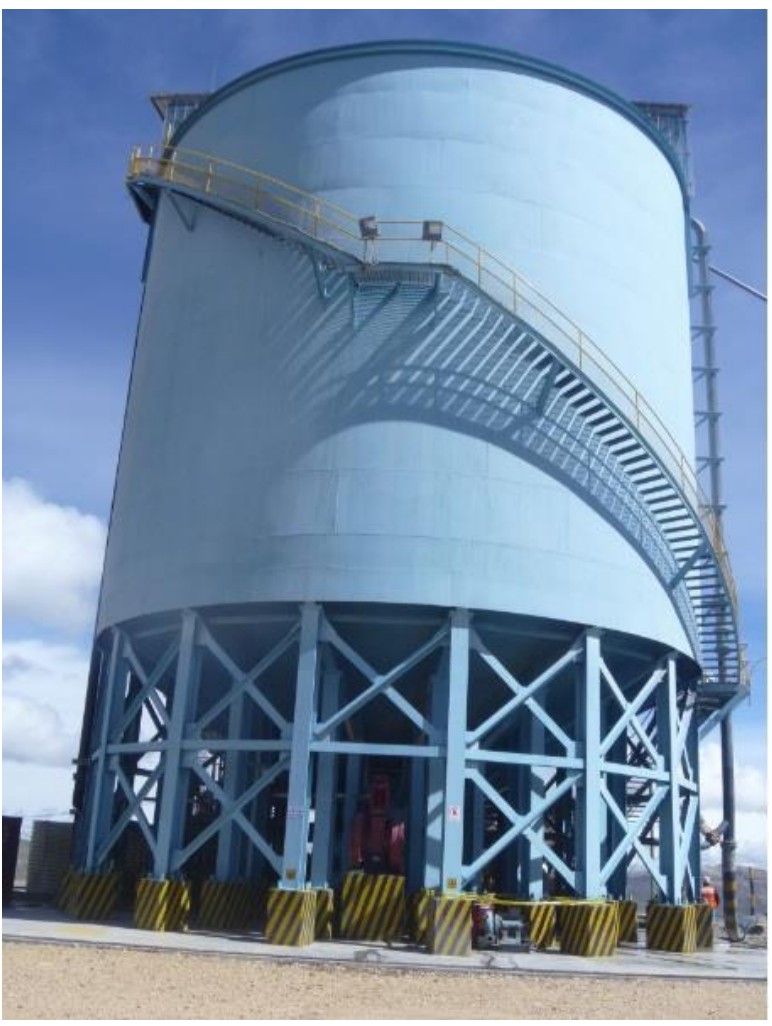

**Figure 55.** Deep Cone Thickener for obtain Paste Tailings—Chungar Thickening Plant.

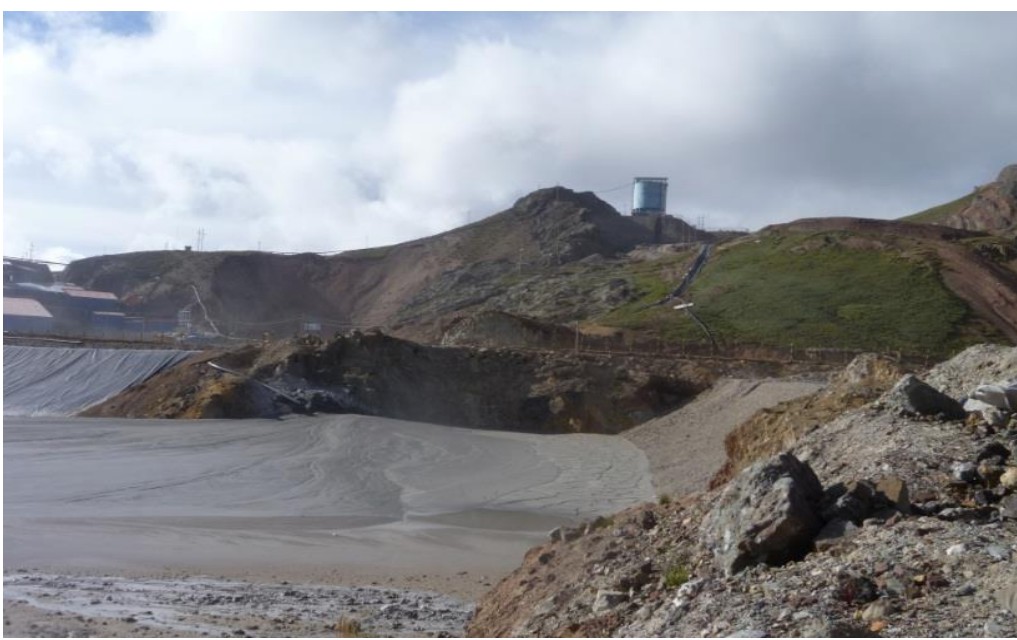

**Figure 56.** Paste Tailings Disposal—Chungar TSF.

The DCT operating discharge flow range is 5500 mtpd, considering the overflow of the hydrocyclone battery plus underground mine water as feed. The solids concentration range of paste tailings for surface disposal is from 71 to 74% solids by weight (Cw), with peaks of 76% solids (Figure 57) [44,72].

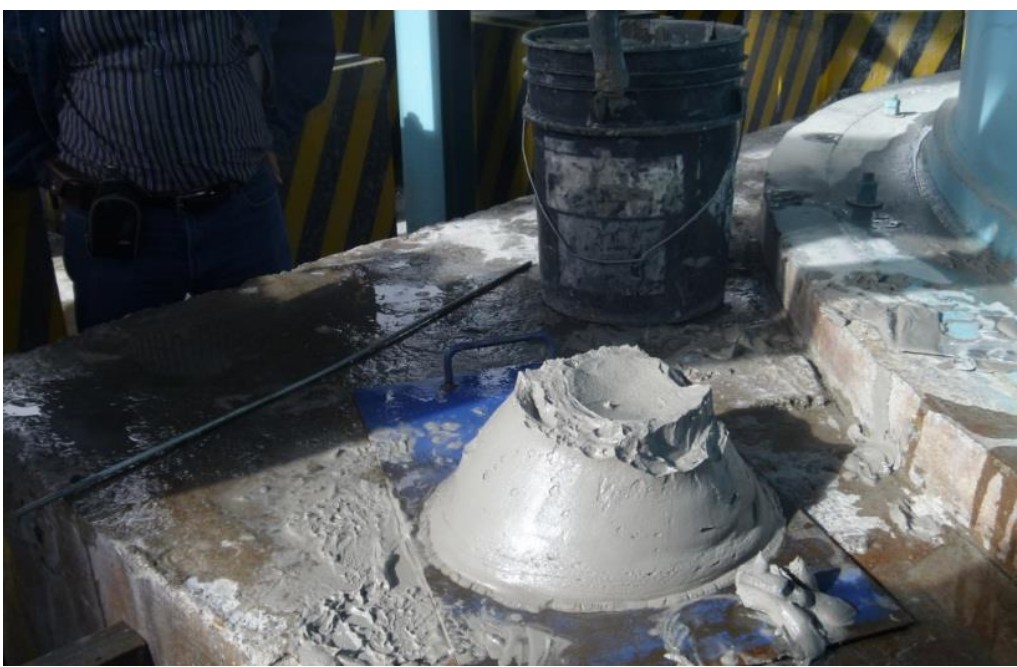

**Figure 57.** Paste Tailings Viscosity and Consistency—Chungar Thickening Plant.

### 7.19. Cobriza Paste Tailings—Down Valley Discharge—Doe Run—Peru

The Cobriza Mine is located in the district of San Pedro de Coris, province of Churcampa, department of Huancavelica, on the left bank of the Mantaro River, in the eastern highlands of Peru, on the eastern flank of the Central Cordillera of the Andes, at 2300 masl [45,73].

The Cobriza mine is an underground mine and is developed from northwest to southeast through underground galleries. The site's reservoir consists of a mantle with highly pyrite characteristics, with slaty and fractured casings. The exploitation of the deposit is by the ascending mechanized cut-and-fill system. The water used in the operations and hydraulic filling of the underground mine gives rise to wastewater that represents an effluent towards the Mantaro River. The mineral concentrator plant was designed to process 3,000,000 tons/year of copper ore with a nominal feed rate of 9000 mtpd, and the froth flotation process is carried out there. The ore previously goes through a crushing and grinding process—classification [45,73].

80% of the tailings generated in the metallurgical process of the mineral from the concentrator plant were discharged directly into the Mantaro River, thus constituting the main problem of environmental contamination of the Cobriza mine, which was identified and documented by Centromin in the Program of Adaptation and Environmental Monitoring (PAMA). It is estimated that the tailings were transported and deposited in the Mantaro River at a rate of approximately 180 tons/hour, thus constituting a total of almost 4500 tons of tailings per day. The disposal of tailings directly to The Mantaro River continued to be managed by Doe Run until 30 June 2004, the date on which tailings dumping operations into the river were halted. In order to meet the objectives defined in the PAMA for the treatment of tailings, Doe Run carried out the studies for the designs of the 3 tailings ponds that currently exist and that are located in the different locations of the La Expansión and Cobriza. The tailings deposits considered are El Platanal, El Limonar and Chacapampa (Figure 58).

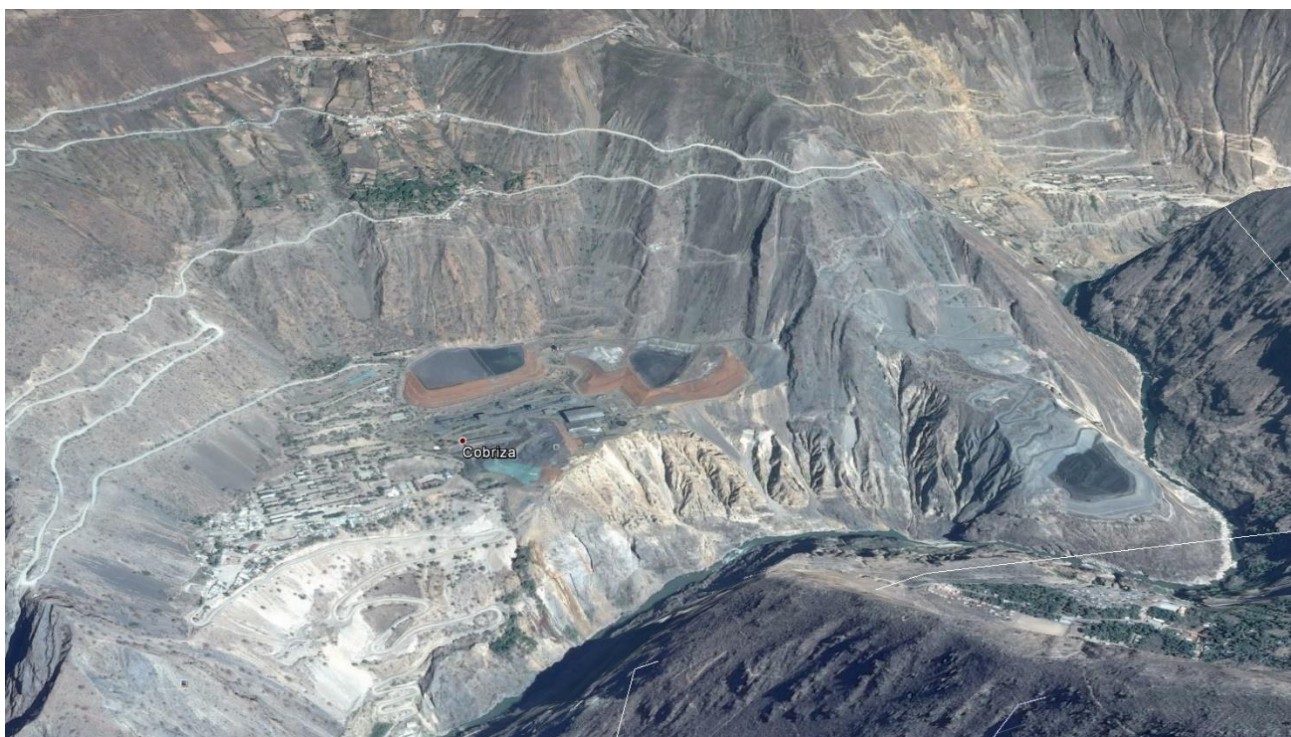

**Figure 58.** Cobriza Mining Project Layout.

Cobriza has decided to implement paste tailings technology, for which it has built a deep cone thickener with a diameter of 14 m and a height of 21 m (Figure 59). The total tailings produced is equivalent to a rate of 9000 mtpd, being classified by a battery of hydrocyclones where the coarse tailings produced are around 3500 tmpd and the fine tailings obtained are around 5500 mtpd. The coarse tailings are used for hydraulic filling of underground mines and the fine tailings are disposed of on the surface using paste tailings technology. The concentrations of solids by weight (Cw) obtained for the tailings thickened in the DCT thickener reach values of the order of 70% [45,73].

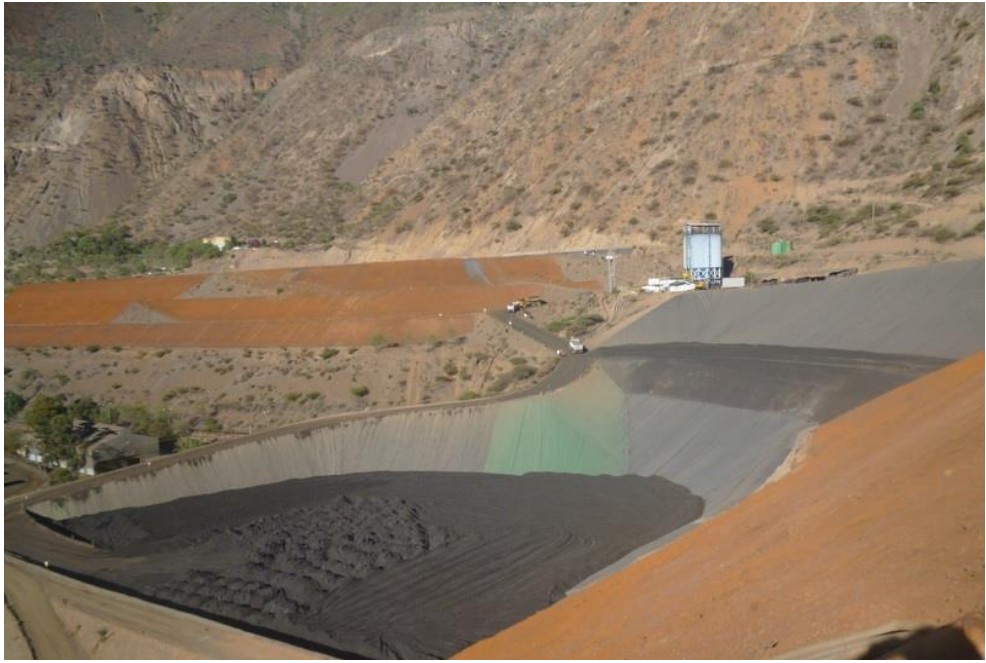

**Figure 59.** Cobriza Paste Thickener and Limonar TSF.

The paste tailings are pumped by pumps and pipes to the high points of the El Platanal and El Limonar deposits, where they are discharged from a series of spigots, which allow it to reach a deposition slope of the order of 3.0 to 4.0% (Figures 59–61) [45,73].

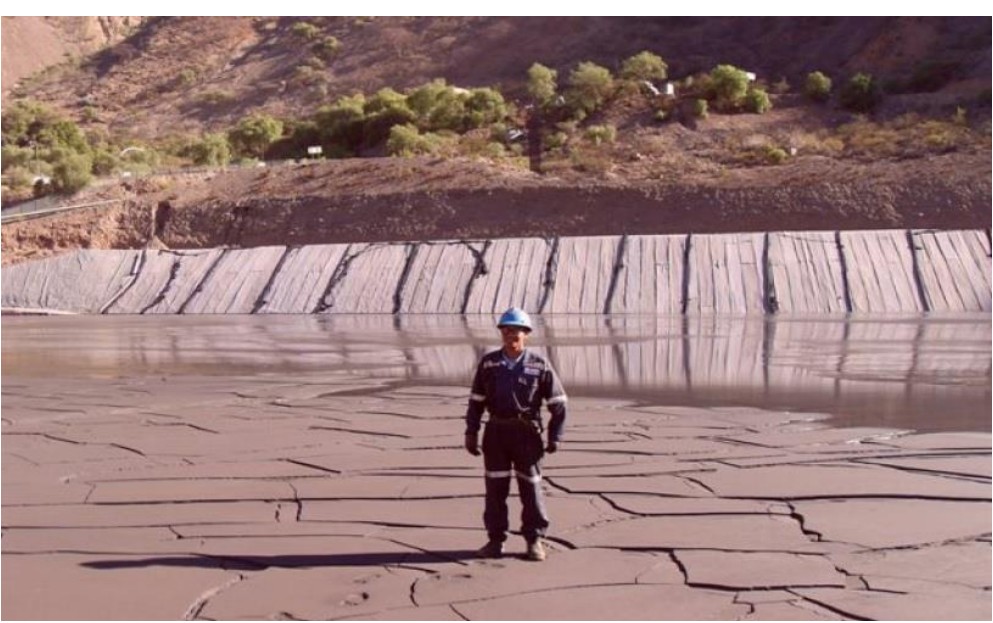

**Figure 60.** Cobriza Paste Tailings Disposal in Limonar TSF.

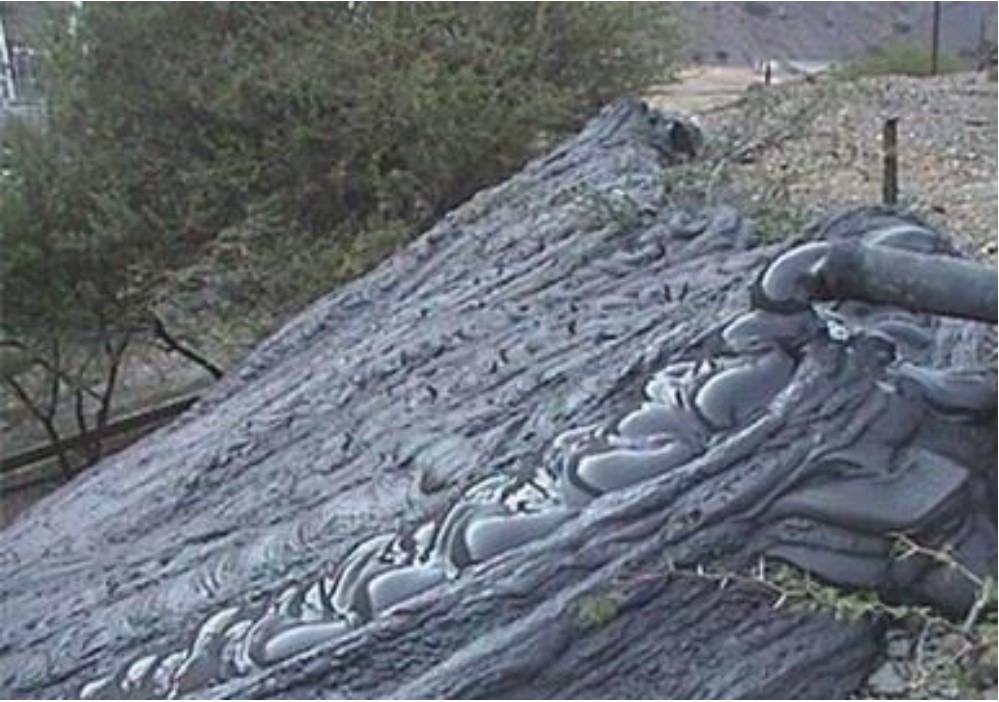

**Figure 61.** High Viscosity in Cobriza Paste Tailings Disposal.

Due to the complex topography of the site and the lack of space to deposit tailings in a deposit, Cobriza decided to mechanically remove the tailings deposited in paste form in the El Platanal and El Limonar deposits with a hydraulic excavator and dump truck to be transported to a new deposit called Chacapampa and place them in compacted layers similar to a dry stack deposit. In this way, the capacity of the El Platanal and El Limonar deposits is released to continue filling them with paste tailings, increasing their useful life (Figure 62) [45,73].

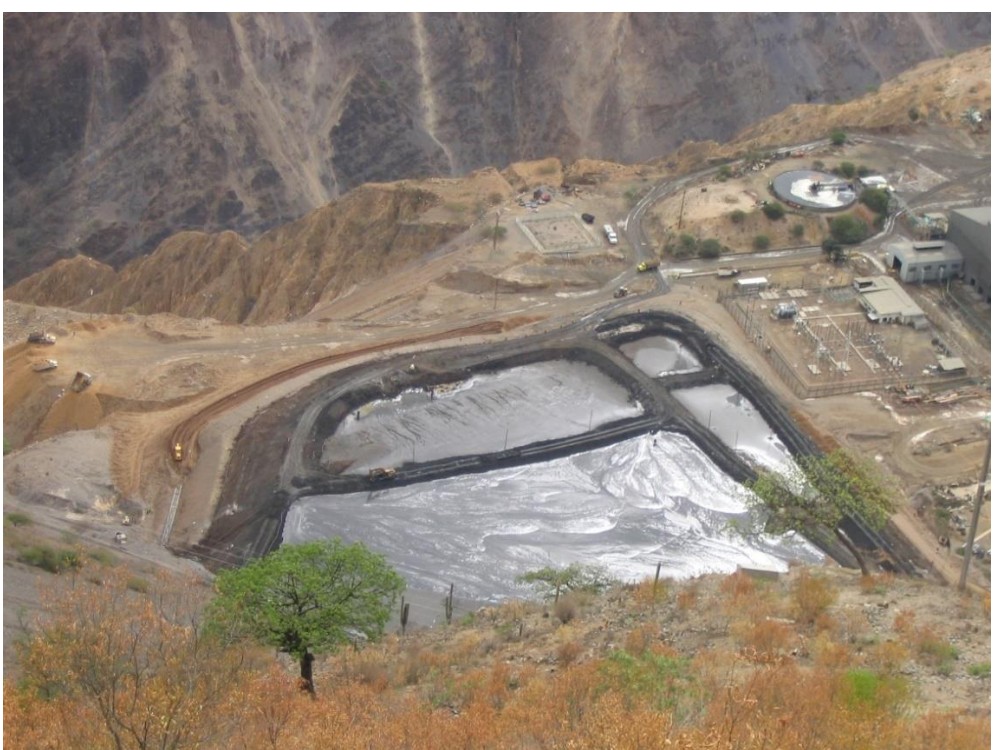

**Figure 62.** Cobriza Paste Tailings Disposal in Platanal TSF.

*7.20. Rumichaca Paste Tailings—Down Valley Discharge—Volcan—Peru*

The Yauli and Rumichaca mining unit is located between 4000 to 4900 masl in the department of Junín, 40 km from La Oroya and 170 km from Lima. The unit is made up of four underground mines and one open pit, whose ore is treated in three concentrator plants, with an installed capacity of 11,400 mtpd. The zone presents mineralogy related to polymetallic epithermal systems, replacement bodies and polymetallic vein systems, as well as mineralization related to porphyries and skarns, which indicates the great potential of the area [46].

The Victoria concentrator plant produces polymetallic mining tailings from the underground mine called Carahuacra at a rate of 8000 mtpd. These tailings are taken to a battery of hydrocyclones that allow the tailing to be classified into the coarse fraction and the fine fraction. The coarse fraction obtained is of the order of 2000 mtpd and is used for hydraulic backfilling of underground mines, while the fine fraction of the tailings is of the order of 6000 mtpd and is taken to a deep cone thickener of 20 m diameter (Figure 63) and deposited in a surface tailings storage facility [46].

The fine tailings are pumped by pipeline from the Victoria concentrator plant to the area called Rumichaca, where the deep cone thickener is located. In this place, the fine tailings are thickened until reaching a concentration of solids by weight (Cw) of around 65%. The paste tailings are then pumped and transported by pipeline to the discharge points located in the upper part of the Rumichaca tailings deposit and thus be able to deposit the tailings with a slope. The deposition slope values achieved in this case are of the order of 2.0%. The tailings are deposited from different discharge points or spigots, allowing the laminar depositing of the tailings in paste, also allowing the tailings to dry and consolidate for certain periods of time (Figure 64) [46].

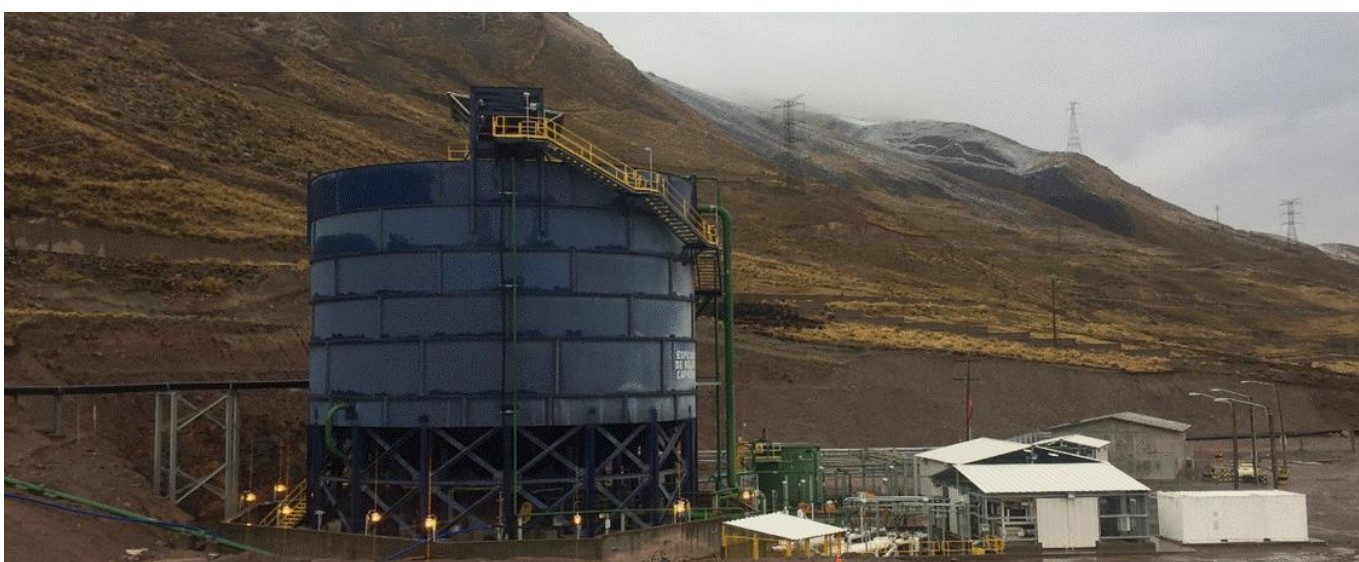

**Figure 63.** Rumichaca Paste Tailings Thickener.

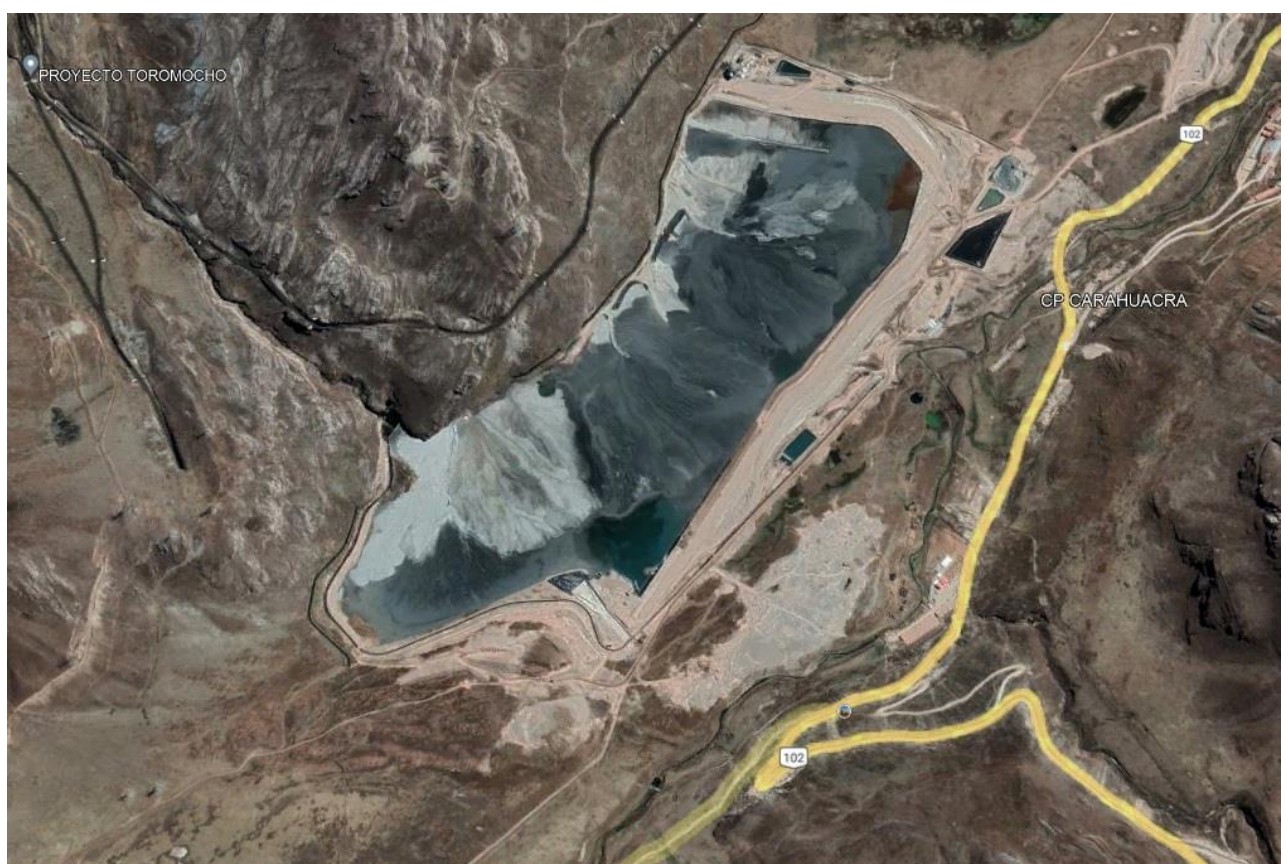

**Figure 64.** Layout View of Rumichaca Paste TSF.

The tailings deposit has a perimeter dam built with borrowed material obtained from nearby quarries, where the material is placed, compacted and the growth of the dam is carried out in stages downstream as the tailings deposit is filled (Figure 65) [46].

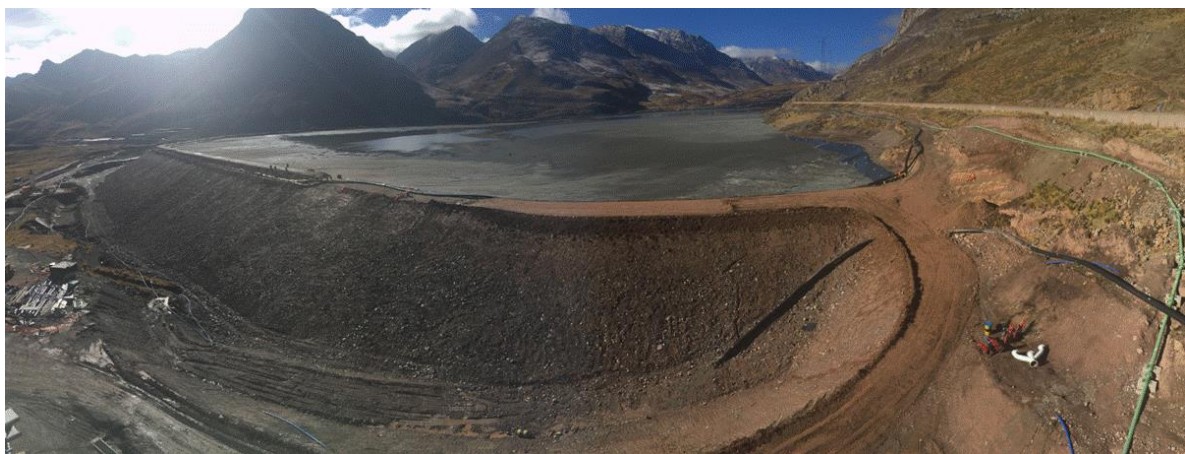

**Figure 65.** Typical Dam Crest View of Rumichaca Paste TSF.

*7.21. Huachuacaja Thickened Tailings—Down Valley Discharge—El Brocal—Peru*

Sociedad Minera El Brocal S.A.A. operated the Colquijirca Mining Unit which is located to 4300 masl in Peru Region Pasco Province Pasco and District Tinyahuarco considering a 296 km of distance from Lima city. Sociedad Minera El Brocal S.A.A. is a polymetallic mining company dedicated to the extraction, concentration and commercialization of Cu, Pb, Zn and Ag [47,74].

Sociedad Minera El Brocal S.A.A. was founded on 7 May 1956, it operates 2 continuous mines:

1.     Tajo Norte: Open-Pit Mine—produces minerals such as Ag, Pb, Zn, and Cu
2.     Marca Punta Norte: Underground Mine—produces Cu

Sociedad Minera El Brocal S.A.A. current mining strategy is to treat 11,000 mtpd of Pb-Zn (Open Pit), and 7000 mtpd of Cu (Underground) [47,74].

The production of tailings is thickened reaching a solid concentration by weight (Cw) of 65% [47,74]. The thickened tailings generated are storage in the Huachuacaja tailings storage facility (Figure 66) which was designed considered the following parameters:

- Tailings depositional slope of 2.0%
- Tailings discharged from west side of the valley and from tailings dam
- 28 discharge tailing points, which were built every 200 m approx. (Figure 61)
- Dry tailings deposition density of 1.59 t/m$^3$
- Average production of tailings: 16,740 mtpd
- Annual average volume of the deposit tailing pond: 3.0 million m$^3$
- Capacity: 123 million tons
- Estimated lifetime: 20 years
- Spigots (every 200 m) of diameter of 8" from HDPE SDR-13

The thickened tailings are discharged from a series of spigots located on the perimeter of the tailings deposit, in such a way as to achieve the maximum tailings deposition slope, which reaches a value of around 2.0% (Figure 67).

The tailings deposit dam is built with borrowed material in stages with a construction method in a downstream direction [47,74].

The water associated with the TSF includes direct precipitation on the impoundment area, water expelled from the tailings due to consolidation. Contact water also includes runoff from upslope areas not intercepted by the non-contact water diversion channel during major storm events. Water will be collected in a supernatant water pond located in the central part of the impoundment. Water in the impoundment dam will be pumped to the plant using a floating barge pump in the supernatant pond [47,74].

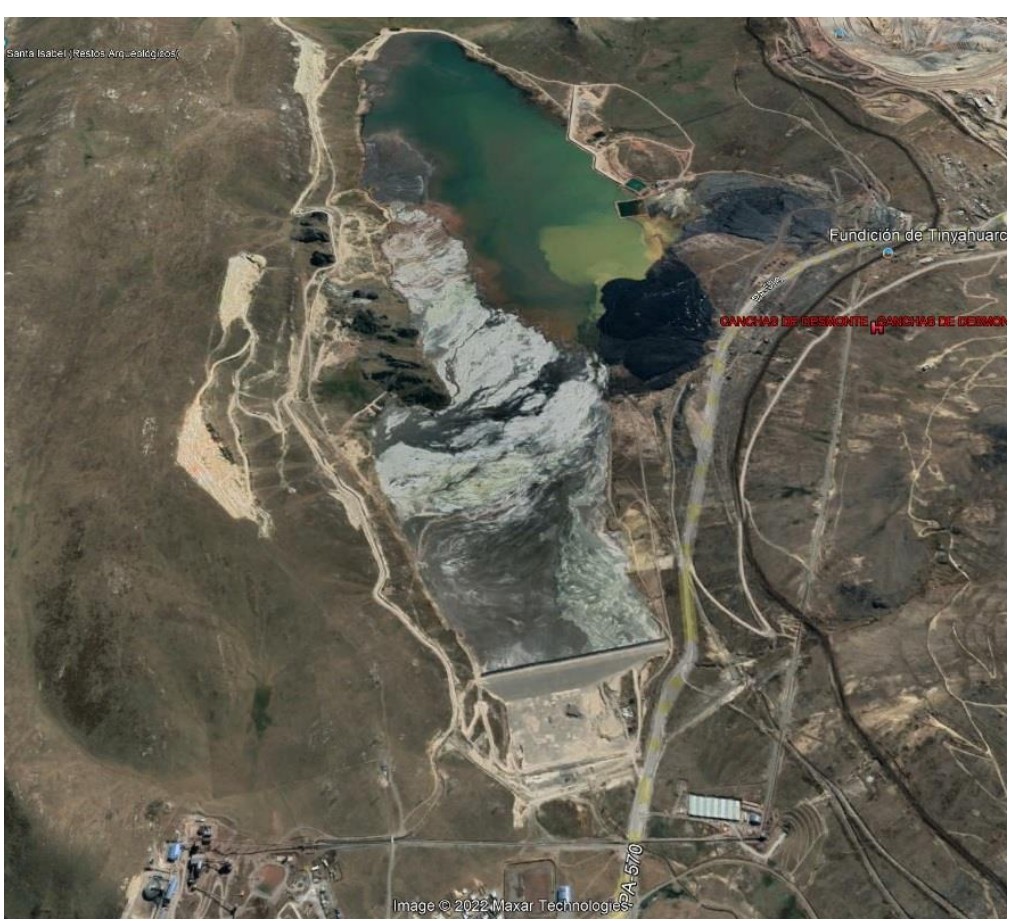

**Figure 66.** Huachuacaja Thickened TSF Layout.

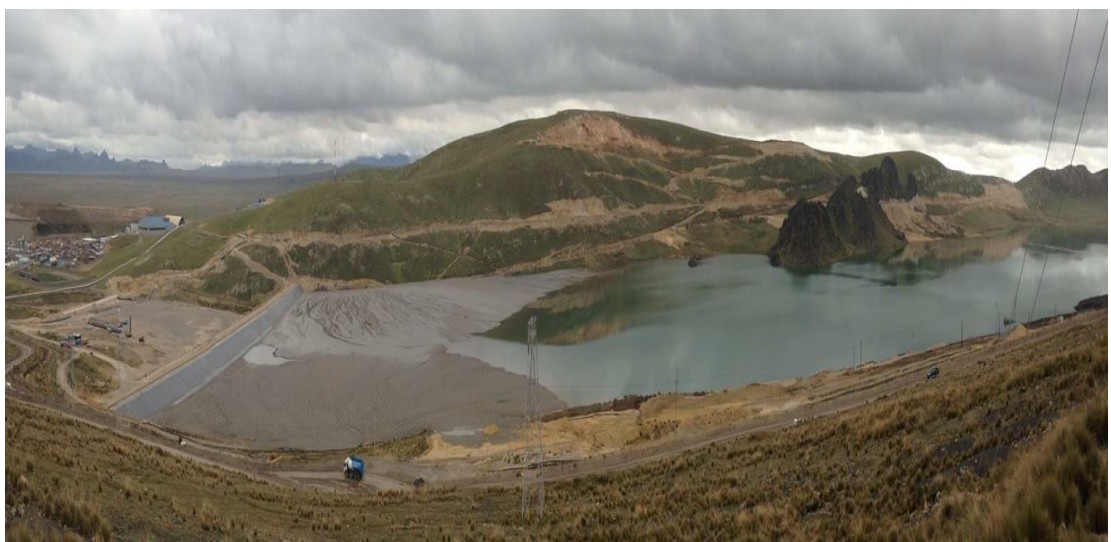

**Figure 67.** Huachuacaja Thickened TSF Side View.

*7.22. Cerro Corona Thickened Tailings—Down Valley Discharge—GoldFields—Peru*

Gold Fields La Cima S.A.A. (GFLC), a subsidiary of Gold Fields Limited, owns the Cerro Corona mine, an open-pit copper mine with significant gold content. The mine is located in northern Peru in the province of Cajamarca, approximately 600 km north-northwest of Lima, approximately 80 km by road north of the departmental capital of Cajamarca and approximately 1.5 km west-northwest of Hualgayoc. The elevation of the mine site ranges

from approximately 3600 to 4000 masl. The mine has been in production since 2008 and it is estimated that the current mineral reserve will be depleted in 2023. The Cerro Corona deposit is exploited using the open pit surface mining method (Figure 68) [48,75].

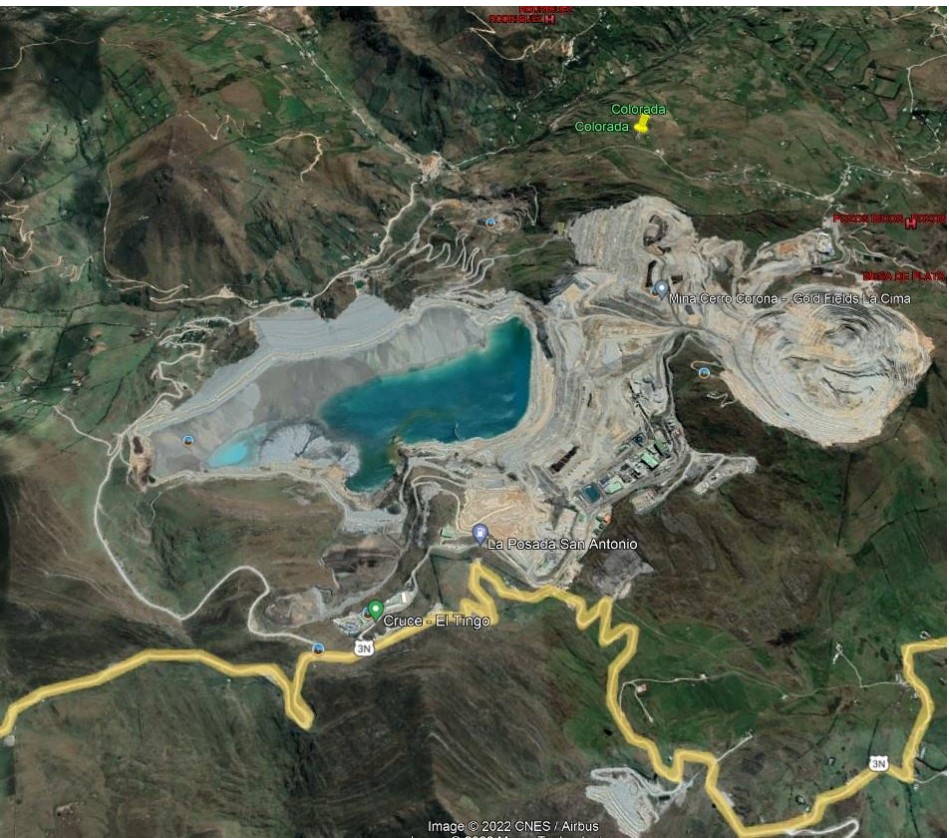

**Figure 68.** Cerro Corona Mining Project Layout.

Tailings produced by the Cerro Corona processing plant are deposited in the Tailing Storage Facility (TSF) which is located northwest of the plant site (Figure 69). The TSF stores both rougher scavenger tailing (RST), which is deposited sub-aerially, and cleaner scavenger tailing (CST), which is generally deposited sub-aqueously to reduce the potential for acid generation. RST makes up approximately 95% of the tailing stream. It is thickened to a solids content by weight (Cw) of approximately 55% prior to disposal. The RST tailing is then conveyed to the TSF via HDPE tailing delivery pipelines and disposal into the TSF through a number of spigots which run along the upstream face of the TSF dam. Water is removed from the TSF impoundment and reclaimed in the mine process circuit by a floating decant barge located in the impoundment (See Figure 69) [48,75].

The TSF dam is constructed by centerline construction method, and the dam cross section consists of low-permeability core materials (Zone 1 and 5) placed between upstream and downstream rockfill (Zone 2, 2B and 2C). The upstream rockfill (mine waste rock) was incorporated into the design to provide upstream stability based on engineering analyses considering the anticipated dam raise heights and measured tailings strengths. The core and rockfill materials are separated by drain and filter zones (Zone 3 and 4) placed immediately downstream of the low permeability materials to serve as transition zones to reduce the potential for migration of the core materials into the rockfill. To decrease seepage rates from the facility, a grout curtain has been installed in the foundation materials along the axis of the dam [48,75].

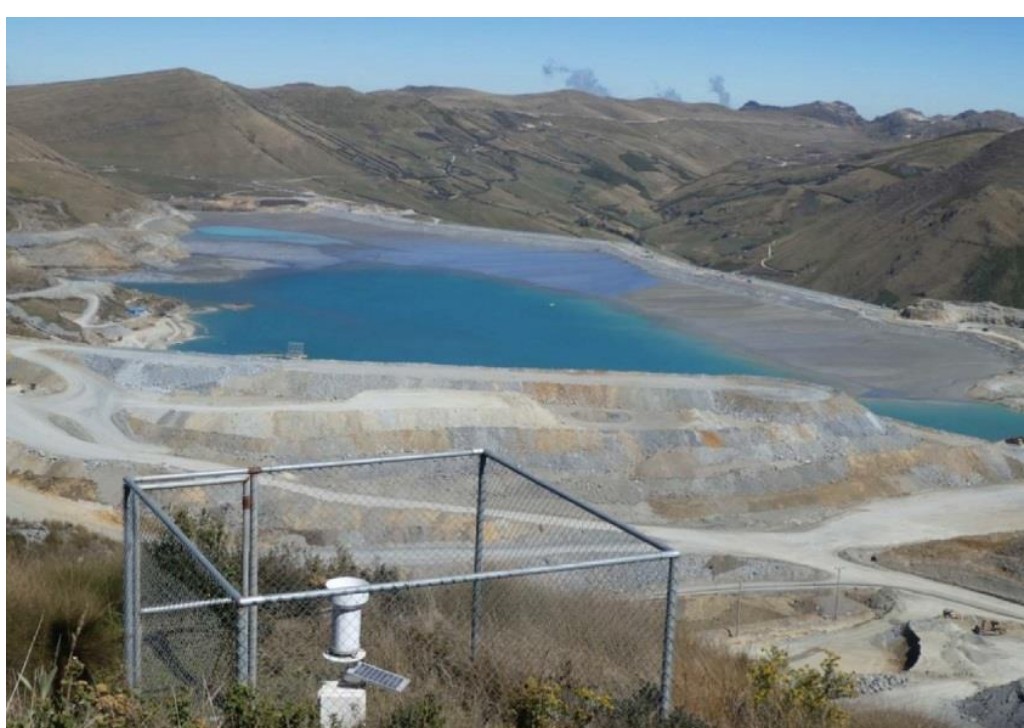

**Figure 69.** Cerro Corona Mining Thickened TSF.

*7.23. La Quinua Thickened Tailings—Down Valley Discharge—Newmont—Peru*

Yanacocha is South America's largest gold mine, located in the province and department of Cajamarca, approximately 800 km northeast of Lima, Peru. Yanacocha's operations is operated by Newmont and the facilities are situated between 3500 to 4100 masl with development activities in four primary basins. The operation is a joint venture between Newmont (51.35%), Minas Buenaventura (43.65%) and Sumitomo Corporation (5%).

Yanacocha produces ore from two open pits and is reclaiming three. First production came from the Carachugo pit, followed by Maqui Maqui in 1994 and the San José Sur pit during 1996. Cerro Yanacocha, opened in 1997, and La Quinua, where production started during 2001, provide the current production.

Since the oxide ore is porous, run-of-mine ore can be heap-leached without crushing and the solution treated by the Merrill Crowe process. Cyanide solutions are rendered harmless by the Inco process. Subsequent pit development required three more leach pads and two more process plants, while working La Quinua necessitated adding a 120,000 mtpd crushing and agglomeration facility in 2001.

To treat high grade sulphide ore Yanacocha implemented a milling process at the La Quinua operation. However, limited sites were available for the tailings storage facility. To best utilise the available area the tailings storage facility is fully contained within an active heap leach pad. Consequently, high concentration thickened tailings were required to reduce the amount of water placed in the facility (Figure 70). The tailings are produced at 17,000 mtpd, then are thickened at the mill plant in a high-rate thickener before being pumped to the storage facility by a train of centrifugal slurry pumps [49]. Some characteristics of the thickened tailings management at La Quinua are:

- Pipe material: Steel
- Pipe diameter: 300 mm
- Pipeline Length: 4 km
- Pump station: Centrifugal pumps
- Solid concentration by weight (Cw): 65 to 71%
- Yield stress: 8.5 Pa

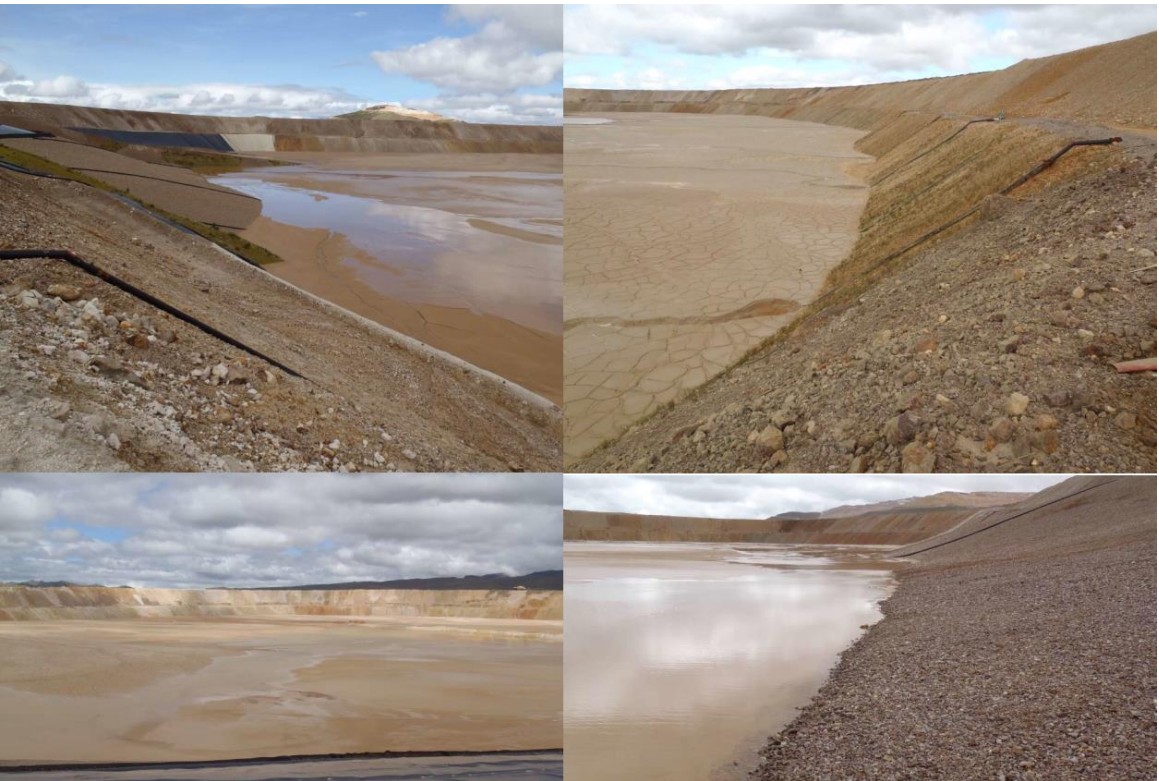

**Figure 70.** La Quinua Thickened TSF.

The thickened tailings are deposited in the leach pad facility, this is a unique facility of its kind, which requires controlled management of the construction and operation of the leach pad and tailings disposal [49].

## 8. Opportunities for Tailings Management Improvement in Chile, Peru and Worldwide

There are many challenges that must be overcome in Chile, Peru and worldwide to achieve a safe tailings management state-of-the-art outcome. Traditional tailings disposal methods create environmental problems as they can: (i) take up large surface areas; (ii) be highly visible; (iii) entrain and possibly store large volumes of water; (iv) seep unwanted water in-to the ground; (v) release TSF drainage into surface streams; and (vi) cause dust problems [76,77].

Avoiding these issues, and their associated risks, requires a commitment to rigorous planning and application of leading practices over the full mine life cycle. Such outcomes also require foresight and recognition that tailings facilities can incur environmental and social costs in the long-term if leading practice principles are not heeded [76,78]. TSFs must meet operator, public health and safety, community, and environmental protection objectives. These objectives can only be met if TSFs are designed, operated, closed and rehabilitated to a level of risk that is acceptable to stakeholders for the full operating life of the facility and beyond [11,76]. A systematic approach to effective tailings management is therefore advocated. Management strategies need to be risk-based and account for the viewpoints and expectations of the communities in which companies operate [79]. The main tailings-related risks to people and the environment can be characterized for the operational and closure phases. The principal objective of a TSF is for tailings solids and any stored water to remain contained.

The risk-based approach applied to tailings management must have sufficient flexibility to allow changing circumstances to be managed. These changes could involve routine and anticipated TSF raises, unforeseen expansions, or bringing online completely new facilities and/or new disposal methodologies [79].

The definition of value according to the theory of value is any action or proceeding to be conducted that leads to realize a certain goal. Under this view and having defined a spatial-temporal scale, the evolution of tailings management in Chile and Peru during the 20th century has made progress on technical and economic aspects, with recent significant environmental progress over the past two decades. This undoubtedly is valuable, and efforts are recognized for managing tailings controlled and physically stable. They are still pending issues in the pursuit of an integrated or holistic management of tailings management, which requires an interdisciplinary approach involving professionals who can support the physical, geochemical and hydrological stability of tailings deposits, with measures and techniques for the stages of design, operation and closure. Hard work and dedication have been carried out in technical, economic, risk, environmental and social studies, showing that this has led to better decisions recently, which reduce the negative environmental impact, allowing anticipating potential damage and to mitigate or eliminate adverse effects.

## 9. Discussions—Strengths and Limitations of Thickened Tailings from Practical Experiences Presented

Thickened and paste tailings are deposited hydraulically, or loosely, and beach, or settle, at somewhat steeper slopes than conventional tailings slurry. In theory, the beach slope can be up to 4.0%, however, in practice steep slopes are only achieved for a short distance and the remaining beach is sloped at less than 2.0%. The steeper beach slope of thickened and paste tailings storage facility, compared to a conventional slurry beach, provides an opportunity to store tailings above the dam elevation, which reduces the footprint and height of the dam.

Thickened and paste tailings should be, by definition, largely non-segregating (i.e., fine and coarse particles do not separate during deposition), however minor segregation could still occur depending on the tailings particle size distribution and solids content at deposition.

Compared to conventional tailings storage facilities, less bleed water and consolidation water is released from thickened and paste tailings deposited. This water, along with precipitation and runoff, collects in ponds on the tailings surface often near the perimeter dams, or can be directed off the surface to external collection ponds. The reclaim pond size on the tailings surface is dependent on topography, tailings surface geometry, hydraulic design of water conveyance structures, and seasonal climate variations.

The capital cost of specific equipment, such as thickeners, pumps and pipelines and the operational costs, including high dosages of flocculants, high energy consumption and specialized operators usually make thickened tailings alternatives more expensive than conventional disposal [80]. The main economic benefit of using thickened tailings is associated to with a smaller embankment and high water recovery from tailings. These advantages can be achieved in flat and abrupt topographies, managing the thickened tailings deposition angle in the impoundment. When savings in embankment (dam) and water recovery costs can be realized, thickened tailings disposal is a very attractive option.

The following paragraphs presents the strengths and limitations of thickened tailings technology based on the practical experience recorded in Chile and Peru.

### 9.1. Strengths

- Higher water recovery during processing, less water to be managed at the TSF.
- May be non-segregating, producing a tailings product with potentially low hydraulic conductivity.
- Thickened and paste tailings can be more easily closed as a "dry" facility than a conventional TSF.
- Failure if it occurs, would likely be local slumping and consequences would be restricted to the local area.

*9.2. Limitations*

- High-density thickeners require operational attention and are subject to system "upsets" from tailings variability, gradation or operator error.
- It could take months to years to optimize the thickening system to produce a consistent tailings product and the achieved solids content (Cw) is often at least 5% lower than the design target.
- Positive displacement pumps (PD Pumps) may be required for tailings transportation, which are more expensive and more challenging to operators.
- Beach slopes are difficult to predict and will vary depending on operational practices, tailings properties and weather.
- Significant drying time (if required for physical stability) is often not achieved in wet climates and may require a large drying area and rotation of the tailings discharge points (spigots).

## 10. Conclusions

The way in which tailings are managed reflects the history, the regulatory framework and the environment of the country and locale of the mine. Despite many attempts to find an environmentally friendly strategy for tailings management that considers a balanced relationship between society—ecosystem, there is no world-wide agreement regarding the best available practices of tailings management [76,81]. This article reviews the evolution of thickened and paste tailings management in Chile and Peru, current practices, and changes that could or may need to be made to improve practices, as a response to local environmental conditions. The paper also examines current development of an incipient progress of thickened and paste tailings management, changing the conventional tailings management focus. The paper also defines thickened and paste tailings management as one of the best available technologies (BATs), briefly examining case histories of Chilean and Peruvian tailings facilities using this technology, which have achieved benefits such as: (i) reducing the makeup water supply, (ii) reduction of failure risks in seismic zones, thus improving TSF physical stability, (iii) maximum water recovery, (iv) and obtaining smaller TSF footprints and effective dust control, all of which reduced negative environmental impacts. These benefits are a good reason to promote to new or existing large mining operations a shift from conventional slurry tailings disposal facilities to alternative solutions incorporating thickened and paste tailings disposal facilities, which day by day are more accepted, similar to an environmentally sustainable solution.

The conditions of water scarcity in the Atacama Desert (northern Chile and southern Peru) and a shared use with the communities, has resulted in mining operations implementing more efficient technologies for the management of mining tailings, reducing the maximum water losses, this is how the technology of thickened tailings and paste has been an attractive alternative. On the other hand, in Peru, the demands of the community to avoid the contamination of groundwater with leaks from the tailings storage facilities and to reduce the risk of generating acid rock drainage in the cyclone tailings sand dams has allowed the acceptance of the application of thickened tailings technology [82].

Considering the data presented in this article, it is possible to conclude that the countries of Chile and Peru are currently world leaders in the implementation of thickened and paste tailings technology, considering small, medium and large-scale mining operations. In addition, considering the cases of practical studies presented, it is possible to mention that the good performance and application of thickened tailings technology on a large industrial scale is a reality. The cases presented in this article demonstrate that it is possible to apply thickened tailings technology with productions of the order of 100,000 mtpd or greater. In this scenario, there is still a need for more reliable equipment for paste tailings thickening plants on large scale specifically, focusing in the tailings water recovery enhancing for its reuse in mining processing.

The implementation of Codelco's Talabre TTD TSF project for the production of tailings of 400,000 mtpd will undoubtedly mark a relevant historical milestone in the application

of Thickened Tailings Disposal (TTD) technology, being a deposit of dewatered tailings of great magnitude never seen before.

Each mine is unique with respect to its environment, material properties, water supply, mineral process used, environmental obligations and energy cost. Therefore, it is difficult to prescribe a tailings disposal method; each case must be evaluated on its own merits. However, considering the failures of tailings facilities registered in recent years worldwide with conventional tailings technology, it has questioned the communities and authorities about the safety of having dams of large dimensions and heights, this has positioned as a safer and more feasible alternative to thickened and paste tailings technology. Thickened and paste tailings allow a much smaller amount of water to be stored in tailings facilities, considerably reducing the risks of liquefaction, piping (internal erosion by seepage) and overtopping compared to conventional tailings technology. Encouragingly, technologies to recover more water from tailings for reuse in the process near the concentrator plant are advancing rapidly, and where water is a scarce resource, environmental risks are high, or the mining process justifies reagent recovery or metal values in the water, safe mine tailings management can be achieved.

The scope of this article has been to carry out a review of the practical cases of application of thickened tailings technology in both Chile and Peru. The present investigation is limited to this geographical area of the world that represents an important part of the world copper mining production, but it allows leaving the possibility to carry out a future investigation considering integrating more cases from other countries of the world.

A better environmental perception about TTD of authorities and communities, considering that this technology allows to satisfy the needs of stable and safe TSFs, make the TTD be more acceptable, popular and one of the best available technologies (BATs) for operations with mine tailings.

**Author Contributions:** Conceptualization, C.C.V. and A.M.P.; formal analysis, C.C.V.; investigation, C.C.V.; resources, A.M.P.; writing—original draft preparation, C.C.V.; writing—review and editing, C.C.V.; visualization, C.C.V.; supervision, A.M.P. All authors have read and agreed to the published version of the manuscript.

**Funding:** The research is funded by the Research Department of Catholic University of Temuco, Chile.

**Data Availability Statement:** The data presented in this study are available on request from the corresponding author.

**Conflicts of Interest:** The authors declare no conflict of interest.

## Abbreviations

| | |
|---|---|
| TSF | Tailings Storage Facility |
| BATs | Best Available Technologies |
| TTD | Thickened Tailings Disposal |
| PTD | Paste Tailings |
| CT | Conventional Thickener |
| HRT | High Rate Thickener |
| HDT | High Density Thickener |
| HCT | High Compression Thickener |
| DCT | Deep Cone Thickener |
| Cw | Slurry tailings solids content by weight |
| mtpd | Metric tonnes per day |
| DVD | Down Valley Discharge |
| CDD | Cell Dyke Disposal |
| PD Pumps | Positive Displacement Pumps |
| masl | Meters above sea level |
| U/F | Thickener Underflow |
| FR | Thickener Feed Solid Rate |
| FD | Thickener Flocculent Dosage |

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
