# Peer review of "Sustainable Management of Thickened Tailings in Chile and Peru: A Review of Practical Experience and Socio-Environmental Acceptance"

_sustainability, doi:10.3390/su141710901_

Round 1
Reviewer 1 Report
Good work, however
it would be nice to have a better quality of illustrations
Reviewer 2 Report
Dear respected authors,
1. The study's main aim and content have been briefly reflected in the Abstract section.
2. The keywords should be selected based on the phrase that has been used frequently in the text. For instance, the keyword of “Sustainable Development” has not been used in the text at all. Additionally, the words “Environment” and “Community” are general words. It is suggested to use more specific and appropriate ones for this study, considering the aims of selecting Keywords in a scientific article.
3. Defining Acronyms/Abbreviations and using them is appropriate for those phrases that have been mentioned frequently in the text. According to this issue, defining “BAT” as the “best available technology” is unnecessary. Additionally, “tailings storage facility” should be mentioned in the Introduction section for “TSF” again. Another instance is in section 3 where “FD” has bend defined twice in the first paragraph, or in section 5, where “DVD” has been defined at least three times. The manuscript should be seriously checked considering these issues. There are a lot of such issues. Finally, it is recommended to make a nomenclature list before the Introduction section to mention all the Acronyms/Abbreviations, as the manuscript is too long.
4. It is recommended to sort the references according to their sequence of appearing in the text. For clarity, the first reference number in the text is [2] and the second one is [4]. Using EndNote, Mendeley, or similar ones, may facilitate achieving this issue. Additionally, considering this manuscript as a review article, more studies among the literature should be covered in the text. It is easy to find that many of the listed items in the Reference list are related to the website addresses of the photos used in this study. Such an issue decreases the scientific level of the manuscript, and based on that, it is more recommended to use scientific articles to analyze the data, obtained from those websites, scientifically.
5. The statement of the technologies has been well articulated in the Introduction section but the aim of the study, and the necessity of having (publishing) this study should be highlighted in this section.
6. In the second paragraph of the Introduction section, the “tail-ings deposits” should be corrected as “tailings deposits”. Similarly for “ad-equate” in that paragraph. Please check the whole text for prevention of such issues if there is no intention to keep them as they are.
7. Those figures include more than one shape, photo, graph, process, etc. like Figures 1, 6, 7, 9, and 11 should be explained in detail to ease potential readers’ understanding.
8. The content of Table 2, which shows a comparison between different thickened tailings management technologies, needs more explanation and discussion related to the comparisons, and the obtained conclusion for such comparisons should be reported. Moreover, the other similar tables should be explained considering this point of view.
9. As the manuscript is too long, figures like Figures 2 to 5, Figure 8, and Figures 14 to 64, which include just a photo of a location, tool, or equipment, can be moved to the appendix or reported as a supplementary document.
10. Discussion and Conclusion sections are too short, considering the number of covered technologies, projects, obtained data, etc. in this study. Both the sections should be extended. Moreover, a separate paragraph for the limitations of the research and for the potential future studies should be highlighted in the Conclusion section.
11. The text needs a minor grammatical check.
Reviewer 3 Report
The article name is “ Sustainable Management of Thickened Tailings in Chile and Peru: A Review of Practical Experience and Socio-Environmen-tal Acceptance”
The evaluation of the article is as follows;
In the introduction part of the article, the purpose and scientific contribution of the article should be written.
In general, the current situation is explained.
Round 2
Reviewer 2 Report
Dear respected authors,
Thanks for your patience and the precise corrections and modifications to the manuscript. According to the reviewer's point of view, the manuscript's revised version is worth publishing in the respected journal of Sustainability.